# Neural encoding of instantaneous kinematics of eye-head gaze shifts in monkey superior Colliculus

A. John van Opstal [1]✉

The midbrain superior colliculus is a crucial sensorimotor stage for programming and generating saccadic eye-head gaze shifts. Although it is well established that superior colliculus cells encode a neural command that specifies the amplitude and direction of the upcoming gaze-shift vector, there is controversy about the role of the firing-rate dynamics of these neurons during saccades. In our earlier work, we proposed a simple quantitative model that explains how the recruited superior colliculus population may specify the detailed kinematics (trajectories and velocity profiles) of head-restrained saccadic eye movements. We here show that the same principles may apply to a wide range of saccadic eye-head gaze shifts with strongly varying kinematics, despite the substantial nonlinearities and redundancy in programming and execute rapid goal-directed eye-head gaze shifts to peripheral targets. Our findings could provide additional evidence for an important role of the superior colliculus in the optimal control of saccades.

[1] Section Neurophysics, Donders Centre for Neuroscience, Radboud University, Nijmegen, The Netherlands. ✉email: john.vanopstal@donders.ru.nl

A saccadic eye-head gaze shift ($\Delta G$) is the rapid directional change of the fovea in space and is determined by the sum of the changes in orientation of the eye-in-head and the head-on-body: $\Delta G(t) = \Delta E(t) + \Delta H(t)$[1,2]. The gaze-control system of human and non-human primates is optimally suited to reorient the fovea as fast and as accurately as possible to a peripheral target to allow vision to identify objects with high resolution during intermittent fixations.

Programming a gaze shift is a redundant task, as it can be generated by many combinations of eye- and head contributions. Yet, under controlled initial conditions the system appears to select quite reproducible movement strategies. It has therefore been hypothesized[3–9] that saccadic gaze shifts may result from an effectively optimal control principle that minimizes a total performance cost by trading off speed, accuracy, and control effort.

Because of the different plant dynamics of eyes and head and the eyes' limited oculomotor range (roughly 35°), not all eye-head combinations are possible, or equally efficient, in reorienting gaze. Typically, small gaze shifts are associated with small head movements, and large gaze shifts with larger head movements, but the latter also depends on the initial eye-in-head orientation[10–15]. Whenever a head movement contributes to the gaze shift the gaze peak-velocity is reduced because of the head's inertia and because the eye- and head controls are thought to interact[16–25].

Thus, eye-head gaze shifts do not follow the stereotyped nonlinear main-sequence relationships of visual evoked head-restrained eye-only saccades[26]. Instead, any given gaze-shift can be associated with a wide range of gaze peak velocities and durations[14,27–30]. In this paper, we investigate the potential contribution of the midbrain Superior Colliculus (SC) to the encoding and generation of visually evoked eye-head gaze shifts. In what follows, we first describe the collicular mechanisms underlying our concepts in more detail.

The SC contains a topographically organized motor map of saccadic eye movements[31–34]. Prior to and during an eye saccade, a large population of cells encodes amplitude and direction by its location within the map[32,35–37].

It has been suggested that the saccade kinematics may be encoded by the SC cells as well (e.g[36,38,39]), albeit that there is controversy regarding the nature of this encoding. For example, some studies proposed that the SC could influence the saccade kinematics indirectly through a global feedforward gain control[26,40], whereby higher mean firing rates lead to faster saccades than lower rates. Other proposals assumed precise instantaneous neural control of the saccade kinematics through dynamic local feedback of eye velocity to the motor map[20,38,41–44]. However, several studies, including our own, argued that local dynamic feedback does not explain the SC firing patterns during saccades[45–50].

We formulated a simple linear computational model, in which each spike in the burst from each recruited neuron, $k$, contributes an infinitesimal movement, $\Delta \vec{m}_k$, to the planned saccade[33]. This mini spike-vector is solely determined by the cell's location in the map and specifies its horizontal/vertical connection strengths with the brainstem burst generators via the SC-to-brainstem efferent mapping[32,51]. According to this dynamic ensemble-coding model, the instantaneous desired saccade trajectory is then simply encoded by the cumulative linear sum of all SC spike vectors:

$$\Delta \vec{E}_D(t) = \sum_{k=1}^{N_{POP}} \sum_{s=1}^{N_{spk,k} < t < T_{DUR}} \delta\left(t - \tau_{k,s} - \Delta T\right) \cdot \Delta \vec{m}_k \qquad (1)$$

where $\delta\left(t - \tau_{k,s} - \Delta T\right)$ represents a single spike of cell k, fired at time $t = \tau_{k,s}$, $N_{spk,k} < t < T_{DUR}$ is the total number of spikes of cell

$k$ up to the current time $t$ (until saccade duration, $T_{DUR}$); $\Delta T$ is the lead time of SC output neurons ($\sim 20$ ms), and $N_{POP}$ the total number of active cells. It is important to note that Eq. 1 by itself does not necessarily generate the actual eye trajectory, $\Delta \vec{E}(t)$, which is likely to be modified by the downstream neural circuits in brainstem, cerebellum, and plant dynamics. Second, the spike timings are not prescribed by the model and have to be determined by actual recordings from many SC cells. Third, Eq. 1 does not predict that each individual neuron should encode the instantaneous eye-movement trajectory.

Yet, simulations of Eq. 1 with measured spike trains from a population of SC cells, under the assumption of an overall linear brainstem burst generator, demonstrated that the simple feedforward model of Fig. 1a fully accounted for the nonlinear main-sequence properties and velocity profiles of fast and (blink-perturbed) visually evoked saccades[33]. In other words, $\Delta \vec{E}_D(t) \sim \Delta \vec{E}(t)$. As a logical consequence, the main-sequence nonlinearity of eye saccades[26] must (mainly) reside in the distribution of spike trains (i.e., firing rates) within the motor SC[52] (orange in Fig. 1a), and not at the level of the horizontal/vertical-torsional brainstem burst generators[40,53].

In this way, the SC motor map would act as a nonlinear vectorial pulse generator, while the downstream neural control circuits and oculomotor plant dynamics conceptually implement a linear low-pass filter with delay on the SC motor command. Indeed, the Laplace transfer characteristic of the downstream system of Fig. 1a (green; adopting a simple linear first-order eye plant with time constant, $T_E$) is given by:

$$\frac{e(s)}{SC(s)} = \frac{B}{s \cdot \left(s + B \cdot \exp(-s\Delta T)\right)} \qquad (2)$$

where B (brainstem burst generator gain; order 50-80) and $\Delta T$ (the system's pure delay; order 12–20 ms) are the only two free parameters of the model.

In addition, Goossens and Van Opstal[33] found that the cumulative number of spikes in the bursts of single cells related linearly to the instantaneous displacement of the eye along a straight, intended, trajectory, $\Delta e(t)$, from initial to final eye position, for fast and for slow (blink-perturbed) saccades:

$$\sum_{s=1}^{N_{spk,k} < t < T_{DUR}} \delta\left(t - \tau_{k,s} - \Delta T\right) \propto \Delta e(t) \qquad (3)$$

This property suggested that all cells in the population have highly synchronized burst profiles[33,48,54], which was supported by subsequent cross-correlation analysis. Moreover, this result implied a tight correlation between the instantaneous vectorial velocity of the eye and the instantaneous firing rate of single neurons, even though this does not follow directly from the linear ensemble-coding scheme of Eq. 1.

Several studies have challenged the validity of linear summation of the SC output, like in Eq. 1. For example, Lee et al.[36] argued that linear vector summation of cell contributions[51] cannot explain why saccade endpoints deviate away from the site of a small reversible SC lesion, while the saccade into the center remained normometric. This result could be readily explained by (nonlinear) vector-averaging. However, this could not account for their additional finding that all saccades after the local lesion were substantially slower than control saccades. Goossens and Van Opstal[33] explained these micro-lesion results on both metrics and kinematics with one additional extension to Eq. 1: a downstream mechanism that stopped the brainstem burst generator as soon as the total number of spikes exceeds a

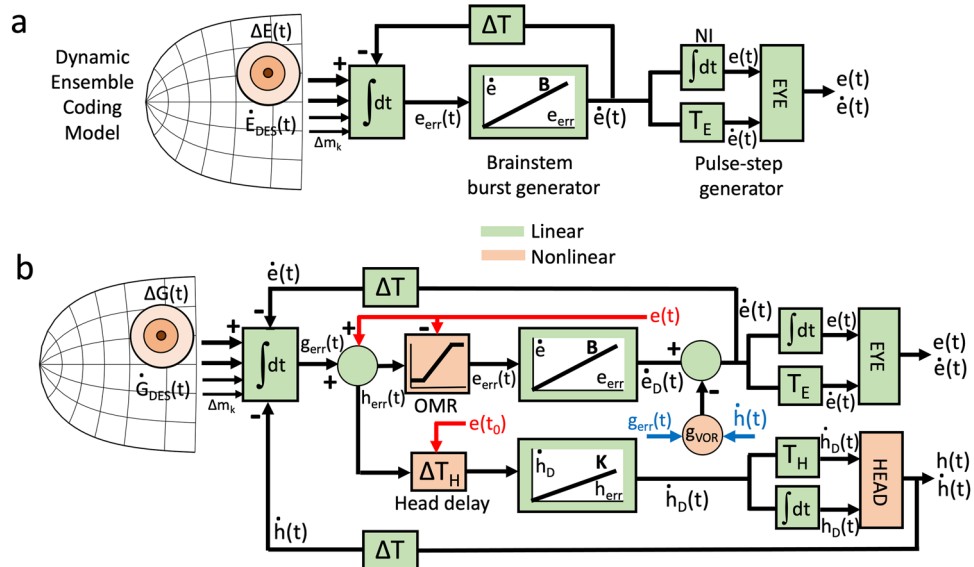

**Fig. 1 Models for saccade generation. a** Conceptual representation of the dynamic ensemble-coding model (Eqs. 1 and 2) for eye-only saccades (after[33]). For simplicity, details like an omnipause neuron timing gate, and separate horizontal/vertical burst generators have been omitted. Despite a linear brainstem burst generator, the model (free parameters: delay, $\Delta T$, and burst gain, B) reproduces the nonlinear main-sequence relations, straight oblique trajectories with horizontal/vertical component cross-coupling, and the velocity profiles of real saccades when driven by real neural recordings. **b** An extended ensemble-coding model for eye-head gaze shifts contains at least four unavoidable downstream nonlinearities (orange): (i) involvement of the VOR modifies the desired eye velocity command from the burst generator, $\dot{e}_D(t)$, with a gaze-error dependent gain, $g_{VOR}$; (ii) the eye-position dependent timing of the eye-head onset delay, $\Delta T_H$, which determines the contribution of the head to the gaze shift, (iii) a saturating oculomotor range (OMR), and (iv) a nonlinear head plant. $g_{err}$ = gaze error, $h_{err}$ = head error, $e_{err}$ = eye error; NI = neural integrator; $e(t)$ = eye position, $\dot{e}(t)$ = eye velocity, $e(t_0)$ = initial eye position; $\dot{h}_D(t)$ = desired head velocity; B, K: gains of eye- and head burst generators; $T_{E,H}$ = time constant of the eye and head plants.

fixed threshold (their Fig. 10[55]):

$$N_{POP}(t) = \sum_{k=1}^{N_{POP}} \sum_{s=1}^{N_{spk,k}} \delta\left(t - \tau_{k,s}\right) \leq \Theta_{FIXED} \qquad (4)$$

Other studies that refuted the model of Eq. 1 based their arguments on the property of Eq.3[49,50]. For example, Peel et al.[49] showed that under specific conditions (cooling of the Frontal Eye Fields, FEF) the number of spikes in single neurons for memory-guided saccades (but not for visual-evoked saccades) was substantially lower than for matched pre-cooling saccades. Without cooling, spike counts in SC cells were the same for all tested saccade types (memory, delayed, gap, and direct visual-evoked). The cooling results indicated that the spike-timing encoding in the SC (not prescribed by Eq. 1), is partly determined by inputs from the FEF, and the authors proposed that in the absence of this input other circuits (presumably cerebellar) could modify the effective signal strength from SC ($\Delta\vec{m}_k$ in Eq. 1). Zhang et al.[50] recently argued that the kinematics of saccades are dissociated from the instantaneous firing rate of single neurons, indicating that Eq. 3 may not apply to the population for all conditions. Yet, the variability in neural activity among the individual neurons in their study was much larger than the relatively small variation in behavioral kinematics (visual vs. memory-guided saccades), making the presence of any correlation very difficult to detect.

Finally, activity for saccades towards a moving target does not always incorporate the pursuit-induced eye displacement, which suggests that the readout of SC activity may also be task dependent[56].

A potentially strong argument against a crucial role for the SC in saccade control is that after complete bilateral SC lesions monkeys can still generate saccades[57], indicating that the SC has no central role in saccade control[58]. It should be noted, however, that the lesioned monkeys were tested after a considerable

recovery period, and that the saccades were slower and with longer reaction times. Moreover, deficits in saccade metrics, latencies, and kinematics are immediately observed after acute reversible microlesions in the SC[36,59,60]. Furthermore, Hepp et al.[61] provided evidence that the immediate effect on saccades after total bilateral SC inactivation is in fact quite dramatic, as monkeys were no longer able to generate any normal saccades in the light other than some slow, infrequent spontaneous eye movements.

Here we take the stance that under normal conditions (i.e., without lesions or microstimulation) the SC acts as the primary controller for saccades, but that after chronic or acute lesions, its role may be (partly) taken over by other neural regions such as the FEF[57]. Whether in those cases these regions exhibit, or learn to exhibit, similar encoding properties as observed in the SC motor map is not known.

Recordings and microstimulation studies from head-unrestrained cats (e.g[42,62–64]) indicated that SC activity is best understood as encoding combined eye-head gaze shifts. Note that gaze (the eye in space) is an 'abstract' signal, mathematically defined by the summed coordination of eyes and head. Extending the concepts from the simple (non-redundant, linear) oculomotor model of Eqs. 1–3 and Fig. 1a to combined eye-head gaze shifts is far from trivial.

Figure 1b presents a possible scheme for eye-head gaze control by the SC population (after[65]). Behavioral studies have shown that eyes and head are both driven by their own goals: an oculocentric gaze error for the eye vs. a craniocentric error for the head[14,66], which requires feedback from an eye-in-head position signal (red arrows in Fig. 1b).

Even if we assume, for the sake of argument, linear eye- and head burst generators, at least four unavoidable nonlinearities enter the system (orange in Fig. 1b). First, a gaze shift can extend far beyond the limited oculomotor range (OMR ~ 35 deg). A way

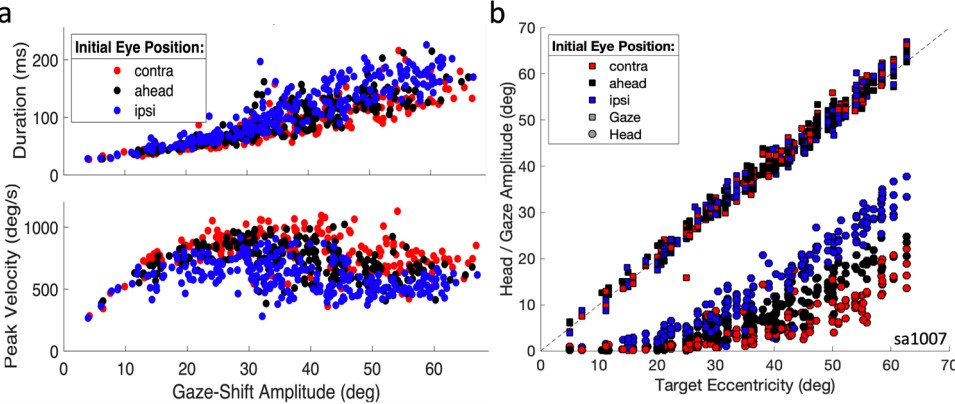

**Fig. 2 Gaze-shift kinematics: influence of initial eye position. a** Changes in initial eye position (colour coded) have a strong influence on the gaze-shift duration (top) and peak gaze velocity (bottom). Contralateral eye positions lead to the fastest gaze shifts (red), while ipsilateral positions induce much slower gaze shifts (blue). The reduction in peak velocity can be > 50%. **b** Changes in eye position and the evoked gaze-shift amplitude (squares) both induce considerable and systematic changes in the head-movement contribution to the gaze shift (circles). Session Sa1007.

to avoid an excessive drive for the eyes is to recalculate the gaze-centered goal into a head-centered goal, soft-limited by an OMR saturation. Second, gaze-shift kinematics heavily depend on the head contribution to the saccade, which varies with initial eye position. Because a gaze shift can be generated by many different head-contributions, the relationship between goal and motor output is nonlinear. Third, during gaze shifts, the VOR gain is modified (blue errors in Fig. 1b[67]); introducing yet another nonlinearity: the oculomotor burst, $\dot{e}_D(t)$, no longer specifies actual eye-velocity, $\dot{e}(t)$, but a signal that varies from eye-in-head velocity ($g_{VOR} = 0$) to gaze velocity ($g_{VOR} = 1$). Fourth, the head biomechanics are severely nonlinear for several reasons: its rotation axes do not intersect in a fixed point[68], and further nonlinearities arise from the head's inertia together with the complexities of force development (force-velocity relationships). Finally, it is not guaranteed that the SC spike timings for eye-head gaze shifts remain identical to those of ocular saccades.

Because of this, it is not readily expected that the simple relation of Eq. 3 for ocular saccades can also capture the complexity of combined eye-head gaze shifts.

**This study**. Here, we characterize the neural activity of SC neurons during eye-head gaze shifts. We performed single-unit recordings from the SC in two head-unrestrained monkeys. Animals were trained to generate accurate eye-head saccades with considerable natural variability in their kinematics, induced by deliberately varying the initial eye-in-head position. We investigated whether Eq. 3 would also hold for head-unrestrained saccades, and whether the nonlinear and variable gaze kinematics and gaze-velocity profiles are reflected in the burst characteristics of SC neurons.

## Results

**Kinematics**. An important goal of this study was to analyze SC neural activity for gaze shifts with a large natural variability in their gaze kinematics, which was achieved with the initial eye-position paradigm (see Methods). Figure 2a shows a representative main-sequence plot (amplitude-duration and amplitude-peak velocity relations) for a large range of gaze-shift amplitudes (5-70 deg) during experiment Sa1007. Initial eye position ($E_0 \in [-15, 0, +15]$ deg) is color-coded (see Supplemental Fig. 1, for an example).

Note that in the experiments, the initial head position always followed the opposite convention: $H_0 \in [+15, 0, -15]$ deg, respectively. That is, when the eye was deviated toward the side

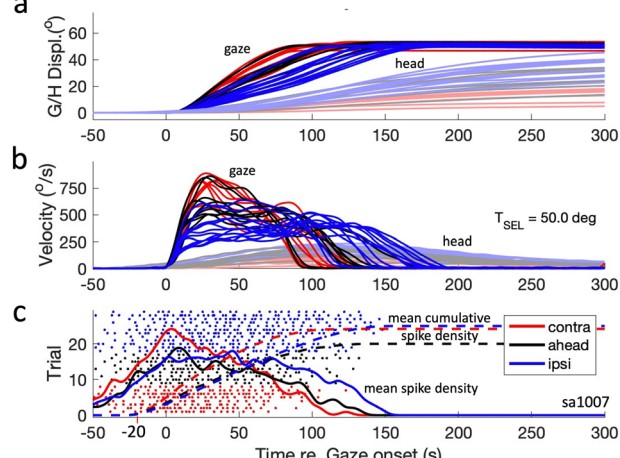

**Fig. 3 Gaze-shift kinematics and related neural activity. a**, **b** Gaze shifts into the cell's movement field with the same amplitude (a range of directions) systematically vary in their head-movement contributions and have very different gaze-velocity profiles. The two panels show the vectorial position and velocity profiles for gaze and head. Gaze- and head-position traces were all aligned at zero starting position, for clarity. **c** Individual spikes for the 29 selected trials and associated mean spike-density functions (color coded solid lines for the three initial conditions) and mean raw cumulative spike counts (dashed lines, starting 20 ms before gaze onset). Note the strong dependence of the spiking patterns on initial eye position, which is evident from the differences in burst duration and average spike-density profiles. Session Sa1007.

ipsilateral to the direction of the imminent gaze shift, the head pointed in the contralateral direction by the same amount, keeping the initial gaze position at $G_0 = 0°$ (see Methods).

Three effects can be observed. First, for ipsilateral eye deviations (at +15°, blue), gaze durations were prolonged, and peak-velocities lowered, when compared to contralateral fixation (at −15°, red; for the approximate size of these effects, see below, Fig. 3b), with the aligned condition between the two (black). Second, there is considerable variability in the gaze kinematics for a given amplitude (also partly due to differences in gaze-shift direction), and third, for amplitudes ≳35 deg, the gaze peak velocity starts to *decrease* for all three conditions, caused by the increasing head-movement contribution (Fig. 2b).

Figure 3 shows a selection of gaze shifts recorded during this experiment and initiated from the three initial eye positions into

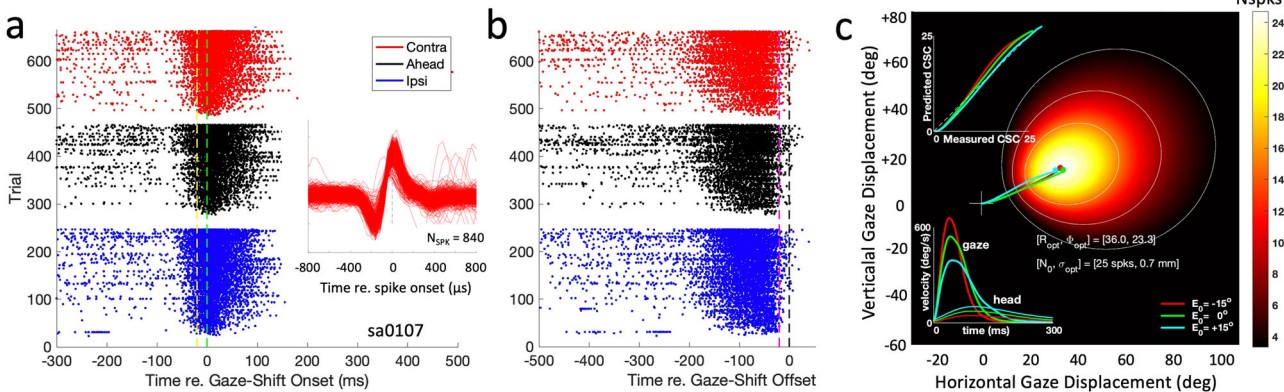

**Fig. 4 Example recording of cell Sa0107. a** Raw spike-recording data of cell Sa0107 aligned with gaze-shift onset (green dashed line). The 664 successful trials were re-ordered according to the number of spikes in the burst from 20 ms before gaze-shift onset (yellow-dashed line) to 20 ms before gaze-shift offset, and initial eye-position (color-coded). Inset: 840 spike waveforms from the 20 trials that yielded the largest number of spikes in the gaze-related burst. **b** Same data arranged with respect to gaze-shift offset (black dashed line). Note the marked offset of the burst, approximately 20 ms before the gaze shift ends (magenta dashed line). **c** Movement field of the cell. Optimal fit parameters (Eq. 9) are indicated. Number of spikes in the burst is encoded by hot color from 0 (black) to 25 (white). The thin white contours indicate gaze vectors with a fixed number of spikes. Three averaged gaze-shift trajectories into the center of the movement field are highlighted (red: ipsi, green: ahead; blue: contra). Insets: Lower-left: Averaged gaze- and head velocity profiles for the optimal gaze shifts. Upper-left: Mean cumulative spike count (CSC), predicted (Eqs. 10-13) vs. measured.

the cell's movement field, with amplitudes within 1% of 50 deg ($\Delta G \in [49.5–50.5°]$), and various directions, $\Delta \Phi \in [-35, +5]$ deg (see also Supplemental Fig. 2). The bottom panel shows the recorded spikes for the three initial conditions, indicating that the burst duration for these fixed-amplitude gaze shifts varied considerably and systematically with the different initial eye positions: shorter bursts for the contra condition, longest bursts for the ipsi condition.

This is also evident from the mean instantaneous spike-density profiles (solid curves) for these trials, which indicates that the peak activity for the contra-condition is higher than for the ahead- and ipsi-conditions. Note the considerable trial-to-trial variability of the interspike intervals for the three gaze-shift clusters.

Note also that the peak spike density for the fastest responses (contra, red curve) coincides closely with the gaze-shift onset and thus leads the peak gaze velocity by approximately 20 ms. The spike-density profile for the contra-condition (blue) is much flatter, so that the timing of its peak is less well defined. The same holds, however, for the gaze velocity profile. The total mean cumulative spike counts (dashed) appear to be little affected by the large variations in gaze-shift kinematics and firing rates. Below, we will analyze these spiking patterns in more quantitative detail on a trial-by-trial basis.

Supplemental Fig. 2 illustrates how the head-movement onset systematically shifts with initial eye position (early onset for ipsi, late for contra) to modulate the head contribution to the gaze shift, as shown in Fig. 2b.

**Raw data**. Figure 4 shows the raw recorded data obtained from cell Sa0107. In Fig. 4a, all action potentials are referenced with respect to the gaze-shift onset (at $t = 0$; green-dashed line) and re-ordered according to the number of spikes in the burst (in descending order) and initial eye position (color). The inset illustrates the stable spike waveform for the 20 most-active trials (which were randomly distributed across the experiment). Note that the cell prepared its neural response with a prelude that started between 30 and 100 ms before gaze-shift onset: the stronger the eventual burst, the earlier the start of this prelude. In Fig. 4b the spikes are aligned to gaze-shift offset (black dashed line). Note that for the majority of trials, the bursts ended

approximately 20 ms before gaze-shift offset (magenta dashed line). This property has also been reported by Freedman & Sparks[64], but is not observed for all cells[33,44,45]. Often, some neural activity persisted after gaze-shift offset. This is also observable for this neuron in the trials with the strongest responses. As motivated in the Methods, in our movement-field and spiking-pattern analyses we only included spikes that occurred within a time window that equaled the gaze-shift duration and started at 20 ms before gaze-shift onset (yellow dashed line) and ended at 20 ms before its offset (magenta dashed line).

Figure 4c presents the results of fitting the cell's movement field with the static (Eq. 9) and dynamic (Eq. 10–13) models. Data from averaged optimal gaze shifts towards the center of the movement field are shown as well as the trajectories from the three different initial eye positions, the gaze- and head velocity profiles (inset lower-left), and the model analysis of the dynamic cumulative spike counts (CSC) for the averaged gaze shifts (upper-left: measured cumulative spike count vs. predicted by Eq. 13). Note that in the latter plot the CSC trajectories nearly fully overlap along the $y = x$ diagonal, despite the considerable variation in gaze-position sensitivity for this neuron was $\varepsilon = 0.003$ spks/deg, the predicted total number of spikes varied by $\Delta N = \varepsilon \times \Delta E_0 \times N_{opt} = 0.003 \times 30 \times 30 = 3$ spikes ($\sim 10\%$ of $N_{opt}$), which can indeed be seen in the displaced endpoints of the CSC trajectories. In Supplemental Fig. 3, we show three other example cells in this format.

**Movement field**. To quantify the movement-field properties of SC neurons, Fig. 5 shows the results of the fit for cell Sa0107 for all 664 saccades (same cell as in Fig. 4). Figure 5a compares the predicted, $N_{STMF}$, and measured number of spikes, $N_{spk}$ ($r = 0.94$; fit parameters provided in the inset) of the static movement-field fit (Eq. 9). The cell had its movement-field center at $(R_c, \Phi_c) = (36.0, 23.3)$ deg (Fig. 4c), a maximum of $N_0 = 25$ spikes, and an eye-position sensitivity of $\varepsilon = 0.003$ spikes/deg. The raw phase-trajectories between the straight-line gaze displacements and smoothed cumulative spike counts (according to Eq. 3) are shown in Fig. 5b for all 664 gaze shifts. Endpoints of the trajectories correspond to the measured total number of spikes (abscissa of

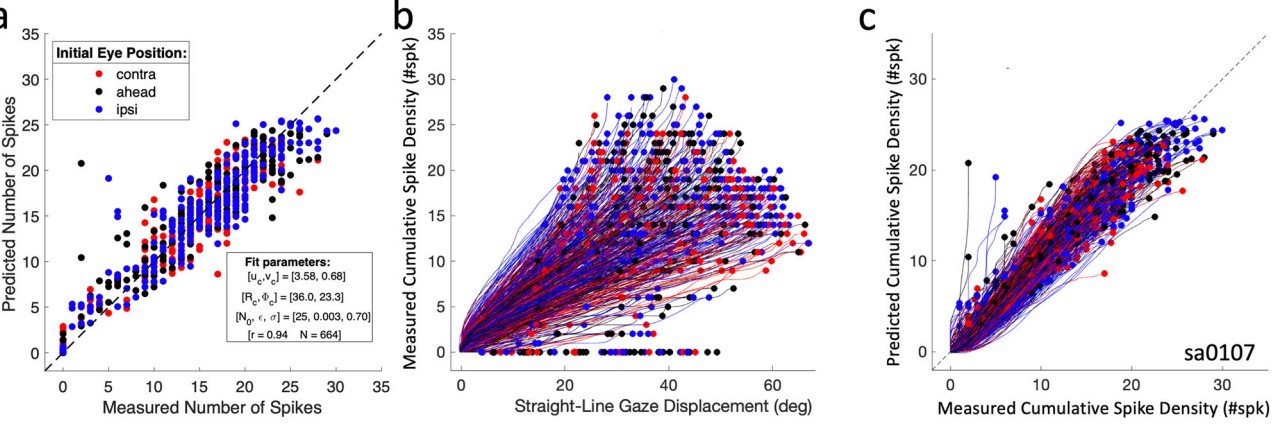

**Fig. 5 Movement-field properties of cell Sa0107. a** Static movement field of Eq. 9 comparing predicted $N_{MFST}$ vs. measured $N_{spk}$. **b** Dynamic movement field, showing the measured cumulative number of spikes in the burst (delayed by 20 ms) as function of the instantaneous gaze displacement along the straight line between the gaze on- and offset positions for all 664 gaze shifts of **a** into the movement field (see also Supplemental Fig. 3). **c** Predicted instantaneous cumulative spike density (Eqs. 10, 11) vs. measured cumulative spike density shown in **b**. Number of data points in **b**, **c**: N = 68,814 (1 ms sampling).

panel 5a), and the colors encode the initial eye-position. Note that individual phase trajectories are close to linear (i.e., in line with Eq. 3) and occupy a large part of the phase space, because of the large range of saccade vectors and associated spike counts. For example, for a fixed number of, say, 15 spikes (ordinate), saccade vectors can have amplitudes (abscissa) that range from about 18 to 62 deg (all saccades that lie on the ellipsoidal iso-spike count contour of 15 spikes around the center of the movement field; see, e.g., Fig. 4c). To test whether the dynamic movement-field description of Eq. 10–13 provided an adequate model for these fanning phase trajectories, Fig. 5c confirms the excellent agreement between the predicted and measured cumulative spike counts (Eq. 10–13; $r^2 = 0.90$).

**Spiking variability**. Note that the static and dynamic movement-field models in Fig. 5a, c predict the final and instantaneous number of spikes for every saccade in the movement field quite well on average (the mean signed deviation of the model is zero), but the result in Fig. 5a also suggests that the trial-to-trial variability in the number of spikes can be substantial. For example, for a predicted number of 20 spikes (ordinate), the measured number ranged from 16 to 24 spikes (abscissa), i.e., by ±20%. To check whether the mismatch between predicted and measured number of spikes depended systematically on the mean neural activity (and could thus potentially be described as signal-dependent, or multiplicative, noise), we investigated the relationship between the mean measured number of spikes in the gaze-related burst and the standard deviation of this mismatch.

To that end, we re-ordered all trials in descending order according to the predicted number of spikes of the movement field (Eq. 9) and subsequently selected groups of trials in equal-sized bins of 2 spikes, shifted in one-spike steps across the data base. Note that a predicted fixed number of spikes is calculated for all saccade vectors that lie on an ellipsoidal iso-activity contour (see, e.g., the thin white lines in Fig. 4c, or horizontal cross sections in Fig. 5b). Two examples of selected saccade vectors for $N_{spk} = 10$ and 30 spikes are shown in the inset of Fig. 6b as magenta dots. Therefore, variations in saccade amplitude, direction, and initial eye position are all accounted for by this analysis.

The result is shown in Fig. 6 for neuron Sa2606. The data in Fig. 6a (from 752 trials) show a clear increase in the variability of the burst-related spike count as function of the predicted total number of spikes in the burst (note that the predicted spike count

corresponds well to the mean measured spike count for each gaze shift, as $r^2 = 0.90$ for the static movement field fit of Eq. 9; see Supplemental Table S1). Figure 6b shows a near-linear relationship between the mean and the standard deviation ($r = 0.93$), with an offset close to zero (a = 0.65 spikes), and a (dimensionless) slope of b = 0.156. This result indeed indicates the presence of multiplicative, signal-dependent, noise in the cell's activity[4,9], regardless the saccade vector and the initial conditions, and only a small contribution of constant, additive noise. For cell Sa0107 (Figs. 4 and 5) we obtained: offset=0.28 spikes, slope=0.141, and r = 0.91, respectively. To include the two types of noise in the predicted spike count for the saccade-related burst, Eq. 9 would have to be modified as follows[9]:

$$N_{SPKS} = (1 + n_{MUL}) \cdot N_{STMF}(\Delta G, \Phi, E_0) + n_{ADD}$$
$$\text{with } n_{MUL} \in N[0, C_V] \text{ and } n_{ADD} \in N[0, \sigma_{ADD}]$$

with $N(x,y)$ indicating a Gaussian distribution with mean $x$ and standard deviation $y$. The inset of Fig. 6b illustrates two example distributions (on a common compressed scale for visual purposes) of the saccade vectors that were associated with a fixed number of spikes (N = 10 ± 1 and 30 ± 1 spikes, respectively). It can be seen that the distributions follow parts of the ellipsoid around the estimated movement-field center (cross), which for this cell was very eccentric ([$\Delta G, \Delta \Phi$] ≈[87,21] deg). The blue and pink colored symbols reflect two different analyses: dark/light blue corresponds to the selected vectors for which the measured number of spikes was constant, whereas the pink/magenta vectors correspond to the predicted fixed number of spikes. The distributions for measured and predicted vectors were very similar, again indicating that the static movement-field model of Eq. 9 captures the data quite well. Note also that the variances of the vector distributions were comparable: for N = 30 spikes the amplitudes were on average 59 deg, with σ = 7°; for the N = 10 spikes vectors, the mean amplitudes were 41° with σ = 6°.

Below, we will summarize the results of this regression analysis for the neural population. The regression results for the 20 best-recorded neurons are shown in Supplemental Fig. 4.

**Population results**. To quantify the static movement-fields (Eq. 9) for the population of 43 neurons, Fig. 7a shows a histogram of the quality of the fits (Fig. 5a). For most cells, $r \geq 0.85$, further supporting the appropriateness of the model.

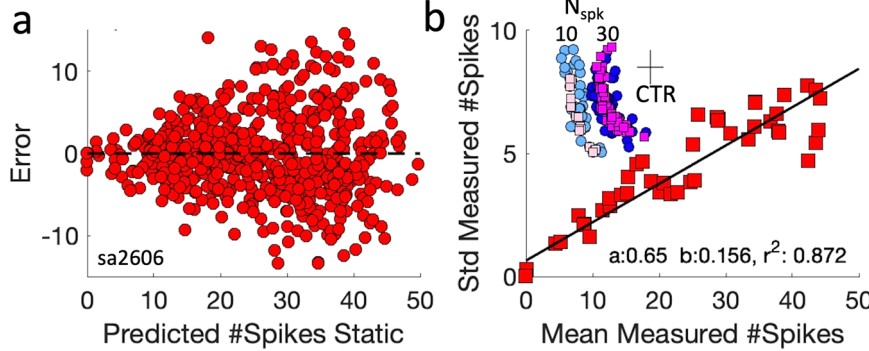

**Fig. 6 Multiplicative noise in neural response of cell Sa2606. a** The error in the predicted number of spikes in the burst (Eq. 9) scatters around zero (i.e., it's the optimal fit), but variability increases with the mean number of predicted spikes, here shown for cell sa2606 (752 trials). **b** Relationship between the mean number of spikes and its standard deviation for all saccades into the movement field ($N = 752$; pooled for initial eye position). The STD increases nearly linearly with the mean, indicating the presence of multiplicative, or signal-dependent, noise. The black line shows the best linear fit through the bins, yielding an offset near zero ($a = 0.65$ spikes), a slope (coefficient of variation) of $C_V = 0.156$, and a correlation, $r = 0.93$. The inset shows two distributions of saccade-vector endpoints (on a compressed scale) for which $N_{spk}$ was either 10 or 30 ($\pm 1$) spikes. CTR: movement-field center at $[\Delta G, \Delta \Phi] = 21,87$ deg; the vectors have amplitudes of about 40 deg (10 spks) and 60 deg (30 spks). Blue: saccade vectors for a fixed measured number of spikes. Pink: vectors for a fixed predicted spike count.

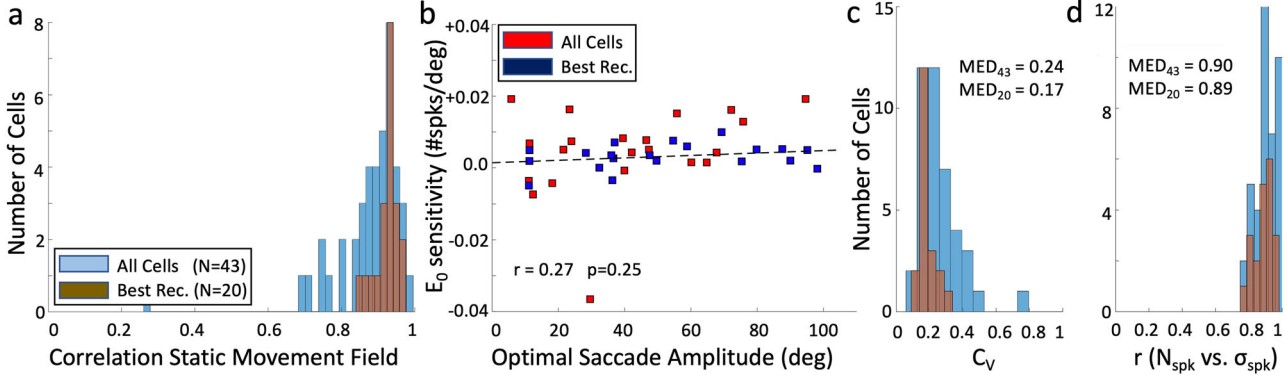

**Fig. 7 Static movement fields: population. a** Histogram of the fit quality (Pearson's r) of Eq. 9 for all 43 static movement fields (blue) and for the 20 best-recorded neurons (brown). **b** At the population level, eye-position sensitivity, ε, of the movement fields does not depend on a cell's optimal gaze-shift amplitude ($p = 0.25$; mean: 0.003; CI: $[-0.004:0.008]$ spikes/deg). **c** Distribution of coefficients of variation, $C_V$. Median values are indicated. **d** Correlation coefficients between mean number of spikes in the burst and standard deviation (e.g., Fig. 6b).

Supplemental Table S1 provides the fit parameters for all cells. Figure 7b plots the fitted eye-position sensitivity, ε (in #spikes/deg) as function of the optimal gaze-shift amplitude. As there is no significant correlation ($p = 0.25$), and the mean is not different from zero, on average the total number of spikes in the burst of SC cells remained invariant to changes in initial eye position, and to the variability in gaze-shift kinematics (e.g., Fig. 3b). Yet, several cells had a small, but significant, eye-position sensitivity. For example, with ε = 0.005 spikes/deg, $N_0 = 30$ spikes, not incorporating the noise, and moderate eye-in-head positions of $E_0 = \pm 15°$, the number of spikes for the optimal saccade is expected to vary between $N_0 \cdot (1 \pm \varepsilon \cdot E_0) = 28$ and 32 spikes, respectively ($\Delta N = 4$-5 spikes, i.e., by ~15% of its peak value).

To summarize the analysis of the variability around the mean number of spikes of SC neurons, Fig. 7c, d provides the results of fitting a linear relationship between the mean measured number of spikes and the standard deviation (as in Fig. 6b) for all 43 cells.

Figure 7c provides histograms for the slopes of the relationship and Fig. 7d the distribution of correlation coefficients between the average number of spikes and standard deviation. We conclude that a linear relation provides a good description for the variability in the number of spikes during the gaze movement-related burst (64–90% of variance explained), indicating the

presence of multiplicative noise in the firing behavior of the entire population of SC cells.

The median slope value for the population, (0.24), is comparable to the mean slope = $0.28 \pm 0.16$ reported by Goossens and Van Opstal[54] for head-restrained eye saccades from >100 neurons across the SC motor map.

The offsets of the linear fits quantify whether the cells were also endowed with additive noise (constant, signal-independent). The results were as follows: for all 43 cells, $a = 0.26 \pm 0.27$ spikes; for the 20 best recordings: $a = 0.41 \pm 0.31$ spikes. As both offsets were very close to zero, the amount of additive noise within the population is negligible when compared to the effect of multiplicative noise to spike-count variability: for a typical maximum of $N_0 = 25$ spikes for the optimal saccade, the standard deviation is $0.24 \times 25 = \pm 6$ spikes (i.e., spike counts could range from about 13-37 spikes; see Fig. 6a).

**Dynamic movement fields**. We quantified the goodness-of-fit of the dynamic movement field (Eqs. 10–13) by fitting a linear regression line through each measured phase trajectory of each gaze shift (as in Fig. 5b) for which the measured number of spikes ≥10 and compared the measured slopes with the predicted slopes

of the dynamic movement-field model, which is given by $\alpha = N_{STMF}/\Delta G$. The result is shown in Supplemental Fig. 5 for cell Sa0508. For this cell, the relationship for the 273 slopes was described well by the solid black linear regression line, with offset $b = 0.01 \pm 0.03$ and slope $a = 1.15 \pm 0.03$ ($r^2 = 0.88$), indicating an overestimation of the slope of the phase-relation for this cell by 15% on average.

We next determined the instantaneous deviation, $\hat{d}(t)$, of each phase trajectory (Fig. 5b) with respect to the best-fit straight line, $\hat{n}_{CS}$, for the measured cumulative spike density as a function of the instantaneous straight-line gaze shift, $\Delta g(t)$, between 0 and $\Delta G$ (gaze-shift direction $\Phi$), for each saccade into the movement field for which the number of spikes was $N_{spks} \geq 10$:

$$\text{Straight} - \text{line gaze shift} : \Delta g(t) = g_H(t) \cdot \cos \Phi + g_V(t) \cdot \sin \Phi$$
$$\text{Fit} : \hat{n}_{CS}(t - \Delta T) = a \cdot \Delta g(t) + b$$
$$\text{Deviation} : \hat{d}(t) \equiv n_{CS}(t - \Delta T) - \hat{n}_{CS}(t - \Delta T)$$

(5)

Supplemental Fig. 5b shows the histogram of the instantaneous deviations (in #spikes) for the same cell as in Supplemental Fig. 5a, which has a standard deviation, $\sigma = 0.68$ spikes. This indicates that straight lines were adequate descriptors for the measured dynamic phase trajectories (Fig. 5b).

Finally, Supplemental Fig. 5c provides the standard deviations for all 43 cells (20 best-recordings in blue). These results were quite comparable across the population at large. In Supplemental Fig. 6 we show the statistics for the results of the differences between predicted and measured slopes for neuron Sa1007, and for the total recorded population.

**Peak spike density vs. peak gaze velocity.** So far, we have analysed the SC movement fields on the basis of the (cumulative) number of spikes in the burst. To relate the instantaneous firing rate of the neurons to the instantaneous gaze kinematics requires a continuous estimate for the instantaneous firing-rate profiles over the same time interval as the gaze shift. To that end, we convolved all discrete spike events from Gaze Onset –20 ms to Gaze Offset –20 ms with a Gaussian kernel (fixed 4 ms width;[43,54,69] see Methods, Eq. 15). We verified that the spike density function was not significantly altered when including spike contributions from the prelude and post-saccadic activity (see Methods, and illustrated in Supplemental Fig. 7).

According to Eq. 13, a neuron's (peak) activity in a trial may be expected to co-vary with the (peak) gaze track-velocity. Figure 8a, b illustrates this point for cell Sa1007. To obtain this plot, we selected clusters of near-identical gaze-shift vectors with their end points falling within a $2 \times 2°$ bin near the cell's movement-field center. For this neuron, we found seven clusters with four or five gaze shifts ending within 10° of the center. Figure 8a shows the gaze trajectories of these clusters in different colors. Figure 8b shows the peak spike density (squares) during each gaze shift and within each cluster against peak gaze track-velocity, as well as the number of spikes (circles). Note the considerable variation of gaze-shift velocity, which is mostly due to the variation in initial eye position (e.g., Figs. 2 and 3). For these gaze shifts, the peak spike density indeed covaried with peak gaze velocity, while the number of spikes remained nearly constant.

We verified that the peak spike density indeed preceded the peak gaze velocity, as is illustrated in the inset. Although determination of the peak location in the spike density function (and also in many ipsilateral gaze-velocity profiles) is noisy and not always unambiguous (see, e.g., Fig. 3), the average lead of the peak SD for the 29 selected gaze shifts of this cell was $-10.5$ ms.

For eight cells we could retrieve enough near-identical gaze shifts, with at the same time sufficient variation of peak velocities within each cluster. Linear univariate regression on the pooled data ($N = 538$ responses; Fig. 8c, d) indicated that the sensitivity of peak gaze velocity on peak spike density (slope: $a = 0.44$ (spikes/s)/(deg/s) CI:[0.37, 0.55]) was substantially higher than for $N_{spks}$ (slope: $a = 0.011$ spikes/(deg/s) CI:[0.007, 0.015]). Over the range of peak velocities of 850 deg/s, the average peak spike density nearly tripled from 210 to 610 spikes/s. In contrast, the number of spikes increased by 50% (from 20 to 30 spikes) over this range.

A significant correlation between peak gaze velocity and number of spikes could be due to the fact that for some neurons the number of spikes changed with the initial eye position (it increases because of a positive eye-position sensitivity), while the gaze velocity decreases for ipsilateral eye positions. In this way, eye position would be the common underlying cause for a correlation.

To dissociate these factors, we performed a multiple linear regression on the dimensionless z-scores of the three variables, to determine whether peak gaze velocity was better predicted by the peak spike density, by the number of spikes, or by both:

$$\hat{V}_{PK} = p_{SD} \cdot \hat{F}_{PK} + p_{NS} \cdot \hat{N}_{SPK} \text{ with } \hat{x} \equiv (x - \mu_x)/\sigma_x$$ (6)

The partial correlations, $p_{SD} = 0.82$ (CI:[0.72, 0.93]; $\alpha = 0.01$) and $p_{NS} = -0.39$ (CI:[$-0.49$, $-0.28$]; see also Supplemental Fig. 8), confirmed that a cell's peak spike density was the major predictor for the peak gaze velocity, and that the number of spikes and peak spike density have opposite effects.

Supplemental Figure 9 provides the results of the peak-timing analysis (same format as the inset of Fig. 8b) for all 538 selected responses from these 8 cells. Despite the large variability in the data (due to the fact that the spike-density profiles and gaze-velocity profiles were often multi-peaked, or broad, and therefore noisy; e.g., Fig. 3b) the peak of the spike-density led the peak gaze velocity in 66% of the trials. We obtained an average difference of $-12.4$ ms for these responses.

**Instantaneous spike density vs. gaze velocity.** The observed linear relation between the instantaneous cumulative spike density (or, equivalently, spike count) and the ongoing straight-line gaze displacement (e.g., Fig. 5b and Supplemental Figs. 5 and 6) predicts that the full profiles of their derivatives (i.e., the spike density, $SD(t)$, and the projection of gaze-velocity along the straight line, $\dot{G}(t\text{-}\Delta T)$) will be linearly related as well (Eq. 13). Although the derivative is a more sensitive measure to reveal a putative relation between neural activity and gaze-shift kinematics, it is also noisier, as the derivative amplifies the higher frequencies present in the original signal.

We investigated the dynamic relationship between the instantaneous (delayed) spike-density (fixed 4 ms kernels, see Methods) and gaze track-velocity on a trial-by-trial basis. Figure 9a exemplifies the result for the trial (#670) that yielded the largest number of spikes during the gaze shift ($N = 30$) in the experiment (cell Sa0107). The central panel shows the smoothed cumulative spike density ($CSD(t)$, black line) and the straight-line gaze-displacement signal, $\Delta g(t)$ (red). As the cumulative spike count necessarily increases monotonically with time, the two traces were highly correlated when $CSD(t)$ was delayed by $\Delta T = 20$ ms ($r = 0.993$). Note that this was also the case for the original discrete spike-count vs. gaze displacement ($r = 0.992$); the spike density transform did not specifically affect this relationship (see also below). The bottom panel shows that the delayed spike density, SD(t-ΔT), of the neuron and the gaze-velocity profile, $\dot{g}(t)$, had a reasonable, albeit lower, correlation

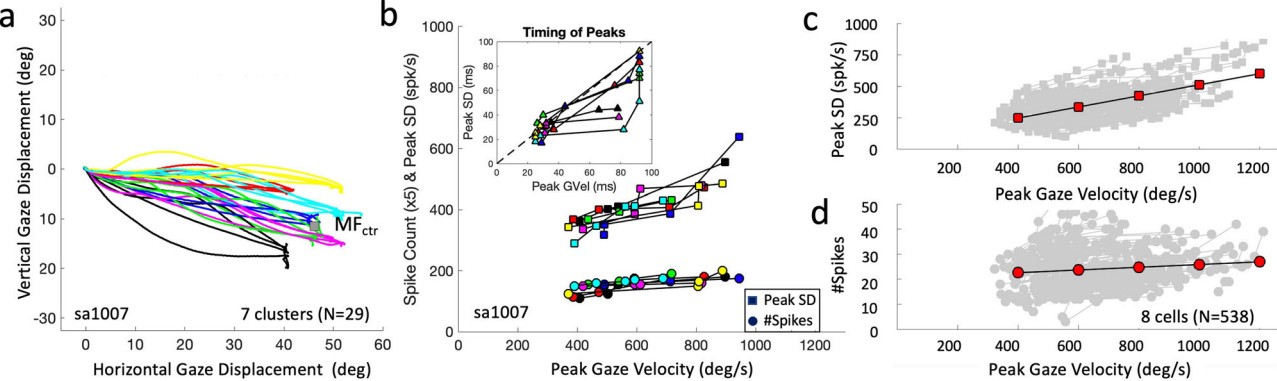

**Fig. 8 Peak spike density vs. peak velocity. a** Gaze shifts ($N = 29$) that ended close to the movement-field center, MF$_{ctr}$, of Sa1007. We retrieved 7 clusters of near-identical gaze displacements (color coded), each with 4–5 responses, but following different trajectories. **b** Peak spike density as function of peak gaze velocity (squares) shows a positive correlation for each cluster. The number of spikes in the bursts (circles) hardly changed, even though peak gaze velocity varied over a range between 350 and 950 deg/s. Inset: time of peak SD vs. time of peak gaze velocity in ms. The majority of points lies below the identity line (mean: −10.5 ms; std: 15 ms). **c** For eight cells, we could collect enough near-identical gaze shifts near the center of their movement field and with considerable variability in their kinematics to do a meaningful regression (538 responses; each cluster with at least 3 responses). Linear regression on the data set indicated that, on average, the peak spike density increased with peak gaze velocity with a gain of + 0.44 (spikes/s)/(deg/s). **d** The velocity gain for the number of spikes was much smaller, at + 0.011 spikes/(deg/s). Red symbols: prediction from the univariate linear regressions, shown at 200 deg/s intervals.

($r = 0.587$) than the associated cumulative signals of the center panel.

Panel 9b shows the 25 trials with the largest number of saccade-related spikes for this neuron, in which the instantaneous gaze track-velocities (red) and spike densities (black) were scaled to their peaks for illustrative purposes. In each subplot, the time axis was set equal to the associated gaze-shift duration, and the panels were ordered according to the total number of spikes in the burst, $N_{spk}$. Note that for many of these trials, the two profiles were similar. Indeed, for 16/25 responses in this panel the correlation was >0.80. This result was typical for the entire experimental session, and it did not critically change when including the effect of the prelude spikes on the spike-density function (Supplemental Fig. 7). The 664 correlations between the two profiles for all trials are summarized by the blue histogram of Fig. 9c. The brown histogram contains only those trials with a relatively large number of spikes, which ensures a more reliable spike-density profile. As an arbitrary criterion, we took $N_{spk} \geq 15$ ($N = 350$ trials). Note that despite some lower correlations in a minority of trials (like the one illustrated in panel 9a), the mode of both distributions was close to $r = 0.9$ (i.e., >80% of the variance explained).

However, a high correlation could in principle arise when gaze-velocity profiles and neuronal SC bursts can both be described by roughly similar single-peaked functions. Especially when the inter-trial variability between velocity profiles in the experiment and between SC bursts across trials would be limited (i.e., stereotypical), a high correlation in the recording session could be simply due to such a mere coincidental similarity, which would have no potential functional consequences whatsoever. To test for this possibility, we randomly shuffled the velocity profiles of the entire experimental session and recalculated the trial-by-trial correlations with the non-shuffled spike density over the time interval set by the selected gaze-shift duration (see Methods). The yellow histogram provides the resulting correlations for the shuffled profiles, showing that the peak at $r = 0.9$ indeed disappeared, and that therefore the high trial-by-trial correlations of the original data were real.

**Population.** In Fig. 9d we show the correlations between the instantaneous spike density vs. gaze track-velocity for all trials of

the 20 best-recorded cells (see Methods; dark-grey histogram; 3981 trials with $N_{spk} \geq 15$, mode at $r = 0.87$; see also Supplemental Fig. 10 for all 43 cells, giving the same result), for the shuffled trials of all 43 cells (blue), and for the 20 best-recorded cells (pink). The red-dashed curve represents the best-fit beta-function through the latter data (3430 responses), according to:

$$P(r) = P_0 \cdot (r + 0.5)^\alpha \cdot (1.0 - r)^\beta, \qquad (7)$$

with $P(r)$ the probability of correlation coefficient $r$, $P_0$ a normalization constant, and α, β dimensionless exponents describing the shape of the distribution. This yielded $P_0 = 131.1$, $\alpha = 1.71$, and $\beta = 0.70$, with the mode of this function at $r = 0.57$.

A one-tailed 2D Kolmogorov-Smirnov test demonstrated that the normal-data distribution is indeed shifted to higher values than the shuffled data set (H = 0 at α < 0.005; KS = $2.2 \times 10^{-4}$; $p = 0.9998$).

Supplemental Table S2 summarizes for the 20 best-recorded cells the number of trials in which the correlation between instantaneous delayed spike density and gaze track velocity exceeded $r = 0.71$ (i.e., at least 50% of the variation explained). The table shows that for 15/20 cells the percentage of such high-correlation trials was 39%, or higher. Only for 4 cells it was lower than 10%. Overall, the total number of high-correlation trials for the 20 cells was $1901/3981 = 48\%$. For neuron Sa0107 this occurred in 287/350 trials (82%). This particular neuron thus contributed $287/1901 = 15\%$ to the high-end correlations in the histogram of Fig. 10d (for $N_{spk} \geq 15$).

**Influence of filtering.** Supplemental Figs. 11–16 compare the results of the 4 ms kernel-based spike-density analyses to those obtained with the adaptive kernel method and the raw spike signal (see Methods, for details). Supplemental Figs. 11, and 12 show that the discrete cumulative number of spikes (a discontinuous, jagged signal) and the associated (smooth) spike-density transforms all correlated well with the instantaneous displacement of gaze along the straight line for the same 25 trials as shown in Fig. 9b (Eq. 10).

Supplemental Fig. 13 shows the same data, together with the discrete instantaneous firing rate (blue; calculated from the inter-spike intervals: $\dot{n}(i) = 1000 \cdot \left(\tau_{i+1} - \tau_i\right)^{-1}$), and the adaptive spike-density profiles (the derivatives of the black and green

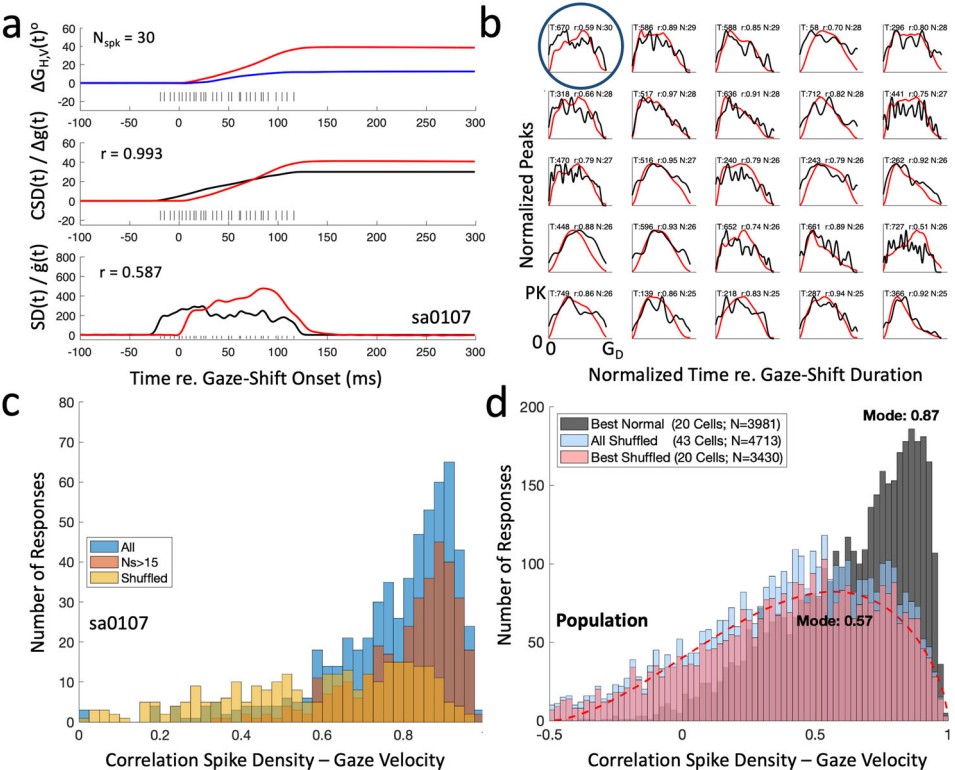

**Fig. 9 Instantaneous spike density vs. gaze velocity. a** A single trial during the recording of cell Sa0107. Top trace: horizontal (red) and vertical (blue) gaze displacement. Black ticks correspond to the individual spikes from 20 ms before gaze-saccade onset to 20 ms before offset ($N_{spk} = 30$). Center: cumulative spike-density, CSD(t), (black) and instantaneous straight-line gaze displacement (Eq. 11). The correlation between these two curves (spike density delayed by 20 ms) is $r = 0.993$. Bottom: instantaneous spike density, SD(t), (black) and gaze velocity along the straight line (red). Their correlation, $r = 0.587$, is modest. **b** Normalized gaze velocity between gaze-onset and -offset (red) and normalized instantaneous spike density on the same time scale (delayed by 20 ms; Eq. 13; black) for the 25 most active trials of this cell. Trial number, correlation, and $N_{spk}$ are given above each panel. Panels are sorted according to $N_{spk}$. Circle: trial in (A). **c** Histogram of the correlations for all 664 trials (blue), and for those 350 trials with $N_{spk} \geq 15$ (brown). Yellow histogram: distribution of the correlations for randomly shuffled trials. **d** Distribution of the correlations for the 20 best-recorded cells for 3981 trials with $N_{spks} \geq 15$ (black). Mode is at $r = 0.87$, indicating that the instantaneous spike-density function explained > 75% of the instantaneous gaze-velocity profile at single-trial/single-unit level for most responses. The light-blue and light-pink histograms show the distributions for the shuffled trials for all cells and the 20 best cells, respectively. The latter two distributions are highly similar but differ from the normal data set; modes are near $r = 0.57$, explaining a mere 30% of the variation. Note also that many shuffled trials yielded negative correlations, which were very rare for the original data. See also Supplemental Figs. 11–16 to compare these results for different methods to calculate instantaneous firing rates.

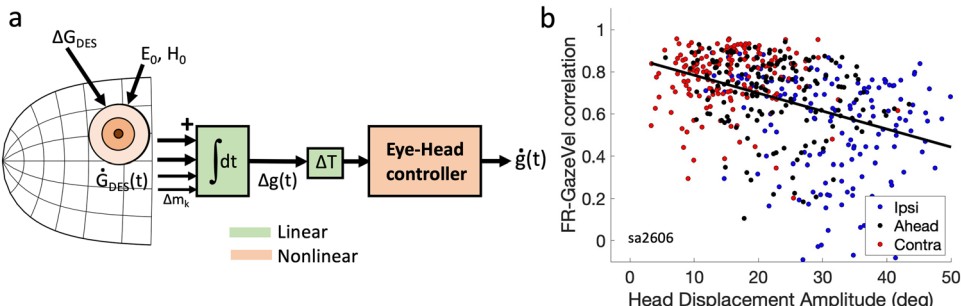

**Fig. 10 Conceptual scheme and influence of head movement. a** Dual feedforward encoding by the SC population: its location encodes the desired gaze-displacement vector, while the firing rates specify the desired gaze velocity. The population activity is modulated by initial eye and/or head position. The nonlinear eye-head gaze controller (see also Fig. 1b) is approximately compensated by a nonlinear control from the SC population, such that, effectively, the overall response from the system remains close to linear, like in Fig. 1a. Potential internal feedback from eye and head velocity (Fig. 1b) is omitted. **b** The correlation between gaze velocity and instantaneous firing rate is slightly, but significantly (correlation, $r = -0.41$), influenced by head-movement amplitude during the gaze shift (regression: offset: $r_0 = 0.87$; slope $-0.009$ deg$^{-1}$). This effect does not depend on initial eye orientation. The variability, however, is considerable (quality of the linear regression, $r^2 = 0.17$). Data from cell Sa2606 (489 trials with $N_{spk} \geq 15$). See Supplemental Fig. 17 for the population results of this analysis.

traces shown in Supplemental Fig. 11). Qualitative inspection suggests that the three different profiles tend to resemble each other, although the latter two are noisier than the 4 ms kernel profiles. As expected, because the adaptive spike-density function is noisier, SD-Gaze velocity correlations were overall lower than for the fixed 4 ms spike kernels (Supplemental Fig. 14). However, the histograms for the two spike-density methods yielded very similar results (Supplemental Fig. 15): although the median correlation for the adaptive method dropped to $r_{med} = 0.74$, the distribution was equally peaked as the 4 ms-histogram (at $r_{med} = 0.84$). Moreover, the shuffled data were very similar for the two methods. This result was further substantiated for the 20 best-recorded cells in Supplemental Fig. 16.

We conclude from these analyses that the high single-trial correlations between the instantaneous spike density and gaze-velocity profiles betray a real relationship that does not critically depend (qualitatively) on the smoothing method used to quantify the neural firing patterns.

## Discussion

We extended our model-based analysis of collicular saccade-related neural activity[33,54] (Eq. 3) to the full range of visual-evoked eye-head gaze shifts. In describing a cell's movement field, we included a potential influence of initial eye-in-head position on the number of spikes in the burst (static gain-field model, Eq. 9). Although, on average, the number of spikes in SC cells was not affected (Fig. 7b), changes in eye position strongly influenced the SC firing-rate profiles (i.e., their spike-density peak, ~shape, and burst durations) in all neurons (e.g., Fig. 3c). An eye-position signal in the motor SC has been reported before[70,71], but its potential role in controlling kinematics of gaze saccades has so far not been discussed.

Note that our paradigm cannot distinguish whether the observed influence on SC firing rates was caused by the change in initial eye position, initial head position, or both, as the behavioral conditions had been symmetrical, keeping the initial gaze direction at straight ahead for most recording sessions. Indeed, also head position has been shown to have access to the SC motor map[72,73]. To dissociate the influence of eye- vs. head orientation on SC firing behavior will require additional experiments in which both are independently manipulated.

We further showed a tight correlation of individual neurons between their spike-density profile and the velocity profile along the straight-line gaze-displacement. This relationship did not critically depend on the chosen smoothing method (Supplemental Figs. 11–16), or on the inclusion of the prelude spikes prior to the saccade onset (Supplemental Fig. 7). These results support the hypothesis that the SC population activity encodes the instantaneous kinematics of the desired gaze shift through its firing rates, whereas the gaze-shift amplitude is encoded by the number of spikes in the saccade-related bursts (Eqs. 1–3). The large proportion of high correlations of single-neuron activity profiles with the behavioral output (resulting from the joint action of many neurons), even at single-trial level and despite the substantial trial-to-trial noise of single-unit activity (Figs. 6 and 7c, d), suggests a tight synchronization of the temporal activity profiles within the recruited population.

For ipsilateral eye-in-head orientations, as well as for large gaze shifts ($\Delta G > 40$ deg), the head contribution to the gaze shift increased (Figs. 2b, 3 and S9), and gaze track-velocity (Fig. 2a) and associated peak SC spike densities (Fig. 3) decreased accordingly. For contralateral eye positions, the reverse held: peak gaze velocities and peak spike densities increased, and head-movement contributions decreased, with respect to eye-head aligned. In all cases, the number of spikes in the burst remained

virtually unchanged. For the far majority of cells, and for a large proportion of trials during each recording session, we obtained tight single-trial correlations ($r > 0.8$–$0.9$) between the instantaneous spike density of a single SC cell and the associated straight-line gaze-velocity profile of a saccade into its movement field (Fig. 9).

This result directly links to our model-derived Eqs. 10–13. Despite larger variability in instantaneous firing rates, partly due to increased noise from differentiating the spike-density function (e.g., Supplemental Fig. 14), and the intrinsic multiplicative noise in the spike counts (e.g., Figs. 6 and 7c, d), correlations remained high. We did not attempt to reduce this noise by additional low-pass filtering or optimizing the lead-delay for each cell (we kept it fixed at 20 ms). Thus, the obtained correlations may even have underestimated true values in some cases.

Our data show that measured phase trajectories relating the instantaneous straight-line gaze-displacement to the dynamic cumulative spike-count (Eqs. 3, 10, 11; Fig. 5b, Supplemental Figs. 4, 5, and 13) were virtually straight for all trials and with little variability (Supplemental Fig. 4b, c). This nontrivial property is a direct consequence of Eq. 3, which we here extended by simply replacing $\Delta e(t)$ through $\Delta g(t)$.

We also demonstrated that the considerable intrinsic trial-to-trial variability of SC activity for identical gaze shifts is best described by signal-dependent, multiplicative noise (Figs. 6 and 7c, d), with a negligible contribution from additive noise. These results corroborate our findings for head-restrained saccade-related SC bursts ([54] Goossens and Van Opstal, 2012). In particular, this property plays an important role in the implementation of optimal control strategies that aim to minimize motor-response variability ([3] Harris and Wolpert, 1998[6]; Van Beers, 2008; see below).

Single-trial and single-unit firing dynamics at a central neural stage correlate well with the instantaneous motor output of a highly complex and nonlinear synergistic system, comprising the multiple-degrees of freedom oculomotor, head-motor, and vestibular systems (see Fig. 1b).

With the head not moving, the overall dynamics of the (intact) oculomotor system seems relatively simple (Fig. 1a). Indeed, our analysis of SC bursts for visual-evoked and blink-perturbed eye-only saccades had shown that the entire downstream system could be conceptualized as a linear low-pass filter of the dynamic SC population command[33,54]; Fig. 1a; Eq. 2). However, because of the variable and task-dependent eye-head coupling (Figs. 1b, 2b, 3 and Supplemental Fig. 2), the inclusion of the VOR, the limited oculomotor range, the complexities of the head plant, and the eye-position influence on SC cells, each of which introduces its own dynamic nonlinearity to the system, the computational complexity of eye-head gaze-control is markedly increased. Moreover, the variable onsets of eye- and head-movements in the gaze shift, and thus their contribution and kinematics (Figs. 2 and 3 and Supplemental Fig. 2), depend on various factors, such as initial eye position, target eccentricity, stimulus modality, and task constraints, which can hardly be preprogrammed in its entirety, and differ fundamentally from head-restrained behaviors. In short, at first sight, one would not immediately expect that the instantaneous firing rates of single SC neurons would correlate at all with the kinematics of a planned (straight) gaze trajectory.

Just like in head-restrained monkeys[54], this important aspect of neural activity betrays a tight synchronization of SC burst profiles in the population also during eye-head gaze saccades. In our recent spiking-neural network models we showed that this synchronization could arise from excitatory-inhibitory lateral interactions among SC cells[74,75], in which a soft winner-take-all mechanism ensures that the most active cells (in the population center) impose their firing profile on the other neurons[54].

We recently extended our spiking-neural network model to eye-head gaze shifts and found that to yield qualitatively similar neural behaviors as shown in this study, the initial eye-position signal may be homogeneously distributed across the motor map, while only affecting two parameters of the model neurons through the same, but site-dependent, modulation mechanism, i.e. the neurons' adaptation time constant, and the local gain of their lateral connection strengths[65]. So far, the underlying biophysical mechanisms for such putative modulations, however, remain elusive.

Theoretical considerations hold that the saccadic main-sequence relations betray a speed-accuracy trade-off strategy that aims to minimize saccade-response variability despite the detrimental effects of multiplicative noise in its neural controls[3–6], together with spatial uncertainty in peripheral vision[76]. Optimal control theory then predicts that by reducing the pulse amplitude from the brainstem saccadic burst generator for large saccades, the system avoids the danger of large saccadic overshoots, which would be detrimental for rapid and accurate target acquisition and identification[76].

We proposed that such a mechanism would be best embedded at a level where signals are still encoded in an omnidirectional, abstract, vectorial format, rather than at the level of individual muscle-controls with many degrees of freedom. As such, the SC motor map would be an excellent candidate to embed a neural correlate for the optimal control of saccades[54]. A tight synchronization of the saccade-related burst profiles within the SC population, in combination with signal-dependent noise and the encoding of the nonlinear saccade kinematics in the motor map (see Introduction), may support this notion[54]. Indeed, simulating saccades with actual neural data applied to Eqs. 1 and 2 produced all kinematic features (main sequence, velocity profiles) and straight trajectories seen in real ocular saccades, without having to resort to an ad-hoc saturating brainstem nonlinearity and a complex cross-coupling scheme to explain the direction-dependent interactions of the horizontal and vertical movement components[33].

It seems reasonable to conjecture that also eye-head gaze shifts would follow an optimal control strategy, albeit that the movement cost to be minimized will likely differ from eye-only saccades. The oculomotor system only needs to account for speed (time to acquire the target) and accuracy/precision (foveation), where energy expenditure would be of minimal importance as the eye has negligible mass. The latter, however, may not be true for the head, and therefore a correlate for metabolic effort may be represented in the sensorimotor control as well[7–9].

We here speculate that the modulatory influence of eye position on SC firing rates (Fig. 3) could be an intrinsic neural adaptation to minimize the overall control cost (comprising errors, speed, effort) for combined eye-head gaze shifts in the presence of multiplicative noise (Figs. 6 and 7c, d). As the head's moment of inertia is considerable, and hence it's initial acceleration rather low when compared to the eye (Supplemental Fig. 2), the system could aim to minimize the head contribution to reduce metabolic costs and movement variability, and at the same time optimize speed.

However, because of the limited OMR, head movements unavoidably need to be planned for large gaze shifts, exceeding 30–35° (Fig. 2b). The intrinsic uncertainty in head-movement control is likely to be higher than for the eye, as the latter will hardly ever be perturbed by unexpected external loads or forces, and has relatively simple plant mechanics (only rotations). Moreover, larger forces are more strongly influenced by multiplicative noise, leading to increased outcome variability. In line with speed-accuracy trade-off, the central command from the SC should account for the added uncertainty in its gaze-control

signals when larger head movements are required. This would be achieved by lowering the SC firing rates (affecting gaze speed and energy use) without (appreciably) changing the total number of spikes (which, according to our model, determines endpoint accuracy and precision).

Conceptually, our finding that SC firing rates tend to correlate strongly with the straight-line gaze-velocity profile, at least for normal visual-evoked eye-head gaze shifts, suggests that the overall transfer between the SC output and the (planned) synergistic motor output, despite all the expected nonlinearities (Fig. 1b), may remain close to linear, even for the vastly more complex motor synergies. This idea is conceptually illustrated in the feedforward scheme of Fig. 10a.

However, unavoidable nonlinearities of (especially) the head-motor system are expected to perturb this simple near-linear input-output relationship, specifically when the relative contribution of the head to the gaze shift increases. This is illustrated in Fig. 10b for cell Sa2606, showing that the correlations become more variable, and tend to decrease (at $-0.009 \deg^{-1}$) with increasing head-movement amplitude for the gaze shift. Linear regression on these data shows a highly significant correlation of $r = -0.41$, but the low coefficient of determination ($r^2 = 0.17$) also indicates a considerable variability. This effect is most visible for the ipsilateral eye-position condition (blue symbols), which is associated with the largest range of head movements. Supplemental Fig. 17 summarizes these properties for the recorded population. Further computational modelling work and electrophysiological study will be needed to identify the underlying mechanisms for these properties.

## Methods

**General**. All experiments were performed in the Department of Neurobiology and Anatomy, School of Medicine and Dentistry of the University of Rochester, NY, while the author was a visiting scientist. Two female monkeys (P: 6.0 kg, and S: 4.5 kg) participated in the experiments. We recorded single units from 67 sites from the intermediate and deeper layers of the left SC (P: 46; S: 21), while monkeys made open-loop head-unrestrained gaze saccades to brief (50 ms duration) visual targets presented in the frontal hemifield in pseudorandom order. 43 single units were kept long enough to allow for a detailed analysis of their movement field (P: 28; S: 15). Cells were typically recruited for large gaze shifts (with optimal gaze-shift amplitudes between 10 and 100 deg, and optimal directions between −60 and +60 deg in the right hemifield; see Supplemental Fig. 18 and Supplemental Table 1). In this paper, 20/43 neurons were considered as 'best recordings', based on the following three criteria:

The number of successful trials exceeded 150 (range between 151 and 1006); most successful responses were made into the cell's movement field; optimal single-cell isolation with little background spiking activity, crisp saccade-related bursts, and reliable parameter estimates of the movement field.

**Animal preparation**. In the first aseptic surgery, a scleral search coil was implanted to monitor eye position[77]. During the same session, a small head-restraint device was secured to the skull, onto which a head-movement recording coil and a holder with three small lasers could be rigidly attached during recording sessions. After full recovery, animals were trained to make fast and accurate head-free gaze shifts to briefly lit visual targets against a small liquid reward. On the evening prior to the experimental session, the water supply was disconnected. The monkeys' health status and body weight were monitored daily. Outside experimental sessions animals could drink water ad libitum. In a second surgery, a recording cylinder (David Kopf

Instruments) was stereotactically placed over the SC. All surgical and experimental procedures were approved by the University of Rochester Animal Care and Use Committee and were in accordance with the National Institutes of Health Guide for the Care and Use of Animals[78–80].

During experiments, animals were seated with the head unrestrained and aligned on the body midline in the center of a 1.2 m cube containing three pairs of magnetic field coils (CNC Engineering). The front–back and left–right faces of the cube contained two pairs of coils in spatial and phase quadrature for accurate horizontal gaze- and head position recordings[81], while the top and bottom faces of the cube contained a third pair of coils for vertical eye- and head position. Accuracy and precision of the method was within 1 deg over the full range of gaze shifts.

Visual targets were presented by red (650 nm) laser diodes that projected on the inside of a 1.5 m diameter white acrylic hemisphere, the center of which was aligned with the center of the field cube.

**Data acquisition**. Custom-written software controlled behavioral and stimulus events and acquired and calibrated horizontal and vertical gaze- and head-position data at a sampling rate of 1 kHz per channel. To calculate gaze- and head velocities, we used the two-point central difference algorithm[82] and applied digital low-pass filtering with a bandwidth of 80 Hz (order 50) to each channel. The eye-in-head position was calculated by $E_{H,V}(t) = G_{H,V}(t) - H_{H,V}(t)$.

Standard amplification (Bak Electronics) and filtering techniques were used to isolate and record single neurons from the intermediate and deep layers of the Superior Colliculus (SC). A time-amplitude window (Bak Electronics) was used to isolate single-unit action potentials and store detected spikes as digital events at a sampling rate of 1 kHz. We also digitized the analog spike train at 25.0 kHz for off-line inspection of the recording quality. Supplemental Fig. 19 shows the recorded signals for an example trial.

Data files contained the calibrated head- and gaze data of the entire trial (typically lasting about 2.8–3.5 s), the detected spikes, and the raw spike signals, as well as the experimental details (e.g., window sizes, reward contingencies, and events). Files were stored in the hierarchical data format (hdf5) for off-line analysis in Matlab (version 2018b, The Mathworks).

**Experimental paradigm**. At the start of each trial, a small spot appeared, which the monkey had to fixate with the eyes. Typically, initial fixation was at the straight-ahead location, but in some cases (e.g., for very large up- or downward gaze shifts) the fixation spot was placed at a more convenient location. After a brief period of steady fixation between [800 and 1200 ms], one of the three head-fixed lasers turned on (pseudo-randomly selected in the experiment). The monkey was trained to rotate its head to align the laser light with the initial fixation point (within a tolerance of 5 deg) within a given time window between [1000 and 1200 ms], while maintaining eye fixation at the (typically straight-ahead) fixation point. In this way, the initial eye-in-head orientation could be $E_0 = -15$ deg (i.e., the eye-in-head looks left from the midsagittal plane of the head, and the head is rotated 15 deg rightward; this is the contralateral initial eye condition, or ipsilateral initial head), $E_0 = 0$ (eye and head aligned; the ahead condition), or $E_0 = +15$ deg (eyes look right from the midline; the head is rotated leftward; the ipsilateral initial eye condition, or contralateral initial head; see Supplemental Fig. S4, for an example).

In the results and figure legends we always describe the paradigm by the initial eye condition. For convenience, gaze- and

head-position traces will be shown aligned at zero degrees at movement onset (e.g., Figs. 3a, 9a, and Supplemental Fig. 2a,c).

As soon as the eyes and head were both in their reward windows, the fixation points both disappeared within 200–500 ms, and 150–200 ms later a brief (50 ms) peripheral visual target would appear at a randomly selected location within the 2D frontal hemifield. The monkey was required to make a fast and accurate eye-head gaze shift to the target within 500 ms, and within a tolerance of 8 deg from the target location.

As the target disappeared before gaze-shift onset (typical latencies around 180 ms) saccades were elicited under open-loop conditions. Target locations were selected for gaze shifts into and around the cell's estimated movement field, but sometimes they could be elicited away from the cell's response field. The monkey was rewarded with a drop of water for each successful trial. In a typical recording session, monkeys obtained between 1500 and 3000 rewards.

**Neural response selection**. In this paper, we tested the predictions that follow from the dynamic spike-counting model as explained in the Introduction. We therefore implicitly assume that only those collicular spikes that are sent to the brainstem while the so-called omnipause neurons in the brainstem raphe nucleus interpositus are disengaged can potentially contribute to the actual motor output (eye-head gaze shift). Thus, only those spikes that fall within a time window equaling the gaze-shift duration (which is the time during which the pause-neurons are inhibited[83]) have been included in our analyses. Given that collicular microstimulation typically yields saccades with a delay of about 15–20 ms, and that in many cells the saccade-related burst shows a sharp rising phase at a lead time of about 15–20 ms[45], we selected those spikes in the burst that started 20 ms before gaze-shift onset and ended 20 ms before gaze-shift offset.

We thus discarded all spikes that belong to the prelude activity (i.e., t < Onset −20 ms) and post-saccadic activity at t > Offset −20 ms (see, Fig. 4a, b). The effect of including the presaccadic prelude spikes on the spike density (firing rate) – gaze velocity correlations resulted to be very small. The reason for this is that the 4 ms spike-density kernel only influences the estimated activity level for a few samples (up to 8 ms) in the past and future, whereas these correlations were typically determined over saccade durations lasting 80 to 200 ms (see, e.g., Fig. 2a). This point is illustrated for cell Sa0107 in Supplemental Fig. 7. Clearly, for the cumulative number of spikes in the burst during the gaze shift, including the prelude had no effect at all as this activity measure necessarily starts at 0. Thus, the number of spikes in the burst and the associated (cumulative) spike-density functions (see below) were all calculated from the selected spikes only (Onset-20 ≤ spike timing ≤ Offset-20 ms).

**Static movement-field**. For every goal-directed gaze shift we identified the spikes in the saccade-related burst from 20 ms before gaze-shift onset to 20 ms before gaze-shift offset[45] (Fig. 9a, Supplemental Fig. 19). The total number of spikes in the burst, $N_{spk}$, was considered to depend on the gaze vector and the initial eye-in-head position. In quantifying the movement field, we used the complex-logarithmic afferent map of the SC[32] to project each gaze-shift vector $(\Delta G, \Phi)$ (deg) onto the anatomical coordinates $(u,v)$ (mm) of the SC motor map:

$$u = B_u \cdot \ln\left(\frac{\sqrt{\Delta G^2 + 2A \cdot \Delta G \cdot \cos\Phi + A^2}}{A}\right)$$

$$v = B_v \cdot \text{atan}\left(\frac{\Delta G \cdot \sin\Phi}{\Delta G \cdot \cos\Phi + A}\right) \tag{8}$$

Here, $B_u = 1.4$ mm, $B_v = 1.8$ mm/rad, and A = 3.0 deg, are fixed scaling- and shift parameters of the SC map, obtained from[31] Robinson's (1972) microstimulation data; $u$ extends along the rostral-caudal axis of the SC, and $v$ along the medial-lateral axis (see Supplemental Fig. 18). The cell's static movement field was specified by a Gaussian sensitivity profile in SC coordinates:[70,84]

$$N_{STMF}(\Delta G, \Phi, E_0) = N_0 \cdot (1 + \varepsilon \cdot E_0) \cdot \exp\left(-\frac{(u - u_0)^2 + (v - v_0)^2}{2\sigma_P^2}\right)$$
(9)

with $N_0$ the cell's maximum number of spikes, $u = u(\Delta G, \Phi)$ and $v = v(\Delta G, \Phi)$ are computed from Eq. 8 for each gaze shift; $u_0, v_0$ are the collicular coordinates for the cell's optimal gaze-shift vector, $\sigma_P$ (mm) is a measure for the extent of the movement field in SC coordinates, and $\varepsilon$ (#spikes/deg) is the potential eye-position sensitivity (gain-field modulation).

The five free parameters of this static gain-field model, [$N_0$, $\varepsilon$, $u_0$, $v_0$, $\sigma_P$], were determined by minimizing the least-squared-error of Eq. 9 for all successful gaze shifts with the measured spike counts. Supplemental Table 1 provides the fit results for all cells; see Figs. 4c and 5a, and Supplemental Fig. 3 for examples.

**Dynamic movement field**. The dynamic ensemble-coding model[33] of Eq. 1 in combination with a linear controller (Fig. 1a, Eq. 2) and assumed synchronized activity profiles within the population, predicts that the cumulative number of spikes in the burst, $n_{CS}(t)$ (from 0 to $N_{spk}$) varies linearly with the instantaneous (intended) gaze displacement along the straight line from the gaze-onset to gaze-offset position[33,54] (see Supplemental Fig. 20 for an illustration of this concept, and Fig. 5b and Supplemental Figs. 11 and 12, for results that further support this idea). Here, we extended the concept of Eq. 3 to instantaneous gaze shift trajectories:

$$n_{CS}(\Delta G, \Phi, E_0, t - \Delta T) = \frac{N_{STMF}(\Delta G, \Phi, E_0)}{\Delta G} \cdot \Delta g(t)$$
(10)

where $\Delta g(t)$ is the desired gaze trajectory along the straight line, increasing monotonically from 0 to $\Delta G$, and $\Delta T$ is the (fixed) delay between the SC burst and the ensuing gaze shift. For simplicity, we took $\Delta T = 20$ ms for all cells and trials (see, e.g., Figs. 4, 5b, and 9). The desired straight gaze trajectory was found by projecting the instantaneous horizontal and vertical components of the (potentially curved) gaze trajectory, [$g_H(t), g_V(t)$], onto the gaze-displacement vector, [$\Delta G_H, \Delta G_V$] (see Supplemental Figs. 11, 12 and 20:

$$\Delta g(t) = g_H(t) \cdot \cos \Phi + g_V(t) \cdot \sin \Phi$$
(11)

in which $\Phi$ is the overall direction of the gaze-shift vector:

$$\Phi = \text{atan}\left(\frac{\Delta G_V}{\Delta G_H}\right)$$
(12)

Note that this straight-line projected trajectory is not the same as the instantaneous vectorial gaze trajectory, which would be calculated as $\Delta g_{VEC}(t) \equiv \sqrt{g_H(t)^2 + g_V(t)^2}$.

The slope of Eq. 10 depends on the gaze-shift amplitude, $\Delta G \equiv \sqrt{\Delta G_H^2 + \Delta G_V^2}$, and the predicted number of spikes, $N_{STMF}$, from the static movement field (Eq. 9). We used the dynamic movement field of Eq. 10 to predict the instantaneous cumulative spike counts for all saccade trajectories of all recorded neurons.

According to Eq. 10, also the *time-derivates* of the left-hand and right-hand variables will be linearly related: if it holds, the instantaneous firing rate of a SC neuron, $\dot{n}(t)$, is expected to covary with gaze velocity along the intended straight line, $\dot{G}_{lin}(t)$:

$$\dot{n}(t - \Delta T) \propto \dot{G}_{lin}(t) \text{ with } \dot{n}(t) \equiv \frac{dn_{CS}(t)}{dt} \text{ and } \dot{G}_{lin}(t) \equiv \frac{d\Delta g(t)}{dt}$$
(13)

**Spike-density**. While Eq. 10 can be tested on the discrete, discontinuous spike-count signal, to test the prediction of Eq. 13 one needs to directly compare the instantaneous firing rate of the neuron with the instantaneous gaze velocity. To that end, the discrete and discontinuous spike train, $s(t) = \sum_i \delta(t - \tau_i)$, with $\tau_i$ the timing of spike $i$, has to be transformed into a continuous signal from which a meaningful derivative can be determined. It has been customary to calculate spike-density functions as continuous estimates for the instantaneous firing rate of a neuron[41,43,69]. Each spike is then typically convolved with a symmetric, normalized Gaussian kernel, K(t), with zero mean (other kernels, like a decaying exponential, or a gamma function, are used too):

$$d_i(t) = \int_{-\infty}^{+\infty} \delta(t - \tau_i) \cdot K(\tau_i) d\tau_i$$

and

$$K(\tau_i) = \left(\sigma_K \sqrt{2\pi}\right)^{-1} \exp\left[-\tau_i^2/(2\sigma_K^2)\right]$$
(14)

The sum of all convolved spikes (since the starting point $t_0$) then yields a continuous spike-density function, which provides a smooth estimate for the (discontinuous) instantaneous firing rate:

$$SD(t) = \int_{t_0}^{t} d_i(s) ds \approx \dot{n}(t)$$
(15)

The smoothing can be done in several ways; the simplest method uses a fixed kernel, for which $\sigma_K$ = constant. More sophisticated methods use an adaptive kernel, in which the kernel width varies with the local inter-spike interval: $\sigma_K(i) = f(\tau_{i+1} - \tau_i)$ for short intervals (high firing rates), the kernel width is reduced for better temporal resolution, while at longer intervals (low rates) the kernel is broadened to avoid small spurious oscillations in the estimate[41]. If the kernel width is too small, the spike-density function will be jagged and noisy like the discontinuous firing rate, but if it's too large, the smoothing will lose all major features of the original firing-rate profile.

Finding the optimal kernel-selection method, however, depends on the underlying spike statistics, research question, and properties of neural firing[85], which is a study in itself that falls beyond the scope of the present paper. Here, we compared two different methods: a fixed Gaussian kernel with a width of $\sigma_K = 4$ ms for all spike trains[43], and an adaptive procedure, in which the Gaussian kernel width was equal to the local inter-spike interval, i.e., $\sigma_K(i) = (\tau_{i+1} - \tau_i)$. Hence, the minimum kernel width was 1 ms, and the maximum could amount to several tens, or even hundreds of ms.

Supplemental Figs. 11 and 12a show the analysis of Eq. 10 for the 25 most active trials of cell Sa0107 for the raw spike count, as well as for the two kernel methods. Supplemental Fig. 12b, c shows the result for the different spike-count measures for the full data set of this cell (criterion: $N_{spk} \geq 15$; 350 trials). Supplemental Fig. 13 shows the results of Eq. 13 (firing rates vs. gaze-shift kinematics) for the same 25 saccades as shown in Supplemental Fig. 11, in the same format as Fig. 9b. In Supplemental Fig. 15, we show the result of Eq. 13 for the two kernel methods by repeating the correlation and shuffling analysis for all spike trains with

$N_{spk} \geq 15$ for this cell, and in Supplemental Fig. 16 for the pooled set of 20 best recordings.

As both kernel methods yielded similar results, in the Results section we report on the spike densities obtained with the fixed 4 ms kernel.

Note that the calculations of SD(t) and the neural spike count always started at the instantaneous level from $t_0 =$ Onset-20 ms and onwards. As explained in Neural Response Selection, in calculating the correlations between SD(t) and instantaneous gaze velocity, we did not correct for a potential pre-saccadic (prelude) contribution at $t_0 =$ Onset-20 ms (Fig. 3c, where SD(t) is shown over a longer time interval) for the following reasons: (1) according to the dynamic spike-count model of Eq. 1 the major output parameter of the SC is the cumulative spike count, with the spike density (firing rate) as a derived quantity. The spike count necessarily starts at zero. (2) Including the contribution of the prelude at $t = -20$ ms only had a minor effect on the overall correlation coefficients as it affected at most only the first few data points (for example, Supplemental Fig. 7).

**Statistics and reproducibility**. We fitted the nonlinear static gain-field model of Eq. 9 to the total number of recorded spikes in the burst, $N_{spks}$, of all successful trials by finding optimal fit parameters via the Melder-Nead simplex algorithm in Matlab's *fminsearch.m*. We verified the goodness of fit, quantified by the coefficient of determination, $r^2$, by applying linear regression on predicted ($N_{STMF}$) vs. measured $N_{spk}$ (e.g., Fig. 5a).

Linear regressions were performed only if the number of included data points exceeded $n = 10$. Optimal slopes and offsets (and their std) were obtained from bootstrapping the data (1000 times). In specified cases, we performed separate analyses on trials with $N_{spk} \geq 15$ to allow for more robust results. Univariate- and multiple linear regressions were done with *regstats.m* in Matlab.

To analyze the predictions of the ensemble-coding model for instantaneous gaze trajectories, we calculated dynamic spike-density functions, SD(t), from the discrete spike trains by convolving each detected spike with a unit-Gaussian ($\sigma = 4$ ms, fixed for all cells, spikes and trials). This yielded the smoothed estimate of the firing rate of the cell in spikes/ms; $FR(t) = 1000 \cdot SD(t)$ is the firing rate in spikes/s. The integral of SD(t) (*cumsum.m*) yielded the smoothed cumulative spike density function, $n_{CS}(t)$.

The fit-results of Eq. 9 were used to calculate predictions for the cumulative spike density (delayed by $\Delta T = 20$ ms) as a function of the instantaneous straight-line gaze displacement, Eq. 10 and 11, and to correlate the delayed firing-rate profiles with instantaneous straight-line gaze-velocity, Eq. 13.

Binwidths of histograms were calculated as $BW = \rho/\sqrt{N}$, with $\rho$ the range of the variable and N the number of data points. In the analysis of Fig. 9c, d, we determined the correlation between the gaze-velocity profile and instantaneous spike density for each trial (as in Fig. 9b).

For each experiment, we also randomly shuffled the order of all gaze velocities in the data set (*Shuffle.m*) and determined the correlations with the unshuffled spike densities over time intervals set by each newly paired gaze-shift duration. The rationale for this latter analysis was that if SC cells do not encode the detailed gaze-velocity kinematics, or if the two profiles would be highly stereotypical and similar (e.g., single peaked with a fixed duration) with little systematic variability, the original and shuffled correlation histograms should be approximately the same.

To test for a statistical difference between sampled data we used estimation statistics[86]. We report the strength of an effect by the mean difference with its bootstrapped 95% confidence interval (CI[87]).

On large sample sizes we employed one-tailed Kolmogorov-Smirnov tests to assess a difference (larger or smaller) between two distributions at a significance level of $p < 0.005$ (Fig. 9c, d).

**Reporting summary**. Further information on research design is available in the Nature Portfolio Reporting Summary linked to this article.

**Data availability**
The raw recording data and all data-analysis scripts and analysed data files and figures of the manuscript and Supplemental Material are stored in the data repository of the Donders institute at Radboud University[88] and will be made available upon reasonable request to the author.

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

## Acknowledgements

EU Horizon 2020 ERC Advanced Grant ORIENT no. 693400 (A.J.v.O.). The author is highly indebted to Ed Freedman, Mark Walton, Julie Quinet, and Stephan Quessy from the University of Rochester, NY, USA, for generously hosting him in their primate lab, helping him with the neural recordings, and to share their excellent facilities and expertise that made these experiments possible. The author also expresses his gratitude to both anonymous reviewers whose constructive criticisms and suggestions greatly helped to improve this manuscript.

## Author contributions

Conceptualization, data analysis, writing, and figure preparations: A.J.V.O.

## Competing interests

The authors declare no competing interests.
