## [Peer Review File · Communications Biology]

Reviewers' comments:

Reviewer #1 (Remarks to the Author):

Neurophysiological works in the awake behaving monkey are not only a very tough job but also socially thankless. Yet they are crucial for identifying the biological substrates of neurological and psychiatric disorders and for understanding the physiological processes that constrain our interactions with the external environment. People who are courageous to do this kind of job have become very rare because of its multiple difficulties.

This reviewer makes this introduction in order to congratulate the author for the beautiful set of data that he managed to collect in two monkeys, which were tested with the head unrestrained. Recording the activity of neurons in the head unrestrained monkey is a very challenging task and the collected data are rare and precious.

It has long been known that the sustained activity of neurons in the deep superior colliculus (SC) is an important aspect of their involvement in orienting gaze. The maintenance of firing only concerns the activity generated during saccades and not the complete activity that elapses from the target onset until the saccade end. Using the microstimulation technique, Drs Stanford, Freedman and Sparks indeed showed that if the SC activity is not long enough, the electrically-evoked movement is truncated; the amplitude specified by the locus of activity in the SC cannot be fully accomplished (Freedman et al., 1996; Stanford et al., 1996). In this article, the author completes this seminal result by recording the firing rate of SC neurons and proposes that, in addition to feeding the premotor neurons with the drive required to change the orientation of the eyes and head, the SC neurons also determine the instantaneous velocity of gaze movements. This proposal is important and timely because this kinematic control has recently been questioned by Zhang et al. (2022) and by Goffart et al. (2018, 2019) based on empirical and logico-theoretical arguments, respectively.

This reviewer made an extensive work in order to 1) help the author to sharpen his arguments, 2) sustain the important value of the data that he collected in the head-unrestrained monkey and 3) reward him for his efforts to complete such a challenge. However, she/he encountered several concerns. The author should be able to correct them because they do not concern the data per se, but their theoretical context and interpretation.

I. The first concern is that the author wrote his article in the spirit of providing arguments that support his preferred hypothesis without considering observations that refuted it, or without explaining why his hypothesis remains valid despite them. More specifically, the author ignored the observations that Lee, Rohrer and Sparks (1988) made on the accuracy and velocity of saccades after injecting small quantities of lidocaine in the SC. He also ignored the observations that Peel et al. (2021) made in the saccade-related burst of SC neurons after cryogenic inactivation of the FEF. From the point of view of this reviewer, these two studies refute the author's hypothesis because they invalidate its predictions: if the cumulative number of spikes in the burst of SC neurons dictates the vectorial gaze movement amplitude, then reducing the number of active neurons in the SC or reducing their firing rate should render the saccades hypometric.

A) Yet, inactivation of the center of the population of active neurons does not lead to hypometric saccades. The figure 3 in Sparks, Lee & Rohrer (1990) and in Lee, Rohrer & Sparks (1988) show that despite a severe reduction in their velocity, the saccades are normometric (see also figure 12A in Goffart et al. 2012).

B) Moreover, by cooling the FEF and recording its consequences on the firing rate of saccade-related burst neurons in the SC, Peel and colleagues (2021) found that the total number of spikes emitted by SC neurons was decreased while the amplitude of saccades remained unchanged. The number of spikes was reduced both at the center and throughout the entire movement field.

Therefore, before extending to the control of combined eye and head movements a hypothesis that is potentially invalid, the author should explain in the introduction why the results of these other studies

do not refute his dynamic ensemble-coding hypothesis.

II. The second major concern is the lack of consideration of results that relate to the firing rate of neurons targeted by the SC neurons, the premotor burst neurons constituting the so-called "brainstem burst generator":

C) Hu et al. (2007) showed in the head-restrained monkey that the premotor burst neurons do not fire at a rate that mimics the eye velocity profile but at a relatively constant rate. They reported instantaneous firing rates that resemble the recordings of van Gisbergen et al. (1981).

D) Walton & Sparks (2011) showed in the head-unrestrained monkey that the peak and average firing rates of premotor burst neurons decrease as the duration of the movement increases and as gaze amplitude increases.

From the point of view of this reviewer, these results are not compatible with the author's model, in which the instantaneous firing rate of neurons in the "brainstem burst generator" dictates the instantaneous eye velocity. Therefore, the author should explain in the introduction why these other empirical data do not invalidate his model. This concern leads to the next one.

III. The third concern is related to the interpretation of results. Studying the correlation between the firing rate of central neurons and the kinematics of gaze shifts should be made with caution. Several elements separate the spikes emitted by SC neurons and the contraction of extraocular muscle fibers. On the one hand, there are the post-synaptic neurons (premotor burst neurons) and their targets, the motor neurons. All these neurons do not passively transmit the collicular spikes to the extraocular muscles. On the other hand, the premotor burst neurons also receive crucial input from saccade-related neurons in the contralateral caudal fastigial nucleus whose activity influences the velocity and accuracy of saccades in the head restrained (Goffart et al. 2004) and the unrestrained monkey (Quinet & Goffart 2007). The fact that the author did not incorporate these findings in his model should not prevent him from warning the readers about the limitations of his hypothesis and about the blinkers of other theoreticians. By warning the readers, he offers to future readers the opportunity either to think about these other results or to understand why they can be discarded.

IV. Still related to results from past studies, the fourth concern is the delay of 20 ms between the emission of action potentials in the SC and the change in eye orientation. Why did the author choose a value of 20 ms instead of 8 ms? Miyashita & Hikosaka (1996) explained that when an electrical microstimulation was applied to the SC during a saccade, a small, conjugate contraversive eye movement was evoked with latencies much shorter than when the stimulation was applied while the eye fixated or did not move. The mean latencies of the stimulus-evoked eye movements were 7.9 ms (ipsilateral eye) and 7.8 ms (contralateral eye) when the stimulation was intrasaccadic. The author should warn the readers and discuss the consequences that taking a delay twice longer have upon the conclusions.

V. The fifth concern is that the author interprets his results within a theoretical framework that seems to rest upon no physiological evidence. In the discussion, the reader is taught "a trade-off for the oculomotor system that deals with the detrimental effects of multiplicative noise in its neural control" (L555). He/she is also taught that "by reducing the powerful, but noisy, pulse from the brainstem saccadic burst generator on the eye plant for large saccades, the system would thus avoid the danger of saccadic overshoots" (L559). In the introduction, the author tells us that "the saccadic gaze shifts result from a control principle that optimizes some joint performance criterion by trading off speed, accuracy and control effort, to minimize the impact of internal noise and uncertainty in the system and the resulting variability in its output performance measures". These speculations are based upon behavioral studies, which did not consider the reliable firing rate of burst neurons reported by Hu et al. (2007) in the pontine reticular formation. Therefore, the author should explain what he means with "internal noise", "uncertainty in the system" and "powerful, but noisy, pulse from the brainstem saccadic burst generator".

VI. Regarding the interpretation of Fig. 8B and Fig. 9B, it is not surprising that the correlation between

the number of spikes and the peak gaze velocity is weak. Indeed, the peak velocity is a measurement made at a specific instant whereas the number of spikes is a measurement made over a longer time interval, the duration of the complete gaze shift. This reviewer does not understand how spikes emitted AFTER the peak velocity of gaze shifts can be related to it. If a causal relation exists between the firing rate of neurons and the gaze velocity profile, then spikes emitted after the peak velocity cannot influence it anymore. Would not it be preferable to plot the relation between the gaze peak velocity and the number of spikes that were emitted before its occurrence? Or more simply, the relation between the average firing rate and the average velocity of gaze shifts?

Besides these general comments, the author will find below several difficulties that the reviewer encountered while reading this manuscript.

INTRODUCTION

L29-31: "Programming a gaze shift is a redundant task, as it can be generated by infinitely many combinations of eye-and head contributions" is an odd statement. The author assumes the existence of a "smart" gaze-control system that selects "reproducible movement strategies". This idea contrasts with the less anthropomorphic view according to which there is no selection. Given the orientation of the eyeballs in the orbits and the orientation of the head relative to the trunk, the combinations of the eye and head movements are not infinite but determined by neuro-muscular constraints.

L35: The meaning of the sentence "the saccadic gaze shifts result from a control principle that optimizes some joint performance criterion by trading off speed, accuracy and control effort" is unclear. Neurophysiologically speaking, what is a "control principle"? It is also unclear how a set of active neurons are informed about physical measurements, such as the amplitude of a movement, its duration and the amplitude/duration ratio? What does "control effort" mean?

L44: Bizzi et al. (1971) and Freedman & Sparks (2000) did not show that head movements depended on the initial-eye-in-head orientation.

L48-49: The idea that "the eye- and head movement interact within a common gaze-feedback loop" is refuted by several observations made in the caudal fastigial nucleus. This region contains a group of cells that project to the pontomedullary reticular formation (Noda et al. 1990) and that burst during saccades without modulating their firing during head movement (Fuchs et al. 2010). Their exclusive oculomotor function is confirmed by the absence of head movement when the caudal fastigial nucleus is electrically stimulated (Quinet & Goffart 2009) and also by the oculomotor deficits when muscimol is injected locally (Quinet & Goffart 2007). After muscimol injection, the amplitude and velocity of contralesional saccades are reduced and these reductions are not compensated, neither by an increase of saccade duration nor by an enhanced contribution of the head, refuting therefore the notion of a gaze feedback loop.

L57-58: The references Gandhi (2012) and Phillips et al. (1995) could be added because they also illustrate these relations in several monkeys.

L65: Scudder (1988) did not provide evidence for "a topographically organize motor map of saccadic eye movements" in the SC. Hafed & Chen (2016) did.

L66-68: The statement that "a large population of cells encodes amplitude and direction by the location of its center within the map" contained in the Superior Colliculus is an inaccurate statement. An injection of lidocaine in the peripheral part of the population of active neurons spares its center but renders saccades inaccurate (Lee et al. 1988; Sparks et al. 1990). Sparks and colleagues also showed that the saccades remain accurate when the center of the active population was inactivated (their velocity is reduced and their duration increased).

L68: The reference Anderson et al. (1998) could be added to support the recruitment of a large population of cells.

L73-74: Contrary to the author's claim, the results of Lee et al. (1988) do not lead to the conclusion that "higher mean firing rates lead to faster saccades than lower rates". By local injections of a small volume of lidocaine in the SC, they showed that the suppression of a subset of SC neurons leads to slower saccades.

L127-128: The references Pelisson et al. (1989), Guitton et al. (1990) are irrelevant to illustrate the point that the author wants to make. The reference Pare & Guitton (1994) is more appropriate. It

would be preferable to replace the reference Munoz et al. (1991) by Munoz & Guitton (1991) in order to prevent the unwary readers from thinking that the activity spreads across the SC during gaze shifts (see Sparks (1992) and Moschovakis et al. (2001)).

L181: "the firing rate of SC neurons during eye-head gaze shifts" instead of "the encoding of eye-head gaze shifts by SC neurons"

RESULTS

L230: It seems that the author means "horizontal and vertical amplitudes" instead of "endpoint" (which refers to a position).

Figure 3: The yellow dashed line is barely visible. A different color (green ?) would make it more visible. The interpretation of the plot A is impossible because all the movements are mixed. This reviewer and the readers would prefer to see the spikes emitted when the monkey made gaze shifts of comparable horizontal and vertical amplitudes. Separating gaze shifts made from the eyes centered in the orbits from those launched from deviated eyes would also be more informative than the mixture shown in panel A. An additional plot showing the same rasters aligned on gaze shift end would show whether this neurons stopped firing before gaze shift end (as Freedman & Sparks (1997) reported) or after (Choi & Guitton (2009)).

L242-244: the text tells us that Figure 4A compares the predicted and the measured number of spikes but it does not tell us what lesson should be taken from this comparison. It would be useful to document this comparison for movements with matched amplitude and direction. The readers are unable to see whether the big mismatch between the predicted and measured number of spikes is due to variable (uncontrolled) gaze metrics. For example, when we consider the case of 20 measured spikes, we see that the number of spikes predicted by equation (7) ranges between 14 and 23. We are also unable to know whether this prediction is good or not if we are not told what movement is associated with 14 and 23 (measured) spikes. In other words, how well does the equation predict the gaze shift amplitude based on the recorded spikes? The plots in Figure 4 show that the red, blue and black dots overlap. Unfortunately, the author does not explain the meaning of the overlap. Likewise, he does not explain the meaning of the slope of the relation between the cumulative number of spikes and the gaze-target vector (so-called "straight-line gaze displacements). Adding a plot showing individual phase trajectories that correspond to the same gaze amplitude would help the readers to interpret them.

L244-246: the sentence "The cell had its center ... spikes/deg" should be moved in the previous section. It should also be rewritten because reading that "the cell had its center at (X,Y)=(36.0, 23.3), a peak of $N_0=25$ spikes, and an eye-position sensitivity of 0.003 spikes/deg" sounds like a jargon. The author likely meant "the movement field of that cell".

L252-253: Is it not equation 9 that predict "cumulative spike counts"?

Figure 5: this reviewer would like to see the plots of cumulative spike count as a function of the gaze-target distance. They would provide support to the statement that "the phase trajectories were remarkably similar" (L270-271). In the legend, the word "velocity" should be inserted between "head-movement" and "profiles" (L276-277).

L268 and L270: "velocity" instead of "kinematics"

L296: "27 and 32 spikes" instead of "27.75 and 32.25 spikes"?

L303: "movement fields" instead of "gain fields"?

L315: Is N_c = measured number of cumulated spikes and N_c with circumflex accent = predicted number of cumulated spikes?

FIGURE 7B: title of y-axis: number of gaze shifts? "Number of Data Points" is unclear.

L331: "differences" instead of "deviations"

L333: "differences" instead of "errors"

L334: "difference" instead of "error"

FIGURE 8:

Different colors should be used to facilitate the distinction between the 7 clusters.

Plot A: titles of x- and y-axes: "Gaze-Amplitude" instead of "Gaze-shift"

Plot B:

Title of x-axis: what is Peak DESIRED Gaze Velocity? Is it not the MEASURED peak velocity of gaze

shift that is plotted?

Title of y-axis: "spk/s" instead of "Hz"

L346-347: "gaze-shift velocity" instead of "gaze-shift kinematics"

L346: The claim that "peak firing rate strongly covaried with peak velocity" is not convincing. The author's perception seems to be biased by the three largest values of peak gaze velocity. Did the author verify that these three gaze shifts were not recorded at the beginning of the recording session, i.e., when the monkey was the most motivated? Moreover, a careful look at the data points in Fig. 8B shows multiple examples where the peak firing rate does not change despite a doubling of peak velocity. The author should teach the readers how to read figures in order to prevent them from making premature conclusions.

L361: what is "gaze peak track-velocity"?

L364: what is the "velocity gain for the number of spikes"?

L369: the meaning of each variable should be specified.

L372: "a cell's maximum firing rate" instead of "a cell's firing rate". The word "strong" should be removed unless the author provides a numerical value. Moreover and more crucially, for the peak firing rate to be a "predictor" for the peak gaze velocity, the time of peak firing rate must precede the time of peak velocity. The author did not document this crucial information.

FIGURE 9:

Why are some dots red colored? Do they correspond to the cell sa1007 (as suggested by L472)?

L386: Considering that collicular spikes precede and contribute to the contraction of extraocular muscles, it is confusing to read that it is the instantaneous firing rate that is delayed instead of the gaze velocity.

The middle panel is meaningless because the interval during which the number of spikes is equal to gaze shift duration. During this interval, both the gaze position and the cumulative spike number increase. It is impossible that the cumulative spike number decrease. Some positive correlation is expected. Moreover, the author distorted the cumulative number of spikes. It should not appear as a continuous ramp but as a kind of staircase. The author should warn the readers about these numerical manipulations.

The plot at the bottom of Fig. 10A does not show the instantaneous firing rate but the convolved firing rate. Each spike was convolved by a Gaussian curve. This transforms a sequence of discrete events (the action potentials) into a continuous curve and enables the author to test the correlation with velocity profile. The author declares this correlation "reasonable" (L390) while this reviewer remains cautious because there is a gap between a single sequence of action potentials and the combination of multiple post-synaptic and poly-synaptic sequences. Studying this kind of correlation prevents the readers from realizing that, between the recorded SC neuron and the contraction of extraocular muscle fibers, there are intermediate neurons, at least two sets of neurons, the premotor neurons and the motor neurons. The premotor neurons receive spikes also from other neurons, some of which are located in the caudal fastigial nucleus and play a major role in the generation of gaze shifts.

L390: what means "25 top-activity trials"?

L394-395: the author should document the delay between the peak of the convolved firing rate and the peak gaze velocity. The difference between the time of smallest interspike interval (highest firing rate) and the time of the peak of the convolved firing rate should also be documented in order to help the readers estimate the impact of convolving the spikes with Gaussian curves.

L417-418: Is the "straight-line gaze displacement" the vectorial gaze displacement amplitude?

L386: what is "gaze track-velocity"? Is it different from the vectorial gaze velocity? If not, then the "gaze track-velocity" should be replaced by "vectorial gaze velocity" throughout the manuscript.

L428 and L445: If "gaze-track velocity" is the same as vectorial gaze velocity, then use the latter wording?

L428: what means "best-recorded cells"?

L434: the author should define what $P(r)$, P_0 and r mean.

L444: the word "convolved" should be inserted between "instantaneous" and "firing rate".

DISCUSSION:

L459: the small values of epsilon (eye position sensitivity factor) in Table S1 does not seem consistent

with the statement that "eye position strongly influenced the SC firing-rate profiles in all neurons". Indeed, all epsilon values are smaller than 0.02 spikes / degree, i.e., 1 spike / 50 degrees.

L463-465: This reviewer also thinks that the high firing rate of SC neurons increases the probability of observing synchronized action potentials within the population of bursting neurons. However, it is unclear why the correlation of the activity of single SC neurons with "the behavioral outcomes" leads the author to imagine "a tight synchronization of the activity patterns within the population". The sentence should be rephrased.

L472: It is unclear why Fig. 9 is mentioned. Do the red data points correspond to the unit #sa1007 during which these movements were recorded?

L476: "unchanged" instead of "unaffected"?

L478: A correlation between the instantaneous firing rate and instantaneous gaze velocity does not imply a causal chain. The changes in firing rate must precede the change in eye velocity.

L543-544: Contrary to the author's belief, Hepp et al. (1993) did not report that "large bilateral injections of muscimol in monkey demonstrated that the animal no longer generated any visual evoked saccades". Nothing was demonstrated in fact. Hepp et al. (1993) merely reported a reduction in the frequency and velocity of spontaneous rapid eye movements made spontaneously in the light. The amounts of reduction were not documented.

L544-547: Likewise, neither Goossens and Van Opstal (2006) nor Peel et al. (2020) or Zhang et al. (2022) showed that small local reversible lesions led to "immediate and specific deficits in the metrics (endpoints away from the lesion) and kinematics (substantially slower, by more than 20%) of saccades". These references should be removed. The study of Goffart, Hafed & Krauzlis (2012) could be added because it documented correlated changes in the latency and peak velocity of contralesional saccades after the local injection of a small amount of muscimol in the SC (their Fig. 12A).

FIGURE S3:

Is "track displacement" the same as vectorial displacement?

FIGURE S8:

The titles of y axes and the legend must be corrected. The plots A and C do not show gaze and head positions as a function of time but the amplitude of gaze and head displacements.

METHODS:

It is unclear whether the deviations of the head relative to body sagittal plane were only horizontal. Presumably, the values of horizontal and vertical orientation of the head were required to be within a range.

L703: the author should indicate how the accuracy of gaze shifts was controlled. Presumably, the gaze direction was required to land within a window relative to the target location. What was its radius?

Figure S1: scales for the gaze position and velocity should be added.

REFERENCES MENTIONED IN THE COMMENTS:

- Anderson, R. W., Keller, E. L., Gandhi, N. J., & Das, S. (1998). Two-dimensional saccade-related population activity in superior colliculus in monkey. *Journal of Neurophysiology*, 80(2), 798-817.
- Choi, W. Y., & Guitton, D. (2009). Firing patterns in superior colliculus of head-unrestrained monkey during normal and perturbed gaze saccades reveal short-latency feedback and a sluggish rostral shift in activity. *Journal of Neuroscience*, 29(22), 7166-7180.
- Fuchs, A. F., Brettler, S., & Ling, L. (2010). Head-free gaze shifts provide further insights into the role of the medial cerebellum in the control of primate saccadic eye movements. *Journal of neurophysiology*, 103(4), 2158-2173.
- Gandhi, N. J. (2012). Interactions between gaze-evoked blinks and gaze shifts in monkeys. *Experimental brain research*, 216, 321-339.
- Goffart, L. (2019). Kinematics and the neurophysiological study of visually-guided eye movements. *Progress in Brain Research*, 249, 375-384.
- Goffart, L., Bourrelly, C., & Quinton, J. C. (2018). Neurophysiology of visually guided eye movements: critical review and alternative viewpoint. *Journal of Neurophysiology*, 120(6), 3234-3245.

Goffart, L., Chen, L. L., & Sparks, D. L. (2004). Deficits in saccades and fixation during muscimol inactivation of the caudal fastigial nucleus in the rhesus monkey. *Journal of neurophysiology*, 92(6), 3351-3367.

Goffart, L., Hafed, Z. M., & Krauzlis, R. J. (2012). Visual fixation as equilibrium: evidence from superior colliculus inactivation. *Journal of Neuroscience*, 32(31), 10627-10636.

Hafed, Z. M., & Chen, C. Y. (2016). Sharper, stronger, faster upper visual field representation in primate superior colliculus. *Current Biology*, 26(13), 1647-1658.

Lee, C., Rohrer, W. H., & Sparks, D. L. (1988). Population coding of saccadic eye movements by neurons in the superior colliculus. *Nature*, 332(6162), 357-360.

Miyashita, N., & Hikosaka, O. (1996). Minimal synaptic delay in the saccadic output pathway of the superior colliculus studied in awake monkey. *Experimental brain research*, 112, 187-196.

Moschovakis, A. K., Gregoriou, G. G., & Savaki, H. E. (2001). Functional imaging of the primate superior colliculus during saccades to visual targets. *Nature neuroscience*, 4(10), 1026-1031.

Noda, H., Sugita, S., & Ikeda, Y. (1990). Afferent and efferent connections of the oculomotor region of the fastigial nucleus in the macaque monkey. *Journal of Comparative Neurology*, 302(2), 330-348.

Peel, T. R., Dash, S., Lomber, S. G., & Corneil, B. D. (2021). Frontal eye field inactivation alters the readout of superior colliculus activity for saccade generation in a task-dependent manner. *Journal of Computational Neuroscience*, 49, 229-249.

Phillips, J. O., Ling, L., Fuchs, A. F., Siebold, C., & Plorde, J. J. (1995). Rapid horizontal gaze movement in the monkey. *Journal of neurophysiology*, 73(4), 1632-1652.

Quinet, J., & Goffart, L. (2007). Head-unrestrained gaze shifts after muscimol injection in the caudal fastigial nucleus of the monkey. *Journal of neurophysiology*, 98(6), 3269-3283.

Quinet, J., & Goffart, L. (2009). Electrical microstimulation of the fastigial oculomotor region in the head-unrestrained monkey. *Journal of neurophysiology*, 102(1), 320-336.

Sparks, D. L. (1993). Are gaze shifts controlled by a 'moving hill' of activity in the superior colliculus? *Trends in neurosciences*, 16(6), 214-218.

Sparks, D. L., Lee, C., & Rohrer, W. H. (1990). Population coding of the direction, amplitude, and velocity of saccadic eye movements by neurons in the superior colliculus. In *Cold Spring Harbor symposia on quantitative biology* (Vol. 55, pp. 805-811). Cold Spring Harbor Laboratory Press.

Sparks, D. L., Holland, R., & Guthrie, B. L. (1976). Size and distribution of movement fields in the monkey superior colliculus. *Brain research*, 113(1), 21-34.

Van Gisbergen, J. A., Robinson, D. A., & Gielen, S. T. A. N. (1981). A quantitative analysis of generation of saccadic eye movements by burst neurons. *Journal of Neurophysiology*, 45(3), 417-442.

Zhang, T., Malevich, T., Baumann, M. P., & Hafed, Z. M. (2022). Superior colliculus saccade motor bursts do not dictate movement kinematics. *Communications Biology*, 5(1), 1222.

Reviewer #2 (Remarks to the Author):

Summary

Over the past decade, Dr. Van Opstal and colleagues have developed and tested a model for how the population of superior colliculus (SC) neurons generate saccadic eye movements. Here, using recordings of SC activity from macaque monkeys generating eye-head gaze shifts, Dr. Van Opstal extends this model to eye-head gaze shifts, and shows that many model predictions are borne out in the neurophysiological data. This is far from a trivial finding, given the many differences between the biomechanics of eye and head motion. The manuscript also includes a considerable sample of neural data collected throughout the extent of the SC, with the responses field of each neuron very well characterized throughout a range of initial positions. The dataset is all the more impressive given the head-unrestrained nature of the preparation. Overall, my opinion is that this manuscript represents a valuable contribution to the literature.

As should be clear from the above, I am enthusiastic about the manuscript. I have a number of points of feedback for further consideration.

1. The author correctly points out a number of the nonlinearities downstream of the SC during eye-head gaze shifts. The author also alludes to the complexity of the head plant in places, but I think more needs to be emphasized here, particularly early on in the manuscript. How the head converts a neural signal to force is itself quite a significant non-linearity, and the sense one gets is that the head is, to some degree, being treated like a big eye. Unlike the eye, the complexities of force generation at skeletal muscles moving an inertial system are substantial, being impacted by things such as initial position of the eye-in-head and head-on-body, as well as other drives to the head that may precede that associated with the gaze shift. The presence of such drives is apparent in the example shown in Fig S8, where the head movement precedes ipsi-gaze shift onset (as an aside, any thoughts on where that drive could be coming from?). I appreciate that incorporating such nonlinearities would go beyond the goals of the current paper, but I think it is important to acknowledge this non-linearity. Within the context of the current model, it strikes me as unlikely that the system can approximately compensate for this degree of non-linearity. The fact that the overall fits shown in the paper are so good may reflect aspects of the task, and/or that fact that the head is a comparatively smaller contributor to a gaze shift than the eye.

2. This line of thinking does lead me to wonder about the data presented in Figures 10 and 11, which show impressive trial-by-trial fits between the temporal profile of SC spikes and movement kinematics. Given my concerns on the comparative differences between the biomechanics of eye vs head motion, I wonder how much of this relationship is driven by similarities over the accelerating (or first half) of the gaze shift, when this is the main contributor to the gaze shift. I can imagine a variety of ways to address this possibility. One way would be to repeat these analyses for the first-half versus second-half of the gaze shift. Alternatively, one could split the analyses along the "ipsi, same, and contra" initial eye position, as done elsewhere. Finally, one could check if the quality of the fits decrease as the head's contribution increases (i.e., plot the correlations in Fig. 11 as a function of head contribution). Essentially, I am wondering if these fits would be worse for gaze shifts (or phases of gaze shifts) where the head is contributing more, presumably reflecting something about the nonlinearities inherent to head motion.

3. The author refers throughout the paper to eye-position gain fields, reflecting the instructed position of the eye-in-head prior to the gaze shift. In the model shown in Fig. 12, this signal is important in customizing the dynamics of SC activity. If I understand the experimental paradigm correctly, every eye-in-head position requires an equal-and-opposite head-on-body position, hence eye-in-head position is not the only thing being manipulated. Head-on-body gain fields have been reported in the SC (Nagy and Corneil, J Neurophys 2010), hence the signal impinging on the SC in Fig 12 may be better thought of as a signal reflecting initial configuration composed of a combination of eye-in-head and head-on-body position. This issue may be one of nomenclature for the current manuscript, but I can imagine head-on-body signaling being important as well for particularly large gaze shifts where the head could theoretically be reaching its limits for excursions relative to the body.

Rebuttal to the reviewers' comments.

Reviewer #1 (Remarks to the Author):

Neurophysiological works in the awake behaving monkey are not only a very tough job but also socially thankless. Yet they are crucial for identifying the biological substrates of neurological and psychiatric disorders and for understanding the physiological processes that constrain our interactions with the external environment. People who are courageous to do this kind of job have become very rare because of its multiple difficulties.

This reviewer makes this introduction in order to congratulate the author for the beautiful set of data that he managed to collect in two monkeys, which were tested with the head unrestrained. Recording the activity of neurons in the head unrestrained monkey is a very challenging task and the collected data are rare and precious.

I greatly appreciate these encouraging and supporting remarks.

*It has long been known that the sustained activity of neurons in the deep superior colliculus (SC) is an important aspect of their involvement in orienting gaze. The maintenance of firing only concerns the activity generated during saccades and not the complete activity that elapses from the target onset until the saccade end. Using the microstimulation technique, Drs Stanford, Freedman and Sparks indeed showed that if the SC activity is not long enough, the electrically-evoked movement is truncated; the amplitude specified by the locus of activity in the SC cannot be fully accomplished (Freedman et al., 1996; Stanford et al., 1996). In this article, the author completes this seminal result by recording the firing rate of SC neurons and proposes that, in addition to feeding the premotor neurons with the drive required to change the orientation of the eyes and head, the SC neurons also **determine** the instantaneous velocity of gaze movements. This proposal is important and timely because this kinematic control has recently been questioned by Zhang et al. (2022) and by Goffart et al. (2018, 2019) based on empirical and logico-theoretical arguments, respectively.*

*I would like to change the term 'determine' in: "specify the desired". I do not believe, nor propose, that the SC activity is as close to the motor output as the motor neurons. It simply cannot be, as the motor map is organised in polar coordinates (amplitude: rostral-caudal, direction: lateral-medial), whereas the motor output is highly redundant and multidimensional (6 extra-ocular muscles, and many more for the head). I therefore decided to slightly change the title and replace the word "**control**" by "**encode**", as it better captures this idea.*

Yet, I do believe that my recordings demonstrate a signal that closely resembles the final output gaze trajectory in many aspects. However, it should be clear, and I am well aware of this, that the SC signal can still be modified by the downstream circuitry. This is especially the case, so it seems, under acute lesions. This study, and its underlying concepts, deal with the intact system that generates visual-evoked gaze shifts to a single target under normal unperturbed conditions.

*There is another problem with the interpretation of the linear summation model that I now hope to have clarified better in the Introduction as well. The linear summation model of Eqn. 1 is a feedforward **population** model that generates a desired trajectory of gaze (head-unrestrained), or the eye (head-restrained), provided that the spiking patterns of all individual neurons are known. Here is the problem: under perturbed conditions, or in the presence of multiple options (more stimuli, the presence of an additional fixation light as in memory-evoked saccades, double electrical stimulation, etc.) the population activity of the motor map is not accessible to the experimenter. Although the model of Eqn. 1 says how all spikes contribute to the desired motor command, in those situations we cannot know all spikes. Therefore, only in the simple condition of a single target in otherwise darkness, and under fully randomized stimulus conditions, the SC motor map may be assumed (at least, this is our assumption) to only contain activity around the image point in the motor map that specifies the upcoming saccade goal.*

As such, the paper does not test the predictions of Eqn. 1 as we did in Goossens & Van Opstal, 2006 for eye-saccades. The main reason is that my recorded population is simply not large enough to allow for a full simulation of eye-head gaze shifts as in that earlier study (although I am still thinking of ways to get around this problem...). The paper DOES test, however, the relationship expressed by Eqn. 3. It is important to note that this equation is

not a direct prediction of the linear summation model of Eqn. 1, because Eqn. 1 does not prescribe or predict the activity patterns of the SC cells. It simply uses them. Eqn. 1 is a generative model: once the activity patterns are known, the (desired) gaze-shift trajectory follows. Eqn. 1 does not predict a straight gaze trajectory either, or a nonlinear main sequence, or a tight correlation between the firing rate of individual neurons with the gaze-velocity (the output of the total system). Only if all the cells in the population would fire in synchrony (i.e., their detailed firing-rate profiles are highly similar), and the effect of the downstream motor circuitries are close to a linear operation of the SC population output, one may expect these high single-cell – to – behavior correlations. The surprising result of my analysis is that they do, on a single-trial and single-cell level, for eye saccades, but also for eye-head saccades.

As to the reason why this all happens, one can only speculate. Although I find it hard to believe that there exist explicit ‘cost’ evaluators in the brain that calculate energy consumption, total jerk, or motor effort and the like, I can imagine that the intrinsic goal for gaze shifts is to optimize some overall behavioral performance measure: get as fast and as accurate as possible to the target with the least amount of effort. It may achieve this through (reinforcement) learning. Optimal Control Theory is in that sense a mathematical metaphor for such a process, but not a detailed neurophysiological account of how individual neurons would implement this.

Behavior, neural recordings, modelling, theory and (neural) implementation go hand in hand when we hope to finally understand the brain. I hope that my paper contributes to this general effort, although I only concentrated on the first four aspects.

This reviewer made an extensive work in order to 1) help the author to sharpen his arguments, 2) sustain the important value of the data that he collected in the head-unrestrained monkey and 3) reward him for his efforts to complete such a challenge. However, she/he encountered several concerns. The author should be able to correct them because they do not concern the data per se, but their theoretical context and interpretation. I thank the reviewer for the extensive efforts and the very constructive remarks.

I. The first concern is that the author wrote his article in the spirit of providing arguments that support his preferred hypothesis without considering observations that refuted it, or without explaining why his hypothesis remains valid despite them. More specifically, the author ignored the observations that Lee, Rohrer and Sparks (1988) made on the accuracy and velocity of saccades after injecting small quantities of lidocaine in the SC. He also ignored the observations that Peel et al. (2021) made in the saccade-related burst of SC neurons after cryogenic inactivation of the FEF. From the point of view of this reviewer, these two studies refute the author’s hypothesis because they invalidate its predictions: if the cumulative number of spikes in the burst of SC neurons dictates the vectorial gaze movement amplitude, then reducing the number of active neurons in the SC or reducing their firing rate should render the saccades hypometric.

These are indeed important observations, and we are well aware of them. We have dealt with some of these arguments in our earlier papers (e.g., Goossens & Van Opstal 2006; Van Opstal and Kasap, 2019, and Alizadeh & Van Opstal, 2022), and in my previous Introduction I alluded to these arguments, albeit, admittedly, briefly. I have now included a brief section in the Introduction (“Critiques on the model”: Lines 126 -161) that aims to recapitulate why I believe that the reported counterevidence does not conclusively kill the linear summation model.

A) Yet, inactivation of the center of the population of active neurons does not lead to hypometric saccades. The figure 3 in Sparks, Lee & Rohrer (1990) and in Lee, Rohrer & Sparks (1988) show that despite a severe reduction in their velocity, the saccades are normometric (see also figure 12A in Goffart et al. 2012).

Actually, this result could also be in line with the vector summation hypothesis, as explained in Goossens & Van Opstal 2006, when one additional assumption is added, namely that the downstream motor circuitry stops the saccade at a fixed number of total spikes from the population. See their Figure 10 for the full simulation results of a central lesion with the recorded spike data from the SC. Interestingly, Lee et al. ’88 interpreted their metrics results as conclusive evidence for the vector-averaging hypothesis (and against linear summation of Van Gisbergen et al., 1987) but their data do not explain why saccades also become slower after such a microlesion, as the vector averaging scheme does not specify the saccade kinematics at all. Therefore, lower, or higher firing rates in the SC should not affect the saccade velocity. However, it is well-established, not only by us, that they do. Also, the vector-averaging model does not explain why saccades become smaller when the stimulation current is lowered. All these factors can in principle be accounted for by the dynamic linear summation model, as is now also mentioned in the revised text (and with the additional Eqn. 4).

B) Moreover, by cooling the FEF and recording its consequences on the firing rate of saccade-related burst neurons in the SC, Peel and colleagues (2021) found that the total number of spikes emitted by SC neurons was decreased while the amplitude of saccades remained unchanged. The number of spikes was reduced both at the center and throughout the entire movement field.

Therefore, before extending to the control of combined eye and head movements a hypothesis that is potentially invalid, the author should explain in the introduction why the results of these other studies do not refute his dynamic ensemble-coding hypothesis.

First, a straightforward interpretation of the Peel et al. result is not easy as it concerns a different type of saccades: memory-guided responses, for which the total number of spikes in single cells was slightly reduced after an acute FEF inactivation. For visual-evoked saccades, however, their results did NOT show a difference. Second, as explained above, the summation hypothesis is a population hypothesis. We do not know whether under the circumstances of Peel et al. the overall number of spikes of the **population** was affected or not. We do not know either whether in the remembered-target condition there is only ONE site activated, and whether it is the same population of cells or not. In memory guided saccades there is additional activity in the fixation zone, which could lead to an interaction between two populations. This may sound like a lame excuse, but as explained above, the summation idea does NOT hold that single units should always correlate with the instantaneous kinematics of saccades, or that the number of spikes in a single neuron should always be the same! This is a wrong corollary of the idea. In an earlier account of my model, I argued that even if NO cells would correlate with the instantaneous velocity at all, the total population could still encode the detailed velocity profile of the saccade (Van Opstal and Kasap, 2019).

Still, the observation that under a wide variety of (normal) conditions cells DO correlate strongly with the motor output, is quite remarkable and indicates (I see no other option) that the population is highly synchronized during visual-evoked saccades to a single target. Finally, I do not want to claim (and perhaps I have inadvertently suggested this) that the SC fully 'controls' the true saccade output. In fact, the claim is that the cells in the SC send out a **desired, straight-line** gaze-velocity signal (i.e., in the case of a simple, single-target saccade task...). The motor-execution circuits that involve brainstem, cerebellum and spinal cord can still substantially alter the desired straight-line trajectory. Even top-down signals from FEF can modify the distribution of cells in the population, and hence the trajectory, but the current study does not deal with these potential complexities.

Yet, also in this study, we see that gaze trajectories to visual targets are close to straight in all directions. The assumption of a synchronized, single population at a particular site in the motor map seems reasonable and may explain this property.

II. The second major concern is the lack of consideration of results that relate to the firing rate of neurons targeted by the SC neurons, the premotor burst neurons constituting the so-called "brainstem burst generator": C) Hu et al. (2007) showed in the head-restrained monkey that the premotor burst neurons do not fire at a rate that mimics the eye velocity profile but at a relatively constant rate. They reported instantaneous firing rates that resemble the recordings of van Gisbergen et al. (1981).

I am not sure if I understand point C: Van Gisbergen et al. 1981 did show that PPRF burst cells encode the instantaneous eye velocity (although Cullen and Guitton later extended the burst analysis to also include acceleration in their regression models). Hu et al. 2007 focused on the variability of instantaneous PPRF firing rates for identical saccades. Although my study does not explicitly test the total downstream gaze-control system (as represented in simplified form in Fig. 1B), the simple model of Goossens and Van Opstal (2006) for eye saccades (Fig. 1A) DOES imply the same (average) relationships for brainstem burst cells as shown by Van Gisbergen et al., 1981. However, this simple model does not include noise in the population code for brainstem neurons. Better still, Fig. 1A is not intended as a detailed neurophysiological account of the brainstem-cerebellar oculomotor circuitry but acts as a proof of principle that a linear filter acting on the SC population signal fully accounts for the nonlinear main-sequence properties and the different shapes of saccade-velocity profiles. We do not claim, though, that individual stages in the brainstem saccadic burst generator act as just simple linear filters....

The difference with most saccade models is that our simple eye-saccade scheme achieves these results without having to assume a (ad-hoc) nonlinear input-output relationship between the instantaneous motor error and PPRF firing rate. In fact, we don't even need to include local feedback. Eqn. 2 summarizes the entire model of Fig. 1A by a simple feedforward linear filter with delay. In my view, this is the neural algorithm of the saccadic system. How this algorithm is actually implemented in the brain by neural populations is a different question, which we cannot resolve on the basis of my SC experiments.

D) Walton & Sparks (2011) showed in the head-unrestrained monkey that the peak and average firing rates of premotor burst neurons decrease as the duration of the movement increases and as gaze amplitude increases. From the point of view of this reviewer, these results are not compatible with the author's model, in which the instantaneous firing rate of neurons in the "brainstem burst generator" dictates the instantaneous eye velocity. Therefore, the author should explain in the introduction why these other empirical data do not invalidate his model. This concern leads to the next one.

The same holds for point D: if the data follow Eqn. 3, we again predict that the peak firing rate of brainstem burst cells will DECREASE with increasing gaze amplitude because of the involvement of the head. Note also that because of the VOR, the output of brainstem ocular burst cells does not represent the actual eye velocity, but a signal that varies, during the gaze shift, between eye-in-head velocity and gaze-in-space velocity, depending on the VOR gain.

Interestingly, the mechanism by how the SC code captures the main sequence of saccades may at first sight be surprising: the main obvious factor is that SC bursts increase their duration with increasing saccade amplitude. However, at the same time the peak firing rate of SC bursts across the motor map decreases from rostral (small saccades) to caudal locations (large gaze shifts; Goossens and Van Opstal, 2006). In that way, the total number of spikes of SC cells for their optimal saccade is invariant across the motor map (i.e., no significant relation with saccade amplitude). Here, I did not analyze this latter aspect specifically (it is hidden in Table S1), but I hope to have made clear in my Results that again the gaze-shift duration and SC burst duration are tightly coupled (e.g., the new Fig. 3C, and also Fig. 4B, which nicely shows the tight synchronization of the burst offset re. gaze-shift offset).

III. The third concern is related to the interpretation of results. Studying the correlation between the firing rate of central neurons and the kinematics of gaze shifts should be made with caution. Several elements separate the spikes emitted by SC neurons and the contraction of extraocular muscle fibers. On the one hand, there are the post-synaptic neurons (premotor burst neurons) and their targets, the motor neurons. All these neurons do not passively transmit the collicular spikes to the extraocular muscles. On the other hand, the premotor burst neurons also receive crucial input from saccade-related neurons in the contralateral caudal fastigial nucleus whose activity influences the velocity and accuracy of saccades in the head restrained (Goffart et al. 2004) and the unrestrained monkey (Quinet & Goffart 2007). The fact that the author did not incorporate these findings in his model should not prevent him from warning the readers about the limitations of his hypothesis and about the blinkers of other theoreticians. By warning the readers, he offers to future readers the opportunity either to think about these other results or to understand why they can be discarded.

I fully agree with these statements. That is why I do not claim to present and test a full model of the gaze-control system. I also believe that the SC output is NOT the true 'final' output of the saccadic system but specifies a desired (optimal) goal for the saccade that tells the control circuits downstream where it should direct gaze, but also in general how it should be done (straight-line kinematics). A complete model should also include the control complexity of the many degrees of freedom of the eye and head plants (3D rotations of the eye, 6 muscles; 3D rotations and also translations of the head: many more muscles). So far, no model can account for this complexity and I definitely do not claim to come even close to that with my conceptual scheme in Fig. 1B and (new) Fig. 11. My 'model' is intended to be a functional model, a computational algorithm if you will, of how the overall system behaves, not of how it implements it in its neural circuitry.

Based on the reviewer's valid remark, I decided to change the title of my paper into 'Neural Encoding of Instantaneous ...', to avoid the potential confusion that I assume that the SC directly controls the behavior.

IV. Still related to results from past studies, the fourth concern is the delay of 20 ms between the emission of action potentials in the SC and the change in eye orientation. Why did the author choose a value of 20 ms instead of 8 ms? Miyashita & Hikosaka (1996) explained that when an electrical microstimulation was applied to the SC during a saccade, a small, conjugate contraversive eye movement was evoked with latencies much shorter than when the stimulation was applied while the eye fixated or did not move. The mean latencies of the stimulus-evoked eye movements were 7.9 ms (ipsilateral eye) and 7.8 ms (contralateral eye) when the stimulation was intrasaccadic. The author should warn the readers and discuss the consequences that taking a delay twice longer have upon the conclusions.

As explained in the Discussion, I did not systematically play (aim to optimize) the delay but 20 ms is a typical response delay to microstimulation to generate a saccade from fixation; second, the burst of the SC often has a clear inflection around that time; third, the peak firing rate of fast gaze shifts falls right at gaze-shift onset, while the acceleration phase of the gaze shift is approximately 20 ms (see, e.g. Fig. 3C, red trace); finally, the burst offset falls around 20 ms before gaze-shift offset (e.g., Fig. 4B). If I would choose an 8 ms lead time, the

correlations between single-unit activity and behavior are much poorer; I therefore saw no immediate reason to do so. Others report about 15 ms, and it may even depend on the recording site, but to not overcomplicate (c.q., overfit) the analyses I simply kept it fixed.

V. The fifth concern is that the author interprets his results within a theoretical framework that seems to rest upon no physiological evidence. In the discussion, the reader is taught “a trade-off for the oculomotor system that deals with the detrimental effects of multiplicative noise in its neural control” (L555). He/she is also taught that “by reducing the powerful, but noisy, pulse from the brainstem saccadic burst generator on the eye plant for large saccades, the system would thus avoid the danger of saccadic overshoots” (L559). In the introduction, the author tells us that “the saccadic gaze shifts result from a control principle that optimizes some joint performance criterion by trading off speed, accuracy and control effort, to minimize the impact of internal noise and uncertainty in the system and the resulting variability in its output performance measures”. These speculations are based upon behavioral studies, which did not consider the reliable firing rate of burst neurons reported by Hu et al. (2007) in the pontine reticular formation. Therefore, the author should explain what he means with “internal noise”, “uncertainty in the system” and “powerful, but noisy, pulse from the brainstem saccadic burst generator”.

We had demonstrated in our earlier work that SC firing rates are endowed with multiplicative noise (Goossens et al., 2012). Triggered by the reviewer’s concern I here also analysed the noise properties of the population of neurons, which is now added in the Results section (Fig. 6 and Fig. 7C,D and Fig. S9). Clearly, the Optimal Control idea is a mathematical hypothesis that I did not specifically test here, but I included it in the Discussion to put the results in a computational context (see also my argument above). Clearly, time will tell whether these general mathematical concepts derived from Optimal Control Theory and Bayesian Inference will survive in the long run, but the appeal for such (abstract) theories is considerable, even though they may not have a one-to-one neurobiological correlate at the single-unit implementation level.

VI. Regarding the interpretation of Fig. 8B and Fig. 9B, it is not surprising that the correlation between the number of spikes and the peak gaze velocity is weak. Indeed, the peak velocity is a measurement made at a specific instant whereas the number of spikes is a measurement made over a longer time interval, the duration of the complete gaze shift. This reviewer does not understand how spikes emitted AFTER the peak velocity of gaze shifts can be related to it. If a causal relation exists between the firing rate of neurons and the gaze velocity profile, then spikes emitted after the peak velocity cannot influence it anymore. Would not it be preferable to plot the relation between the gaze peak velocity and the number of spikes that were emitted before its occurrence? Or more simply, the relation between the average firing rate and the average velocity of gaze shifts? *Maybe I misunderstand the reviewer, but the idea was that if the SC firing-rate profile reflects the details of the gaze-velocity profile (i.e., they have similar shapes, as suggested by (new) Eqn. 13), their peaks are expected to co-vary as well (at least for gaze shifts with identical metrics; the latter is needed to keep the number of spikes constant, as the integral of the velocity profile relates to the total number of spikes). As it is not easy to obtain data with exactly matching gaze vectors with a sufficient range of peak velocities, I succeeded in collecting data from 7 cells to illustrate this point. However, the more stringent test is the correlation analysis on the entire profile that follows in the two subsequent figures.*

Further, the SC activity leads the movement by about 20 ms. So indeed, it could make sense that peak firing rate and peak velocity may correlate. The number of spikes in the burst are not expected to correlate with peak velocity, as the reviewer notices, with one caveat: if the number of spikes changes because of changes in initial eye position (it sometimes does), and the peak velocity changes because of changes in eye position, then the peak velocity and number of spikes can show a spurious relationship. That’s why a multiple linear regression (Eqn. 6) can dissociate whether these two effects had a common cause.

Finally, I fully follow the argument of the reviewer regarding the number of spikes, but it presupposes that there is indeed a relation between instantaneous SC firing behavior and instantaneous motor output! If, e.g., the SC spikes would be merely integrated and only the brainstem fully controls the saccade trajectory after a trigger has been issued (as proposed in the Scudder model), the peak velocity WILL change with the number of (pre-) integrated spikes!

Besides these general comments, the author will find below several difficulties that the reviewer encountered while reading this manuscript.

INTRODUCTION

L29-31: "Programming a gaze shift is a redundant task, as it can be generated by infinitely many combinations of eye-and head contributions" is an odd statement. The author assumes the existence of a "smart" gaze-control system that selects "reproducible movement strategies". This idea contrasts with the less anthropomorphic view according to which there is no selection. Given the orientation of the eyeballs in the orbits and the orientation of the head relative to the trunk, the combinations of the eye and head movements are not infinite but determined by neuro-muscular constraints.

I disagree. They are bounded by neuro-muscular constraints, but there are infinitely many possible combinations. But point taken, I replaced the word 'infinitely' by 'many'.

L35: The meaning of the sentence "the saccadic gaze shifts result from a control principle that optimizes some joint performance criterion by trading off speed, accuracy and control effort" is unclear. Neurophysiologically speaking, what is a "control principle"? It is also unclear how a set of active neurons are informed about physical measurements, such as the amplitude of a movement, its duration and the amplitude/duration ratio? What does "control effort" mean?

As explained in the first part of my rebuttal, it is not my aim to assign neurophysiological correlates to specific details of the optimal control idea. There is a difference between a 'neural algorithm' performing a well-defined function on the one hand, and a specific neural implementation that instantiates that algorithm at single-unit and even transmitter level. By itself, formulating mathematical and quantitative hypotheses about the underlying neural algorithms, which are derived from detailed behavioral study, is a valid and useful way to gain better understanding of the brain, and eventually of the neural implementation. This point is often missed entirely in rodent work (nicely discussed in the commentary article by Krakauer and colleagues in Neuron, 2017).

L44: Bizzi et al. (1971) and Freedman & Sparks (2000) did not show that head movements depended on the initial-eye-in-head orientation.

Ok, removed.

L48-49: The idea that "the eye- and head movement interact within a common gaze-feedback loop" is refuted by several observations made in the caudal fastigial nucleus. This region contains a group of cells that project to the pontomedullary reticular formation (Noda et al. 1990) and that burst during saccades without modulating their firing during head movement (Fuchs et al. 2010). Their exclusive oculomotor function is confirmed by the absence of head movement when the caudal fastigial nucleus is electrically stimulated (Quinet & Goffart 2009) and also by the oculomotor deficits when muscimol is injected locally (Quinet & Goffart 2007). After muscimol injection, the amplitude and velocity of contralesional saccades are reduced and these reductions are not compensated, neither by an increase of saccade duration nor by an enhanced contribution of the head, refuting therefore the notion of a gaze feedback loop.

Thanks, I will update this sentence by also referring to the counter evidence. For my paper, however, the exact organization of the gaze-control system (feedback, or no feedback) is not a crucial point.

L57-58: The references Gandhi (2012) and Phillips et al. (1995) could be added because they also illustrate these relations in several monkeys.

Ok

L65: Scudder (1988) did not provide evidence for "a topographically organize motor map of saccadic eye movements" in the SC. Hafed & Chen (2016) did.

Ok, took it out.

L66-68: The statement that "a large population of cells encodes amplitude and direction by the location of its center within the map" contained in the Superior Colliculus is an inaccurate statement. An injection of lidocaine in the peripheral part of the population of active neurons spares its center but renders saccades inaccurate (Lee et al. 1988; Sparks et al. 1990). Sparks and colleagues also showed that the saccades remain accurate when the center of the active population was inactivated (their velocity is reduced and their duration increased).

I took out the word 'center' in my revised statement, and added the Lee et al. and Anderson studies there.

L68: The reference Anderson et al. (1998) could be added to support the recruitment of a large population of cells.

Ok, included

L73-74: Contrary to the author's claim, the results of Lee et al. (1988) do not lead to the conclusion that "higher mean firing rates lead to faster saccades than lower rates". By local injections of a small volume of lidocaine in the SC, they showed that the suppression of a subset of SC neurons leads to slower saccades.

Indeed. This point actually argues against a mere 'metrics' role for the SC in saccades, which cannot be accounted for by a simple vector averaging model. Instead, the vector summation model can, as we demonstrated earlier (Goossens and Van Opstal, 2006). I took up this point in the revised Introduction ("Critiques of the model", lines 126-138).

L127-128: The references Pelisson et al. (1989), Guitton et al. (1990) are irrelevant to illustrate the point that the author wants to make. The reference Pare & Guitton (1994) is more appropriate. It would be preferable to replace the reference Munoz et al. (1991) by Munoz & Guitton (1991) in order to prevent the unwary readers from thinking that the activity spreads across the SC during gaze shifts (see Sparks (1992) and Moschovakis et al. (2001)).

Thanks, done.

L181: "the firing rate of SC neurons during eye-head gaze shifts" instead of "the encoding of eye-head gaze shifts by SC neurons"

Ok, done.

RESULTS

L230: It seems that the author means "horizontal and vertical amplitudes" instead of "endpoint" (which refers to a position).

The panel to which this refers has been removed in the revised manuscript and is replaced by the new panel Fig. 4C.

Figure 3: The yellow dashed line is barely visible. A different color (green ?) would make it more visible. The interpretation of the plot A is impossible because all the movements are mixed. This reviewer and the readers would prefer to see the spikes emitted when the monkey made gaze shifts of comparable horizontal and vertical amplitudes.

Separating gaze shifts made from the eyes centered in the orbits from those launched from deviated eyes would also be more informative than the mixture shown in panel A. An additional plot showing the same rasters aligned on gaze shift end would show whether this neurons stopped firing before gaze shift end (as Freedman & Sparks (1997) reported) or after (Choi & Guitton (2009)).

I have re-arranged the spike trains of the new Fig. 4 by initial condition (color) and number of spikes (order). Although the metrics vary within each cluster, they vary in the same way, and therefore, the changes in burst duration for contra to central to ipsi-eye positions is made more visible.

I also added the additional plot to also show the spike timings re. gaze-shift end for all 664 trials (Fig 4B). It can be clearly seen that indeed the bursts end approximately 20 ms before the gaze-shift offset, and I refer to the two papers mentioned by the reviewer.

I now also made a new illustration for this point in the new Figure 3, panel C; there, I selected about 30 large gaze shifts (50 deg amplitude) with the same amplitude (within 1 deg) and similar directions into the movement field of the cell and sorted the spike trains according to initial eye position. These results are typical and show that the burst intensity and duration depend strongly on the initial condition.

L242-244: the text tells us that Figure 4A compares the predicted and the measured number of spikes but it does not tell us what lesson should be taken from this comparison. It would be useful to document this comparison for movements with matched amplitude and direction. The readers are unable to see whether the big mismatch between the predicted and measured number of spikes is due to variable (uncontrolled) gaze metrics.

Perhaps the reviewer assumes that data have been selected here for similar gaze vectors. However, this is not the case, and it is not needed either, as the fit results of the static movement-field model (Eqn. 9) applies to every saccade of the experiment and all initial conditions. Here, the number of responses is 664. The graph in (new) Fig. 5A shows the prediction of the model vs. the measured number of spikes for each saccade and illustrates the inherent variability in the measured number of spikes for a single neuron. The noise may seem substantial, but it is a normal property of SC neurons (as I show in the revised version; see also below). Despite this variability,

however, the correlation between prediction and measurement is 0.93, which indicates that the static model is actually very a good predictor for the neuron's number of spikes (on average).

For example, when we consider the case of 20 measured spikes, we see that the number of spikes predicted by equation (7) ranges between 14 and 23.

I show in the new Figure 6 that this variation is due to multiplicative noise. I have now analysed this property in detail and show the data in three additional figures: Figure 6, Fig 7C,D and Supplemental Figure S9.

We are also unable to know whether this prediction is good or not if we are not told what movement is associated with 14 and 23 (measured) spikes.

In the plot of Fig. 5A it this doesn't matter what the underlying movement metrics were, as every predicted point includes the amplitude, direction and initial position of the associated saccade (Eqn. 9). Note also that a fixed number of spikes is expected for a wide variety of saccades, which all lie on an 'ellipse-like' iso-activity contour. I now refer to this in the text by also referring the reader to the white contours in the movement-field plot of Fig. 4C and Fig. S5.

In other words, how well does the equation predict the gaze shift amplitude based on the recorded spikes? The short answer is: not very well, because of the abovementioned reason (many saccades generate the same number of spikes), and because of multiplicative noise in the response. Further, a single cell does not necessarily explain anything, as it is the total population of cells that determines the saccade amplitude and direction. Unless all cells do the same thing!

The plots in Figure 4 show that the red, blue and black dots overlap. Unfortunately, the author does not explain the meaning of the overlap.

This shows that the static movement-field model accounts for any potential variation in the number of spikes that is due to the change in initial eye position.

Likewise, he does not explain the meaning of the slope of the relation between the cumulative number of spikes and the gaze-target vector (so-called "straight-line gaze displacements). Adding a plot showing individual phase trajectories that correspond to the same gaze amplitude would help the readers to interpret them.

This analysis has been done extensively in Goossens and Van Opstal, 2006. Again, it should be noted that the total spike count does not predict the gaze amplitude. Therefore, showing these trajectories for a fixed amplitude is fine (it is actually done in Fig. 5B, for all 664 trials (and this is the raw data), and compared to the model prediction of Eqn. 10 in Fig. 5C, which explains the wide variation in phase trajectories seen in Fig. 5B), but will lead to many different (yet straight) trajectories for the same amplitude, depending on where the vector sits in the cell's movement field. This trajectory is prescribed by Eqn. 10, and (as explained above), it is NOT a prediction of the ensemble-coding model but reflects a remarkable property of SC firing behavior. I have argued in the text that this relation can only be understood if one assumes that all cells in the population have synchronous firing profiles, as it relates the detailed time behavior of a single neuron to the behavioral eye-head motor response, which results from the joint action of (hundreds of) thousands of neurons.

L244-246: the sentence "The cell had its center ... spikes/deg" should be moved in the previous section. It should also be rewritten because reading that "the cell had its center at (X,Y)=(36.0, 23.3), a peak of No=25 spikes, and an eye-position sensitivity of 0.003 spikes/deg" sounds like a jargon. The author likely meant "the movement field of that cell".

[ok]

L252-253: Is it not equation 9 that predict "cumulative spike counts"?

[yes, it is now Eqn. 10]

Figure 5: this reviewer would like to see the plots of cumulative spike count as a function of the gaze-target distance. They would provide support to the statement that "the phase trajectories were remarkably similar" (L270-271). In the legend, the word "velocity" should be inserted between "head-movement" and "profiles" (L276-277).

This figure is now moved to the Supplemental Material. The top-left inset (in Fig. 4C and S5) shows the instantaneous cumulative spike count for the three (averaged) optimal gaze shifts, measured vs. predicted. It demonstrates that for the three initial eye-position conditions the trajectories superimpose ('remarkably similar').

L268 and L270: "velocity" instead of "kinematics" [ok]

L296: "27 and 32 spikes" instead of "27.75 and 32.25 spikes"? [done]

L303: "movement fields" instead of "gain fields"? [ok]

L315: Is N_c = measured number of cumulated spikes and N_c with circumflex accent = predicted number of cumulated spikes?

I adapted the equation and text and hope to have now better explained this analysis.

FIGURE 7B: title of y-axis: number of gaze shifts? "Number of Data Points" is unclear.

It's number of data points (samples), as the analysis is done for every sample of the trajectories. I changed the legend to 'Number of Data Samples' in the (new) figure 8.

L331: "differences" instead of "deviations" [ok]

L333: "differences" instead of "errors" [ok]

L334: "difference" instead of "error" [ok]

FIGURE 8:

Different colors should be used to facilitate the distinction between the 7 clusters. [ok, DONE]

Plot A: titles of x- and y-axes: "Gaze-Amplitude" instead of "Gaze-shift" [ok]

Plot B:

Title of x-axis: what is Peak DESIRED Gaze Velocity? Is it not the MEASURED peak velocity of gaze shift that is plotted?

yes, this was a mistake. It's the peak gaze velocity

Title of y-axis: "spk/s" instead of "Hz"

Changed

L346-347: "gaze-shift velocity" instead of "gaze-shift kinematics"

[ok]

L346: The claim that "peak firing rate strongly covaried with peak velocity" is not convincing. The author's perception seems to be biased by the three largest values of peak gaze velocity. Did the author verify that these three gaze shifts were not recorded at the beginning of the recording session, i.e., when the monkey was the most motivated? Moreover, a careful look at the data points in Fig. 8B shows multiple examples where the peak firing rate does not change despite a doubling of peak velocity. The author should teach the readers how to read figures in order to prevent them from making premature conclusions.

I agree with the reviewer and have modified the text accordingly. The correlation with peak velocity is highly significant, but there is also quite some (expected) trial-to-trial variability.

In fact, looking at the data presented in the Zhang et al. 2022 paper, where they try to make the point that the saccade peak velocity and peak firing rate are fully dissociated is not convincing either: the authors show that cells have large variability (the vertical axis in their Fig. 8d, but the variation in saccade kinematics is in fact very limited.

Here, I show a very large variation in natural peak velocities (between 250 deg/s and 950 deg/s), and then the correlation appears. This actually refers to an old and problematic point: the variability in individual neurons is much larger than the typical natural behavioral variability. That's way it has been difficult in past studies to claim a solid relationship between SC firing rates and saccade kinematics. To show such a correlation at the single-cell level requires a large variation in the behavior as well. This was exactly the goal of my paper, and it was also in Goossens and Van Opstal, 2006 (where the natural behavioral variability was induced by blink perturbation). Inducing behavioral variability through lesions or microstimulation, however, may lead to other interpretation problems for lack of information regarding the neural population dynamics and possible alternative neural pathways that may become involved.

L361: what is "gaze peak track-velocity"? [changed]

L364: what is the "velocity gain for the number of spikes"? [changed]

L369: the meaning of each variable should be specified. [done]

L372: "a cell's maximum firing rate" instead of "a cell's firing rate". The word "strong" should be removed unless the author provides a numerical value. Moreover and more crucially, for the peak firing rate to be a

“predictor” for the peak gaze velocity, the time of peak firing rate must precede the time of peak velocity. The author did not document this crucial information.

I changed the wording. Note that the peak firing rate actually does lead the peak velocity (by about 20 ms, as is shown in Figure 3C (red curve), but also in Figure 10B, which shows all the time-shift firing-rate profiles, superimposed on the gaze-velocity profiles, indicating that the peaks often coincide.

FIGURE 9:

Why are some dots red colored? Do they correspond to the cell sa1007 (as suggested by L472)?

I now explain that the red dots represent the fit results of univariate linear regression on the full data set of 147 responses, shown for 5 values 200 deg/s apart.

L386: Considering that collicular spikes precede and contribute to the contraction of extraocular muscles, it is confusing to read that it is the instantaneous firing rate that is delayed instead of the gaze velocity.

Why is that? It collicular spikes would ‘cause’ the gaze shift, the latter is expected to follow the former. The spikes lead the movement by about 20 ms, so in order to align the two traces, the spikes have to be delayed by this amount.

The middle panel is meaningless because the interval during which the number of spikes is equal to gaze shift duration. During this interval, both the gaze position and the cumulative spike number increase. It is impossible that the cumulative spike number decrease. Some positive correlation is expected.

Yes, that is true, and therefore the correlation with velocity is more informative. Still, if the SC had nothing to do with encoding gaze velocity, and all spikes would have been emitted, e.g., during the first 30 ms, there would be no straight-line relationships (as in Fig. 5B,C and 8) between the cumulative number of spikes and the instantaneous gaze displacement. rather than being distributed in this particular way across the gaze shift. Now, this correlation is 0.993, which is really high.

Moreover, the author distorted the cumulative number of spikes. It should not appear as a continuous ramp but as a kind of staircase. The author should warn the readers about these numerical manipulations.

These are not manipulations but have been done for visual purposes; they have no influence on the data analysis, where the cumulative spike count is really a discrete number. The firing-rate profiles and their correlations with gaze velocity have been derived from the spike-density curves.

The plot at the bottom of Fig. 10A does not show the instantaneous firing rate but the convolved firing rate. Each spike was convolved by a Gaussian curve. This transforms a sequence of discrete events (the action potentials) into a continuous curve and enables the author to test the correlation with velocity profile. The author declares this correlation “reasonable” (L390) while this reviewer remains cautious because there is a gap between a single sequence of action potentials and the combination of multiple post-synaptic and poly-synaptic sequences.

I agree; this trial was actually the worst example of the set of 25 that I provide in panel B! (I have nothing to hide...) Indeed, there is, clearly, substantial trial-to-trial variability in a single-neuron’s response. Not all trials yielded high correlations with the gaze shift for all cells. However, I demonstrate that the mode of the correlations across all trials is around $r=0.85$. Not only for this cell, but for all cells, which is remarkably high and cannot be explained by stereotyped firing behaviors as demonstrated by random shuffling the data.

Studying this kind of correlation prevents the readers from realizing that, between the recorded SC neuron and the contraction of extraocular muscle fibers, there are intermediate neurons, at least two sets of neurons, the premotor neurons and the motor neurons. The premotor neurons receive spikes also from other neurons, some of which are located in the caudal fastigial nucleus and play a major role in the generation of gaze shifts.

I hope to have made this clear in the revised Introduction (and also in the Discussion).

L390: what means “25 top-activity trials”?

This is now explained. I selected the 25 trials with the highest number of spikes.

L394-395: the author should document the delay between the peak of the convolved firing rate and the peak gaze velocity. The difference between the time of smallest interspike interval (highest firing rate) and the time of the peak of the convolved firing rate should also be documented in order to help the readers estimate the impact of convolving the spikes with Gaussian curves.

As I explain in the Discussion and Methods, I took a fixed delay of 20 ms and a fixed kernel width of 4 ms. I did not attempt to make it look better than this by fitting some optimal delay parameter and optimal kernel width for each cell, or even for each trial. Others have compensated the kernel width for higher firing rates to get better profiles, but I decided not to do this to not run into issues over over-fitting and over-interpretation.

L417-418: Is the “straight-line gaze displacement” the vectorial gaze displacement amplitude?

No, it is not. It is the projection of the instantaneous (curved) gaze trajectory onto the straight line that connects the initial and final gaze positions, as calculated by Eqn. 11, and illustrated in Fig. S3, as:

$\Delta g_{lin}(t) = \cos \Phi \Delta x(t) + \sin \Phi \Delta y(t)$ with Φ the direction of the overall gaze-shift vector.

The instantaneous vectorial gaze-displacement amplitude, $\Delta g_{vec}(t) = \sqrt{\Delta x(t)^2 + \Delta y(t)^2}$ is a different movement. This quantity still follows the curved trajectory because instantaneous amplitude and direction are time-dependent. The two only coincide at the start and end positions, or when saccades are absolutely straight. To avoid this possible confusion, I have now added this comment in the Methods.

L386: what is “gaze track-velocity”? Is it different from the vectorial gaze velocity? If not, then the “gaze track-velocity” should be replaced by “vectorial gaze velocity” throughout the manuscript.

Yes, it is different. See the previous comment. It is the (intended) velocity along the straight line.

L428 and L445: If “gaze-track velocity” is the same as vectorial gaze velocity, then use the latter wording?

See above.

L428: what means “best-recorded cells”?

This is explained in Methods (to which I now refer): I selected, for certain analyses, 20 cells that fulfilled three optimal recording criteria. If I did, I always showed the full population data as well. These ‘best’ cells are also highlighted in Table S1.

L434: the author should define what $P(r)$, P_0 and r mean.

Done.

L444: the word “convolved” should be inserted between “instantaneous” and “firing rate”.

Added.

DISCUSSION:

L459: the small values of epsilon (eye position sensitivity factor) in Table S1 does not seem consistent with the statement that “eye position strongly influenced the SC firing-rate profiles in all neurons”. Indeed, all epsilon values are smaller than 0.02 spikes / degree, i.e., 1 spike / 50 degrees.

Indeed, eye position did not systematically change the number of spikes in the bursts (this is quantified by the ϵ), but it DID have a very strong effect on the SC burst characteristics: burst duration, burst shape, and peak firing rate. A nice illustration of this is shown in the new Fig. 3C.

L463-465: This reviewer also thinks that the high firing rate of SC neurons increases the probability of observing synchronized action potentials within the population of bursting neurons. However, it is unclear why the correlation of the activity of single SC neurons with “the behavioral outcomes” leads the author to imagine “a tight synchronization of the activity patterns within the population”. The sentence should be rephrased.

The ‘synchronization’ in the population imposes the same firing-rate profiles on all cells in the population (as shown by Goossens & vOpstal, 2012). It is not just that all cells fire together, they do so with the same profile on a moment-to-moment basis. As I show with the shuffled histograms, the correlation with velocity largely disappears when we reshuffle the trials. Therefore, the detailed profiles matter, not just burst intensity per se.

L472: It is unclear why Fig. 9 is mentioned. Do the red data points correspond to the unit #sa1007 during which these movements were recorded?

I now refer to Figure 3, which is more appropriate.

L476: “unchanged” instead of “unaffected”?

Ok

L478: A correlation between the instantaneous firing rate and instantaneous gaze velocity does not imply a causal chain. The changes in firing rate must precede the change in eye velocity.

They do, by 20 ms! As indicated before, the correlations were calculated for the delayed firing rates. This can also be appreciated from the example shown in Fig. 10A,B.

L543-544: Contrary to the author's belief, Hepp et al. (1993) did not report that "large bilateral injections of muscimol in monkey demonstrated that the animal no longer generated any visual evoked saccades". Nothing was demonstrated in fact. Hepp et al. (1993) merely reported a reduction in the frequency and velocity of spontaneous rapid eye movements made spontaneously in the light. The amounts of reduction were not documented.

I cordially disagree with the reviewer. This is exactly what Hepp et al. reported and mentioned in their abstract as one of their major findings. The monkeys were tested in the light (I am co-author of that study) and did no longer respond with rapid visual-evoked saccades after the bilateral injection. They still generated fast phases of vestibular nystagmus, of which the torsional quick phases appeared unaffected. Horizontal and vertical fast phases were markedly slower. The only spontaneous eye movements that the monkey made were slow shifts of gaze, resembling 'saccades', but these were infrequent (about 10 per minute) and not goal directed.

L544-547: Likewise, neither Goossens and Van Opstal (2006) nor Peel et al. (2020) or Zhang et al. (2022) showed that small local reversible lesions led to "immediate and specific deficits in the metrics (endpoints away from the lesion) and kinematics (substantially slower, by more than 20%) of saccades". These references should be removed. The study of Goffart, Hafed & Krauzlis (2012) could be added because it documented correlated changes in the latency and peak velocity of contralesional saccades after the local injection of a small amount of muscimol in the SC (their Fig. 12A).

I now removed the entire section on the role of the SC from the Discussion, as these studies are now highlighted in the Introduction. There, I also incorporated the reference, as suggested.

FIGURE S3:

Is "track displacement" the same as vectorial displacement?

No, see above.

FIGURE S8:

The titles of y axes and the legend must be corrected. The plots A and C do not show gaze and head positions as a function of time but the amplitude of gaze and head displacements.

This is now changed.

METHODS:

It is unclear whether the deviations of the head relative to body sagittal plane were only horizontal.

Presumably, the values of horizontal and vertical orientation of the head were required to be within a range.

The monkey could move its head freely in all directions, but its torso was constrained within a jacket that was fixed to the chair.

L703: the author should indicate how the accuracy of gaze shifts was controlled. Presumably, the gaze direction was required to land within a window relative to the target location. What was its radius?

The window was typically 8 deg, which the monkey had to enter within 600 ms

Figure S1: scales for the gaze position and velocity should be added.

These traces were normalized in their amplitudes for illustrative purposes. The amplitude was adjusted to the spike scale of that figure.

REFERENCES MENTIONED IN THE COMMENTS:

in red: references incorporated in the revision.

Anderson, R. W., Keller, E. L., Gandhi, N. J., & Das, S. (1998). Two-dimensional saccade-related population activity in superior colliculus in monkey. Journal of Neurophysiology, 80(2), 798-817.

Choi, W. Y., & Guitton, D. (2009). Firing patterns in superior colliculus of head-unrestrained monkey during normal and perturbed gaze saccades reveal short-latency feedback and a sluggish rostral shift in activity.

Journal of Neuroscience, 29(22), 7166-7180.

Fuchs, A. F., Brettler, S., & Ling, L. (2010). Head-free gaze shifts provide further insights into the role of the medial cerebellum in the control of primate saccadic eye movements. *Journal of neurophysiology*, 103(4), 2158-2173.

Gandhi, N. J. (2012). Interactions between gaze-evoked blinks and gaze shifts in monkeys. *Experimental brain research*, 216, 321-339.

Goffart, L. (2019). Kinematics and the neurophysiological study of visually-guided eye movements. *Progress in Brain Research*, 249, 375-384.

Goffart, L., Bourrelly, C., & Quinton, J. C. (2018). Neurophysiology of visually guided eye movements: critical review and alternative viewpoint. *Journal of Neurophysiology*, 120(6), 3234-3245.

Goffart, L., Chen, L. L., & Sparks, D. L. (2004). Deficits in saccades and fixation during muscimol inactivation of the caudal fastigial nucleus in the rhesus monkey. *Journal of neurophysiology*, 92(6), 3351-3367.

Goffart, L., Hafed, Z. M., & Krauzlis, R. J. (2012). Visual fixation as equilibrium: evidence from superior colliculus inactivation. *Journal of Neuroscience*, 32(31), 10627-10636.

Hafed, Z. M., & Chen, C. Y. (2016). Sharper, stronger, faster upper visual field representation in primate superior colliculus. *Current Biology*, 26(13), 1647-1658.

Lee, C., Rohrer, W. H., & Sparks, D. L. (1988). Population coding of saccadic eye movements by neurons in the superior colliculus. *Nature*, 332(6162), 357-360.

Miyashita, N., & Hikosaka, O. (1996). Minimal synaptic delay in the saccadic output pathway of the superior colliculus studied in awake monkey. *Experimental brain research*, 112, 187-196.

Moschovakis, A. K., Gregoriou, G. G., & Savaki, H. E. (2001). Functional imaging of the primate superior colliculus during saccades to visual targets. *Nature neuroscience*, 4(10), 1026-1031.

Noda, H., Sugita, S., & Ikeda, Y. (1990). Afferent and efferent connections of the oculomotor region of the fastigial nucleus in the macaque monkey. *Journal of Comparative Neurology*, 302(2), 330-348.

Peel, T. R., Dash, S., Lomber, S. G., & Corneil, B. D. (2021). Frontal eye field inactivation alters the readout of superior colliculus activity for saccade generation in a task-dependent manner. *Journal of Computational Neuroscience*, 49, 229-249.

Phillips, J. O., Ling, L., Fuchs, A. F., Siebold, C., & Plorde, J. J. (1995). Rapid horizontal gaze movement in the monkey. *Journal of neurophysiology*, 73(4), 1632-1652.

Quinet, J., & Goffart, L. (2007). Head-unrestrained gaze shifts after muscimol injection in the caudal fastigial nucleus of the monkey. *Journal of neurophysiology*, 98(6), 3269-3283.

Quinet, J., & Goffart, L. (2009). Electrical microstimulation of the fastigial oculomotor region in the head-unrestrained monkey. *Journal of neurophysiology*, 102(1), 320-336.

Sparks, D. L. (1993). Are gaze shifts controlled by a 'moving hill' of activity in the superior colliculus? *Trends in neurosciences*, 16(6), 214-218.

Sparks, D. L., Lee, C., & Rohrer, W. H. (1990). Population coding of the direction, amplitude, and velocity of saccadic eye movements by neurons in the superior colliculus. In *Cold Spring Harbor symposia on quantitative biology* (Vol. 55, pp. 805-811). Cold Spring Harbor Laboratory Press.

Sparks, D. L., Holland, R., & Guthrie, B. L. (1976). Size and distribution of movement fields in the monkey superior colliculus. *Brain research*, 113(1), 21-34.

Van Gisbergen, J. A., Robinson, D. A., & Gielen, S. T. A. N. (1981). A quantitative analysis of generation of saccadic eye movements by burst neurons. *Journal of Neurophysiology*, 45(3), 417-442.

Zhang, T., Malevich, T., Baumann, M. P., & Hafed, Z. M. (2022). Superior colliculus saccade motor bursts do not dictate movement kinematics. *Communications Biology*, 5(1), 1222.

Reviewer #2 (Remarks to the Author):

Summary

Over the past decade, Dr. Van Opstal and colleagues have developed and tested a model for how the population of superior colliculus (SC) neurons generate saccadic eye movements. Here, using recordings of SC activity from macaque monkeys generating eye-head gaze shifts, Dr. Van Opstal extends this model to eye-head gaze shifts, and shows that many model predictions are borne out in the neurophysiological data. This is far from a trivial finding, given the many differences between the biomechanics of eye and head motion. The manuscript also includes a considerable sample of neural data collected throughout the extent of the SC, with the responses field of each neuron very well characterized throughout a range of initial positions. The dataset is all the more impressive given the head-unrestrained nature of the preparation. Overall, my opinion is that this

manuscript represents a valuable contribution to the literature.

As should be clear from the above, I am enthusiastic about the manuscript.

I greatly appreciate this encouraging statement.

I have a number of points of feedback for further consideration.

1. The author correctly points out a number of the nonlinearities downstream of the SC during eye-head gaze shifts. The author also alludes to the complexity of the head plant in places, but I think more needs to be emphasized here, particularly early on in the manuscript.

I fully agree, and I have now done so.

How the head converts a neural signal to force is itself quite a significant non-linearity, and the sense one gets is that the head is, to some degree, being treated like a big eye. Unlike the eye, the complexities of force generation at skeletal muscles moving an inertial system are substantial, being impacted by things such as initial position of the eye-in-head and head-on-body, as well as other drives to the head that may precede that associated with the gaze shift. The presence of such drives is apparent in the example shown in Fig S8, where the head movement precedes ipsi-gaze shift onset (as an aside, any thoughts on where that drive could be coming from?)

This is indeed an interesting question; I don't have a straight answer, other than it might be that it is driven by prelude activity from the SC, prior the actual gaze onset. I am currently toying with this idea, trying to set up a simulation model that is driven by the recordings from these 43 neurons, but it might be that the population is be too small for this purpose.

I appreciate that incorporating such nonlinearities would go beyond the goals of the current paper, but I think it is important to acknowledge this non-linearity.

I also explicitly included it now in the conceptual scheme of Fig. 1B, and see below.

Within the context of the current model, it strikes me as unlikely that the system can approximately compensate for this degree of non-linearity. The fact that the overall fits shown in the paper are so good may reflect aspects of the task, and/or that fact that the head is a comparatively smaller contributor to a gaze shift than the eye.

Both could be true. The task in this study was relatively simple: foveate a single peripheral visual target. The overall head movements in these gaze shifts varied with the gaze-shift amplitude and initial eye position. As Figure 2B shows, for gaze shifts of, say 60 deg, the head amplitudes were approximately half, i.e. 30 deg.

2. This line of thinking does lead me to wonder about the data presented in Figures 10 and 11, which show impressive trial-by-trial fits between the temporal profile of SC spikes and movement kinematics. Given my concerns on the comparative differences between the biomechanics of eye vs head motion, I wonder how much of this relationship is driven by similarities over the accelerating (or first half) of the gaze shift, when this is the main contributor to the gaze shift.

In fact, the accelerating phase of gaze shifts is relatively invariant, as the peak velocity is reached at a nearly fixed moment after gaze-shift onset... Most of the variability in velocity profiles is due to the deceleration phase (this can be appreciated also from the different example velocity profiles throughout the paper: Figure 3B, Figure 4C, Figure S5, Figure 10B, and Figure S9. The correlation analysis shows that if I shuffle the responses, most of the correlation is gone. The remaining correlation of about $r = 0.57$ is due to the stereotypical nature of single-peaked velocity profiles: they accelerate for about 20 ms and then decelerate. Yet, the high mode at $r=0.87$ could only emerge when the entire original velocity profile and firing-rate profile are compared.

I can imagine a variety of ways to address this possibility. One way would be to repeat these analyses for the first-half versus second-half of the gaze shift. Alternatively, one could split the analyses along the "ipsi, same, and contra" initial eye position, as done elsewhere. Finally, one could check if the quality of the fits decrease as the head's contribution increases (i.e., plot the correlations in Fig. 11 as a function of head contribution) Essentially, I am wondering if these fits would be worse for gaze shifts (or phases of gaze shifts) where the head is contributing more, presumably reflecting something about the non-linearities inherent to head motion. *I have now checked this interesting latter possibility, by plotting the correlation coefficients as function of the*

head-movement amplitude in each trial and included this result in the new Figure 11B (in the Discussion) and in the Supplemental Material as Fig. S11 for the population of cells.

The reviewer's intuition seems to be confirmed by the results, as I found a weak, but highly significant negative correlation: the larger the head movement amplitude, the lower the Firing-rate/Gaze velocity correlations. The gain for the cell shown in Fig. 11 was a -0.009 decrease of correlation per degree head movement. The variability was large too, as the coefficient of determination (r^2) was only about 0.30. Interestingly, this effect did not depend on initial eye position, as the data points fully overlap.

Therefore, I also tend to believe that it reflects the effect of nonlinearities in gaze control, due to the biomechanical complexities and inertia of the head.

3. The author refers throughout the paper to eye-position gain fields, reflecting the instructed position of the eye-in-head prior to the gaze shift. In the model shown in Fig. 12, this signal is important in customizing the dynamics of SC activity. If I understand the experimental paradigm correctly, every eye-in-head position requires an equal-and-opposite head-on-body position, hence eye-in-head position is not the only thing being manipulated. Head-on-body gain fields have been reported in the SC (Nagy and Corneil, J Neurophys 2010), hence the signal impinging on the SC in Fig 12 may be better thought of as a signal reflecting initial configuration composed of a combination of eye-in-head and head-on-body position.

This is indeed the case, and I have now made explicit reference to this point in both the Methods, and in the Discussion (referring to the works of both Walton et al., 2007; Nagy and Corneil, 2010), and I updated the Figure to include this possibility.

This issue may be one of nomenclature for the current manuscript, but I can imagine head-on-body signaling being important as well for particularly large gaze shifts where the head could theoretically be reaching its limits for excursions relative to the body.

Indeed, in the present experiments we are not able to dissociate whether the influence on kinematics and firing rates is caused by the change in initial eye position or of head position, or of some weighted combination of both. Future experiments will have to be performed by independently varying these two parameters and measure their effects. I now mention this explicitly in the open paragraph of the Discussion.

Reviewers' comments:

Reviewer #1 (Remarks to the Author):

Review made by Dr Laurent Goffart:

The author made substantial changes in his article. His answers to the comments of this reviewer were more or less satisfactory. Part of this reviewer's dissatisfaction comes from answers that were either incomplete or irrelevant (off topic). However, this study remains very important for several reasons:

- its conclusion contradicts the synthesis that this reviewer reached,
- the experimental work (unit recording in the monkey tested with the head unrestrained) was a difficult challenge,
- the collected data are extremely precious and
- this work illustrates very well the extraordinary advancement of knowledge that the community of primate neurophysiologists have achieved in order to understand how the brain networks enable to orient the gaze and head toward a visual target.

That being said and considering the following facts, i.e.,

1) that a model factoring eye position, muscle tension and its first derivative accounts for the firing rate of motor neurons, better than a model factoring eye position and its first and second derivatives (velocity and acceleration) (Davis-Lopez de Carrizosa et al., 2011),

2) that the resemblance between the firing rate of premotor burst neurons and the instantaneous velocity of the saccade disappears when the spikes are not convolved with a Gaussian kernel (see Figs. 2 and 3 in Hu et al. (2007), Fig. 3C in Sparks & Hu (2006) and Fig. 4 in van Gisbergen et al. (1981)),

3) that the activity of premotor neurons is under the influence of neurons located in the left and right caudal fastigial nuclei (see Fig. 9 of Bourrelly et al. 2021),

4) that the cFN is not part of the pathway that controls head movement (Fuchs et al. 2010; Quinet & Goffart 2017, 2009),

5) that the connection between the Superior Colliculus and the premotor burst neurons is not monosynaptic (Keller et al. 2000),

6) that the firing rate of dSC cells is not related to the velocity of saccades toward a moving visual target (see Fig. 11 in Goffart et al. 2017),

7) and that the correlation between the firing rate of single neurons and eye kinematics must be interpreted with the greatest caution, especially for those neurons located several synapses upstream from the motor neurons (Goffart et al. 2018),

it is hard not being skeptical after reading:

a) that "single-trial and single-unit firing dynamics at a central neural stage correlate so well with the instantaneous motor output of a highly complex and nonlinear synergistic system (comprising the multiple-degrees of freedom oculomotor, head-motor, and vestibular systems ...)" (L536-539),

b) that "the population of recruited saccade-related burst neurons in the SC specifies the detailed kinematics (trajectories and velocity profiles) of saccadic eye-head gaze shifts (abstract L8-9) and

c) that "the same principles may apply to a wide range of saccadic eye-head gaze shifts with strongly varying kinematics" (abstract L10-11).

When the author declares having shown that the population of recruited saccade-related burst neurons in the SC "specifies the detailed kinematics (trajectories and velocity profiles)" of saccadic eye-head gaze shifts (abstract L8-9), the readers should be warned that:

I) this conclusion is based upon the analysis of the correlation between a particular analysis of neuronal activity and the instantaneous velocity of gaze shifts. The methods that he used for quantifying this activity are popular (many electrophysiologists used them). However not all neurophysiologists did. According to this reviewer, these methods create artefacts that are foreign to the physiological reality. Indeed, by convolving each action potential with a Gaussian, the author transforms the sequence of discrete events into a continuous function (the so-called spike density

function), which permits to study its correlation with another continuous function like the velocity profile of gaze orienting movement. However, this method has several drawbacks about which the readers should be warned. Firstly, the transformation of the sequence of spikes into a continuous function feeds the illusion that a kinematic parameter is encoded by the activity of neurons and transmitted to post-synaptic neurons. In fact, the post-synaptic neurons integrate input from multiple neurons located in several regions. If the time course of SC activity looks like the velocity of eye (or gaze) saccade, how do we explain the fact that the firing rate of burst neurons does not (see their quasi-constant interspike intervals in Hu et al. 2007). Secondly, the cumulative Gaussian curves create an event (the peak firing rate) that may not have any physiological counterpart if its value and its timing differ from the actual shortest interspike interval and its timing. Secondly, the symmetry of the Gaussian curve creates a situation in which each convolved spike exerts an influence backward in time, shifting the putative peak firing rate toward the gaze saccade onset. As explained in the first review (sixth major concern and comments on the previous Fig. 9), if a causal relation exists between the firing rate of neurons and the gaze velocity profile, then spikes emitted AFTER the time of gaze peak velocity cannot influence the gaze peak velocity value. If they do, it is a mathematical artefact resulting from convolving each spike with a symmetric curve (Gaussian). Therefore, this reviewer invites the author to warn the readers about this concern and to document both the time of peak firing rate (estimated by the spike density function) and the time of shortest interspike interval (ordinate), both as a function of the time of peak gaze velocity (abscissa).

II) the conclusion that the population of recruited saccade-related burst neurons in the SC specifies the detailed kinematics of gaze shifts cannot be supported by the reported data insofar as the physical trajectory of gaze direction (as schematized in Fig. S3 by $G1(t)$ or $G2(t)$) is projected onto the imaginary line segment (DELTA θ) that connects the starting gaze position to the final gaze position. Because of this projection, it is not the detailed kinematics that is described. Let's consider the trajectory $G2(t)$ in Fig. S3, it is below the imaginary line segment DELTA θ . We can imagine its symmetrical image $G2SYM(t)$ with respect to the same line segment DELTA θ . The projection of $G2(t)$ onto DELTA θ is identical to the projection of $G2SYM(t)$ in spite of different trajectories, in spite of different ratios between the horizontal and vertical velocities. Yet, the firing rate of SC neurons can still support these two different trajectories ($G2(t)$ and $G2SYM(t)$) if the detailed distribution of activity is different in the SC, skewed toward the medial border of the SC during $G2(t)$, skewed toward the lateral border of the SC during $G2(t)$. This a point that the author could have explained to the readers. However, the concern remains that IMAGINARY line segment (DELTA $\theta_2(t_0)$) is correlated with the ARTIFICIAL estimate of neurons' firing rate (density function). This reviewer does not want to prevent the author from analyzing his data the way he likes; she/he merely requests that the author warns the readers about its associated limitations and about the theoretical estimated and empirical data.

Despite these two major concerns, this reviewer is inclined to agree with the idea that the population of active neurons in SC steers gaze toward the target location and that the saccade curvature results from fluctuations in the firing rate of recruited neurons. However, as mentioned in her/his first review, the development of this idea largely benefited from electrical microstimulation studies in the head-restrained and head-unrestrained monkey (Stanford et al. 1996; Freedman et al. 1996). These studies showed that electrically-evoked gaze shifts stop as soon as the microstimulation stops and that a minimum number of electrical pulses is required to evoke the saccade vector (amplitude and direction) specific to each site of stimulation. They also showed that reducing the stimulation frequency (increasing the duration of the interval between two electrical pulses) evoked slower saccades, the amplitude of which was maintained if the stimulation was prolonged (see Figures 5, 8 and 12 of Stanford et al. 1996 and Figure 3 and 4 of Freedman et al. 1996). Although this reviewer appreciates very much the author's efforts to corroborate, with electrophysiological recordings in the head-unrestrained monkey, the hypothesis that Dr Sparks and his colleagues defended 27 years ago, she/he does not share the same surprise as the author wants to transmit to the readers (abstract L11). By neither reminding their results nor by explaining how his recording results are consistent with them, the author does not do a fair justice to the seminal studies of Dr Sparks' group. Likewise, it is also unclear why the author does not discuss his results with regard to the suggestion that the locus of

activity in the Superior Colliculus would encode target eccentricity (relative to gaze direction) instead of gaze saccade amplitude (Bergeron et al. 2002; 2003). Finally, it is important reminding the readers that the firing rate of neurons also depends upon the animal's alertness and motivation, both of which do not vary from one trial to the next. This massive and sluggish influence likely explains the loss of trial-by-trial correlation between the firing rate and gaze velocity when the velocity profiles are shuffled (Fig. 10).

Still concerning the relation to previous studies, the report of the results obtained by Peel et al. (2021) is biased. During FEF inactivation, the number of action potentials emitted by SC neurons is neither increased, nor slight, nor restricted to memory-guided saccades (L141-143). Their figure 1b shows an impressive impairment of delayed-saccades: the total number of spikes during FEF cooling is reduced almost by half (N=13) in comparison to the number of spikes recorded during control saccades (N=22). The horizontal and vertical amplitudes of saccades were barely changed whereas their velocity was severely reduced. The longer interspike intervals associated with the slowing of saccades is consistent with the influence of SC activity upon saccade velocity. Sparks, Lee & Rohrer (1990) reported a similar dramatic decline in velocity with no change in saccade direction and amplitude (see their Figure 3). Although Sparks and colleagues could not record the firing rate of neurons, the observations of Hanes et al. (2005) that the response fields of neurons adjacent to a small collicular lesion do not expand leads to infer that the lidocaine injection also reduced the number of active neurons. Thus, the suppression of neurons and the lower firing rate of peripheral neurons (peripheral with respect to the center of the normally active population of neurons) can account for the reduced velocity. It is also worth reminding that Soetedjo et al. (2002) showed that after muscimol injection in the pontine reticular formation, the number of spikes emitted by SC neurons remained the same despite prolonged saccade duration (as expected by the author). However, they provided a different explanation (feedback to the SC). Note that projections from the nucleus prepositus hypoglossi to the SC are consistent with such feedback. It is unclear why the author ignores this anatomical fact (Hartwich-Young et al. 1990). All these observations constitute precious empirical clues that are useful to identify the physiological parameters that can vary from one trial to the next, i.e., the duration of activity, the firing rate, the number of spikes and the number of active neurons.

From the viewpoint of this reviewer, these neurophysiological results obtained sometimes during tedious experiments in the awake and trained monkey do not deserve less emphasis than theoretical speculations such as "additive" (L354, 356) and "multiplicative" (L320, 347, 357) noises in the brain. The neurophysiological meaning of these notions is unclear and possibly misleading. Unit recordings show that the firing rate of any SC neuron is rarely the same from one trial to the next in spite of quasi-identical initial conditions. In his article, the author prefers reporting the mismatch between his equations and the real data rather than reporting the actual data. Neither the variability of interspike intervals (during the burst) nor the variability of cumulative spikes counts are documented. By not showing the staircase of segments composing the cumulative spike numbers, the author prevents the readers from realizing that the emission of action potentials by SC neurons is neither a periodic process nor a continuous one. Freedman and Sparks (1997) showed that the interspike intervals are neither constant nor reproducible from one trial to the next (see their Fig. 5). By barely reporting the measured data, he prevents the readers not only from understanding what is variable but also comparing them with observations reported by others, seeing consistency and discrepancies.

Still concerning the report of real data, this reviewer noticed that the values of gaze amplitude and direction in Table S1 do not match the values plotted in Fig. S2. For instance, according to Table S1, $R_0=98.2$ and $\text{THETA}_0=-68.9$ for the neuron Sa0907. This neuron is not visible in Fig. S2. Reciprocally, Fig. S2 shows two blue circles with R values larger than 80 whereas Table S1 reports only one R value larger than 80 ($R_0=94.6$, neuron Pc2406).

Likewise, contrary to the author's claim, the data in Fig. 6A do not show "the variability of the burst-related spike count as [a] function of the expected total number of spikes in the burst". In the legend of Fig. 6A, one reads "the error in the predicted number of spikes" (L938). If this definition is correct,

then one is led to infer that the plotted error (ordinate axis) corresponds to the difference between the number of spikes predicted by equation 9 and the number of spikes that were actually measured. In other words, it is an index of the goodness of the model (the equation) to capture the data (measured values). Consequently, plotting the difference between the prediction of a model and the data as a function of the predicted number of spikes does not describe a so-called "signal dependent noise" (L305) if by "signal dependent noise", the author refers to something related to the firing rate of neurons. The difference between the number of spikes predicted by an equation and the number of spikes emitted by a biological neuron has no physiological meaning. Plotting this difference as a function of a predicted value is even more speculative. This other example illustrates the necessity to warn the readers about differences between empirical data and theoretical notions.

Regarding the increase of the standard deviation with the mean number of spikes (Fig. 6B), it would be useful to show this relation for the neuron sa0107 also (Fig. 5B). Indeed, we can estimate (although with some difficulty) the various amplitudes of gaze shifts that were recorded with this neuron (note that a plot or an inset showing the variability of amplitude and direction of gaze shifts would be helpful in Fig. 5). Note Fig. 5B shows a counter example to the increase of the standard deviation with the mean number of spikes. Considering a number of spikes close to the maximum ($N > 25$), the values of straight-line gaze displacement ranged from approximately 25 to 48 degrees. Curiously, considering a smaller number of spikes ($N = 14$), the values of straight-line gaze displacement ranged from approximately 15 to 66 degrees. In other words, the variability is not larger with more numerous spikes. The description of Fig. 5B should be improved because it contains a lot of information. On the one hand, we know that the measured number of spikes depends upon the location of the visual target with respect to the center of the response field (RF). It is larger when the visual stimulus falls in the center of the RF (leading to the maximum saccade-related burst) than when the visual stimulus falls closer to the boundary of the RF (leading to a less vigorous burst). On the other hand, we also know that the size of the RF increases with the target eccentricity, and thus with the location of collicular activity along the rostrocaudal axis (see Fig. 2 of Taouali et al. 2015 and Fig. 8 of Munoz & Wurtz 1995). Therefore, we wonder whether the relation between the mean measured number of spikes and the standard deviation of the measured number of spikes is not the consequence of presenting the target at various locations in the response field of neurons. Finally, it is worth reminding the readers that the firing rate also depends upon the alertness and the motivation of the animal. Keeping these physiological facts in mind, the readers can eventually understand, or not, the meaning of "multiplicative noise in the cell's activity".

The statement of "excellent agreement between data and model" (L366) illustrates the theoretical inclination of the author. Unfortunately, such a statement is qualitative and rushed. After drawing the $y=x$ line, one sees that the majority of data points are situated above the diagonal. In other words, the predicted slopes overestimate the measured slope. Table S1 tells us that for cell Sa0508, when the gaze amplitude is 11 deg, the predicted number of spikes is 23, making a ratio of 2.09 spikes per degree. The turquoise line in Fig. 8A shows that for a predicted slope of a slope of 2.09, the measured slope is 1.82 spikes per degree. In other words, for a gaze amplitude of 11 deg, the average measured number of spikes was $11 \times 1.82 = 20$ spikes, i.e., 3 spikes less than the predicted number of spikes ($N_0 = 23$). Equation (9) overestimated the measured number of spikes by 15% for a gaze amplitude of 11 deg. Unfortunately, from looking at Fig. 8A, it is impossible to see neither the number of spikes measured during a gaze shift of 44 deg amplitude, nor the predicted number of spikes. Therefore, to facilitate the readers' understanding, before plotting the predicted number of spikes as a function of the measured number (as in Fig. 8A), the author should plot these two numbers (measured and predicted numbers of spikes) as a function of gaze amplitude for iso-directional gaze shifts ($\text{THETA} = \text{THETA}_0 \pm$ e.g. 5 deg or 10 deg). The plot in Fig. 8A could then be presented using open and filled symbols to distinguish iso-directional gaze shifts from others. Then, the slopes and y-intercepts of the relations between predicted and measured spikes could be documented for all cells so that the readers can see whether equation (9) systematically overestimated the measured number of spikes for each cells.

Below, the author will find more detailed comments on the figures and on the text. They are not

exhaustive because reading this article was rather time-consuming and difficult for this reviewer.

COMMENTS ON THE OTHER FIGURES:

Figure 3:

panel A: Looking at the values for the head at gaze onset, this reviewer suspects that the title of the ordinate axis is G/H Displacement instead of G/H Position. It is also very rare that gaze direction is strictly equal to zero. The legend should be corrected accordingly (L900 and 901).

Panel B: one red curve (out of eight) is shifted toward negative values. It seems that the onset of the corresponding gaze shift was not properly detected (its velocity is larger than 250 deg/s at gaze onset). The author should check the impact of this mistake on the red curves illustrating the spike density function and the mean cumulative spike count.

Figure 4:

Panel C: The blue curves are barely visible. The author could use either a lighter blue (e.g., turquoise) or change the color of the background from black to grey. A scale for the color codes (number of spikes) would be welcome or an indication of the number of spikes corresponding to the thin white contour lines. What is Fopt? The same comments hold for Fig. S5.

Figure 5:

Legend:

L929: NSTMF instead of NMFST, in accordance with equation 9 (L771)

The author name this plot "Dynamic movement field" whereas the term "phase trajectory" is used in several place in the text (L289, L293, L364, L378, L523, L962). If the two terminologies denote the same plot, the author should use only one to remove ambiguities. If they do not denote the same plot, then the author add an explanatory note.

Panel B: Ordinate axis: Which "Cumulative Spike Count" is plotted? The predicted spike count (as in panel C) or the measured spike count? It would be useful to make it clear in the figure.

Panel C: It may be preferable to plot the measured (ordinate) as a function of the predicted cumulative spike count (abscissa) as in Fig. 4C and Fig. S5. Maybe not.

Figure 6:

Title of y-axis: Model's prediction error

Figure 8:

Panel A: The scales of x and y-axes do not facilitate the calculations that some readers would like to perform. A step of 0.5 is preferable to a step of 0.52. It is fine to use 3.0 as maximum values.

Drawing the diagonal $y=x$ will enable the readers to see that the predicted slope overestimates the measured slope (majority of data points above the diagonal). The turquoise line likely illustrates the slope of the scatter. Unfortunately, it is not defined in the legend. Its slope, y-intercept and the number of datapoints should be documented in the legend.

Panels A and B: The cell identification number (Sa0508) should be inserted in order to facilitate the distinction between the plots corresponding to one cell and those corresponding to several cells.

Figure 9:

Panel A: As requested in the previous review, it is preferable, for conventional reasons, to use the word "amplitude" instead of "shift".

Panel B: An inset should be added to show the delay between the time of peak firing rate and the time of peak gaze velocity. It is important for the readers to realize that in order to influence the peak velocity, the peak firing rate leads and not lags time to peak firing rate (see major comment).

Panel C: A graph plotting the time to peak gaze velocity as a function of time to peak firing rate should be added to assure that the peak firing rate led (and not lagged) time to peak firing rate (see major comment).

Table 1

The names of parameters do not match between table 1 and L778.

Legend, second line: FMFST=?

Fig. S5:

Legend: last sentence: "faster" instead of "fastest"?

MORE DETAILED COMMENTS:

L30-35: It would be useful to warn the readers that gaze shifts may also consist in multiple saccades (e.g., Bergeron & Guitton 2002; Anastasopoulos et al. 2015). The readers should also be warned that very large gaze shifts performed within a single step may result from experimental constraints (reinforcement learning) if the monkeys are required to perform their primary gaze saccades so that gaze lands within a (relatively) small window for obtaining a reward.

L 48-49: The references Quinet & Goffart (2007, 2009) are inappropriate to raise an objection against the author's statement. Indeed, the results of both studies may indicate that the activity of neurons in the caudal fastigial nuclei contributes to coupling the eye and head movements insofar as inactivating one side impairs the eye-head coupling. Together with other anatomical and electrophysiological data (see their discussion), they indicate that eye-head gaze shifts are under cerebellar control and that the fastigial exerts its influence primarily upon the oculomotor network. This reviewer invited the author to consider these works in order to warn the readers that the kinematics of eye and head movements is also influenced by the activity of neurons in the caudal fastigial nucleus and that a gaze feedback control is controversial.

L140: "Peel et al. (2021)" instead of "Peel et al. (2020)".

L143: "smaller" instead of "slightly higher".

L147: The word "however" should be removed because the following sentence ("the variability ... difficult to detect") is consistent with the previous sentence ("Further, Zhang et al. (2022) ... for all conditions"). Or the argument is not clear.

L153: This reviewer fears that the suggestion that "the SC has no central role in saccade control (Daye et al. 2014)" (L153) misleads the readers. She/invites the author to either remove it or insert "Moreover" before "immediate deficits in saccade metrics ..." (L156) and add "immediately" between "observed" and "after acute reversible microlesions" (L157).

L162-166: It would be useful to inform the young readers that this crucial role of both SC and FEF was suggested by the lesion study performed by Schiller, True & Conway (1980).

L196: It would be useful to add a reference showing that the rotation axes of the head do not intersect in a fixed point.

L216: "during experiment Sa1007" instead of "of experiment Sa1007"

L220: "when the eye looked ipsilateral to the target" is clumsy and potentially confusing because the target is being foveated. Does the author mean "when the eye was deviated toward the side ipsilateral to the direction of the imminent gaze shift"?

L223: "with ipsilateral eye deviations" instead of "with the eye ipsilateral"?

L224: It would be neat to document the percentages of prolongation (for the duration) and reduction (for the peak velocity).

L234: "Figure 3 shows a selection of gaze shifts recorded during this experiment and initiated from the three initial eye positions" instead of "Figure 3 shows a selection of gaze shifts from this experiment from the three initial eye positions"

L237-248: The author should check whether the statements about the peak firing rate and its timing remain true after modifying Fig. 3. L249: "ipsi-condition (blue)" instead of "contra-condition (blue)"

L244-253: The author should warn the reader that the interspike intervals vary considerably across the trials and that the mean cumulative spike counts masks the variable firing rate of SC neurons, which contrasts with the clock-like firing rate of premotor burst neurons (Hu et al. 2007).

L249: "the timing of its peak" instead of "its peak location"

L258: "all action potentials" instead of "all neural responses"

L264-265: Choi & Guitton (2009) showed that the population the burst does not always ends before gaze end (see middle row of Figure 6C). Therefore, this reviewer suggests replacing "as reported in earlier studies (Freedman & Sparks, 1997; Choi and Guitton, 2009)" by "as reported by Freedman & Sparks (1997), but not always (see e.g., Choi & Guitton 2009)".

L268: "toward the center of the movement field" instead of "into the center of the movement field".

L268: "as well as" instead of "as well:"

L271: "cumulative spike count (CSC) instead of "CSC" because the abbreviation has not been previously defined.

L271-272: The author remarks that "the CSC trajectories nearly fully overlap along the $y=x$ diagonal, despite the considerable variation in gaze-shift kinematics". The overlap indicates a good match between the measured and the predicted cumulative spike counts". Note that the methods used to measure the number of spikes (Fig. 5) are not documented.

L275: "which is indeed seen in the CSC trajectories": an arrow would be useful to indicate at which location the reader must look to see.

L274: $N_{opt}=30$ whereas $F_{opt}=25$ in Fig. 4C. What is F_{opt} ? What is N_{opt} ?

L291: Fig. 5B shows that for a given value of measured total number of spikes (e.g., $N=16$), the straight-line gaze displacement amplitude can range from 19 deg to 62 deg? The author should explain why a large

L301: "predicted" instead of "expected"

L302: "ranged from 16 to 24 spikes" instead of "varied between 16 and 24 spikes"

L303: "the mismatch between the predicted and the measured number of spikes" instead of "this variability". By using the word "variability", the author makes a semantic drift which leads the readers to confound the variable firing rate of neurons visible across the trials (as illustrated in Fig. 3 and Fig. 4) with the ability of an equation to fully account for the measured values.

L303: +/- 20% ? How was this calculated ? Is it the average value?

L307-312: This paragraph is unclear and should be rephrased.

L309-310: "a predicted number of spikes is calculated for all saccade vectors that lie ..." would be preferable instead of "a particular expected number of spikes is predicted for all saccade vectors that lie ..."?

L318: "predicted" instead of "expected"

L317-318: See major comment

L322: "slope=0.141" instead of "Cv=0.141"

L332: Suggested change: "invariant to changes in starting eye positions" instead of "invariant to changes in starting eye positions"

L344: For the sake of simplicity, the sentence "Figure 7C provides histograms for the coefficients of variation (slopes of the relationship), Fig. 7D shows the distribution of correlations" can be replaced by "Figure 7C provides histograms for the slopes of the relationship and Fig. 7D the distribution of correlation coefficients between the average number of spikes and standard deviation.

L346: The wording "the neuronal variability" is vague. The author meant "the variability in the number of spikes during the gaze movement related burst".

L347: The author should explain the physiological meaning of "multiplicative noise".

L349-351: For consistency, the sentence "The median value for the population, $CV = 0.24$, is comparable to the mean $CV = 0.28 \pm 0.16$ reported by Goossens and Van Opstal (2012) for head-restrained eye saccades from >100 neurons across the SC motor map." Could be replaced by "The median slope value for the population (0.24) is comparable to the mean slope = 0.28 ± 0.16 reported by Goossens and Van Opstal (2012) for >100 neurons across the SC motor map during head-restrained eye saccades."

L351-352: The sentence "The low values of CV ($\ll 1.0$) 352 indicate that SC cells are typically characterized as being low variance" should be rephrased because it is unclear. It sounds like a tautological statement.

L358-359: The sentence "for a typical maximum of 25 spikes, the standard deviation is $0.24 \times 25 = \pm 6$ spikes (i.e., ranging from 19-31 spikes)" is unclear and seems to be wrong. The author estimates the minimum value to $25-6=19$ spikes and the maximum value to $25+6=31$ spikes, but the interval mean +/- one standard deviation does not contain 100 % but 68 % of values. In fact, 99 % of values are contained in the interval mean +/- 3 standard deviations.

L364: Could "(as in Fig. 5B)" be added after "phase trajectories"? If it cannot, then it is unclear what the measured slope is.

L368-379: This paragraph is unclear and should be rephrased. This reviewer understands that Fig. 8B

plots the difference between measured cumulative number of spikes and the predicted cumulative number of spikes. Contrary to the predicted cumulative number of spikes, the measured cumulative number of spikes is NOT a line, i.e., a continuous function. The cumulative number of spikes is a step function (like a staircase) with irregular steps (see for instance Sparks 1976, 1978, Sparks & Porter 1983; Moschovakis et al. 1988; Freedman & Sparks 1997; Goffart et al. 2017). It is important to warn the young and unwary readers that the interspike interval is neither constant nor reproducible from one trial to the next. Therefore, this reviewer requests a figure that shows the difference between the predicted and the measured cumulative number of spikes by plotting 1) both of them as a function of time from saccade onset to gaze end and 2) the difference between the measured interspike intervals and the predicted interspike intervals. See also major comment.

L402-409: See major comment

L471: "Kolmogorov" instead of "Kolomogorov"

REFERENCES:

- Anastasopoulos, D., Naushahi, J., Sklavos, S., & Bronstein, A. M. (2015). Fast gaze reorientations by combined movements of the eye, head, trunk and lower extremities. *Experimental brain research*, 233, 1639-1650.
- Bergeron, A., & Guitton, D. (2002). In multiple-step gaze shifts: omnipause (OPNs) and collicular fixation neurons encode gaze position error; OPNs gate saccades. *Journal of neurophysiology*, 88(4), 1726-1742.
- Bergeron, A., Matsuo, S., & Guitton, D. (2003). Superior colliculus encodes distance to target, not saccade amplitude, in multi-step gaze shifts. *Nature neuroscience*, 6(4), 404-413.
- Bourrelly, C., Quinet, J., & Goffart, L. (2021). Bilateral control of interceptive saccades: evidence from the ipsipulsion of vertical saccades after caudal fastigial inactivation. *Journal of Neurophysiology*, 125(6), 2068-2083.
- de Carrizosa, M. A. D. L., Morado-Díaz, C. J., Miller, J. M., de la Cruz, R. R., & Pastor, Á. M. (2011). Dual encoding of muscle tension and eye position by abducens motoneurons. *Journal of Neuroscience*, 31(6), 2271-2279.
- Freedman, E. G., & Sparks, D. L. (1997). Eye-head coordination during head-unrestrained gaze shifts in rhesus monkeys. *Journal of neurophysiology*, 77(5), 2328-2348.
- Freedman, E. G., & Sparks, D. L. (1997). Activity of cells in the deeper layers of the superior colliculus of the rhesus monkey: evidence for a gaze displacement command. *Journal of neurophysiology*, 78(3), 1669-1690.
- Fuchs, A. F., Brettler, S., & Ling, L. (2010). Head-free gaze shifts provide further insights into the role of the medial cerebellum in the control of primate saccadic eye movements. *Journal of neurophysiology*, 103(4), 2158-2173.
- Goffart, L., Bourrelly, C., & Quinton, J. C. (2018). Neurophysiology of visually guided eye movements: critical review and alternative viewpoint. *Journal of Neurophysiology*, 120(6), 3234-3245.
- Goffart, L., Cicala, A. L., & Gandhi, N. J. (2017). The superior colliculus and the steering of saccades toward a moving visual target. *Journal of Neurophysiology*, 118(5), 2890-2901.
- Hanes, D. P., Smith, M. K., Optican, L. M., & Wurtz, R. H. (2005). Recovery of saccadic dysmetria following localized lesions in monkey superior colliculus. *Experimental Brain Research*, 160, 312-325.
- Hartwich-Young, R., Nelson, J. S., & Sparks, D. L. (1990). The perihypoglossal projection to the superior colliculus in the rhesus monkey. *Visual neuroscience*, 4(1), 29-42.
- Hu, X., Jiang, H., Gu, C., Li, C., & Sparks, D. L. (2007). Reliability of oculomotor command signals carried by individual neurons. *Proceedings of the National Academy of Sciences*, 104(19), 8137-8142.
- Keller, E. L., McPeck, R. M., & Salz, T. (2000). Evidence against direct connections to PPRF EBNs from SC in the monkey. *Journal of Neurophysiology*, 84(3), 1303-1313.
- Moschovakis, A. K., Karabelas, A. B., & Highstein, S. M. (1988). Structure-function relationships in the primate superior colliculus. II. Morphological identity of presaccadic neurons. *Journal of Neurophysiology*, 60(1), 263-302.
- Munoz, D. P., & Wurtz, R. H. (1995). Saccade-related activity in monkey superior colliculus. I. Characteristics of burst and buildup cells. *Journal of neurophysiology*, 73(6), 2313-2333.
- Quinet, J., & Goffart, L. (2007). Head-unrestrained gaze shifts after muscimol injection in the caudal

fastigial nucleus of the monkey. *Journal of neurophysiology*, 98(6), 3269-3283.

Quinet, J., & Goffart, L. (2009). Electrical microstimulation of the fastigial oculomotor region in the head-unrestrained monkey. *Journal of neurophysiology*, 102(1), 320-336.

Schiller, P. H., True, S. D., & Conway, J. L. (1980). Deficits in eye movements following frontal eye-field and superior colliculus ablations. *Journal of neurophysiology*, 44(6), 1175-1189.

Soetedjo, R., Kaneko, C. R., & Fuchs, A. F. (2002). Evidence that the superior colliculus participates in the feedback control of saccadic eye movements. *Journal of neurophysiology*, 87(2), 679-695.

Sparks, D. L. (1978). Functional properties of neurons in the monkey superior colliculus: coupling of neuronal activity and saccade onset. *Brain research*, 156(1), 1-16.

Sparks, D. L., Holland, R., & Guthrie, B. L. (1976). Size and distribution of movement fields in the monkey superior colliculus. *Brain research*, 113(1), 21-34.

Sparks, D. L., & Hu, X. (2006, January). Saccade initiation and the reliability of motor signals involved in the generation of saccadic eye movements. In *Percept, Decision, Action: Bridging the Gaps: Novartis Foundation Symposium 270* (pp. 75-91). Chichester, UK: John Wiley & Sons, Ltd.

Sparks, D. L., Lee, C., & Rohrer, W. H. (1990). Population coding of the direction, amplitude, and velocity of saccadic eye movements by neurons in the superior colliculus. In *Cold Spring Harbor symposia on quantitative biology* (Vol. 55, pp. 805-811). Cold Spring Harbor Laboratory Press.

Taouali, W., Goffart, L., Alexandre, F., & Rougier, N. P. (2015). A parsimonious computational model of visual target position encoding in the superior colliculus. *Biological cybernetics*, 109, 549-559.

Van Gisbergen, J. A., Robinson, D. A., & Gielen, S. T. A. N. (1981). A quantitative analysis of generation of saccadic eye movements by burst neurons. *Journal of Neurophysiology*, 45(3), 417-442.

Reviewer #2 (Remarks to the Author):

I have read Dr. Van Opstal's response to reviews and the revised manuscript very closely, and I thank him for his considered responses. He has address all of my points thoroughly in his response. The only minor modification to the manuscript that I would suggest, based on my previous reviews, is in lines 194-195 (referencing the clean version), in regards to the new text on the non-linearity of head biomechanics. As written, it sounds like the non-linearities arise solely from the comparative rotational axes. In reality, the head's inertia and the complexity of force development in muscles that can be dynamically loaded (e.g., the force velocity curve) are likely dominant factors in the nonlinearity. My recommendation is to include such factors explicitly in the text near 194-195.

Reviewer 1:

The author made substantial changes in his article. His answers to the comments of this reviewer were more or less satisfactory. Part of this reviewer's dissatisfaction comes from answers that were either incomplete or irrelevant (off topic). However, this study remains very important for several reasons:

- its conclusion contradicts the synthesis that this reviewer reached,
- the experimental work (unit recording in the monkey tested with the head unrestrained) was a difficult challenge,
- the collected data are extremely precious and
- this work illustrates very well the extraordinary advancement of knowledge that the community of primate neurophysiologists have achieved in order to understand how the brain networks enable to orient the gaze and head toward a visual target.

I thank the reviewer again for the substantial amount of work put into this review. I will do my utmost best to be complete in my revisions and responses and apologize for the (seemingly) irrelevant remarks in my previous rebuttal. To avoid making more of these irrelevant responses, I indicated sections in the review that do not seem to require a response (as they do not specifically appear to call for an action, or an answer) in blue.

That being said and considering the following facts, i.e.,

1) that a model factoring eye position, muscle tension and its first derivative accounts for the firing rate of motor neurons, better than a model factoring eye position and its first and second derivatives (velocity and acceleration) (Davis-Lopez de Carrizosa et al., 2011),

2) that the resemblance between the firing rate of premotor burst neurons and the instantaneous velocity of the saccade disappears when the spikes are not convolved with a Gaussian kernel (see Figs. 2 and 3 in Hu et al. (2007), Fig. 3C in Sparks & Hu (2006) and Fig. 4 in van Gisbergen et al. (1981)),

3) that the activity of premotor neurons is under the influence of neurons located in the left and right caudal fastigial nuclei (see Fig. 9 of Bourrelly et al. 2021),

4) that the cFN is not part of the pathway that controls head movement (Fuchs et al. 2010; Quinet & Goffart 2017, 2009),

5) that the connection between the Superior Colliculus and the premotor burst neurons is not monosynaptic (Keller et al. 2000),

6) that the firing rate of dSC cells is not related to the velocity of saccades toward a moving visual target (see Fig. 11 in Goffart et al. 2017),

7) and that the correlation between the firing rate of single neurons and eye kinematics must be interpreted with the greatest caution, especially for those neurons located several synapses upstream from the motor neurons (Goffart et al. 2018),

The reviewer raises a number of valid points that all relate to brainstem-cerebellar properties of neural activity (except point 6). I would like to stress once more that my study does not test models of the gaze-control system that include, or focus on, parts of brainstem-cerebellar circuitries. My SC recordings are related and quantified exclusively with respect to the output of the system, being the gaze-movement trajectory, and I do not pretend to have a model that let the SC spikes generate these directly through a model circuit, as was the case for the simple eye-movement study of Goossens & vO in 2006. Instead, I express my surprise that despite all the complexities that the reviewer correctly identifies (and that, partly, I schematically also included in Fig. 1B as a background) the SC neural activity appears to relate quite well to the motor output.

it is hard not being skeptical after reading:

- a) that "single-trial and single-unit firing dynamics at a central neural stage correlate so well with the instantaneous motor output of a highly complex and nonlinear synergistic system (comprising the multiple-degrees of freedom oculomotor, head-motor, and vestibular systems ...)" (L536-539),
- b) that "the population of recruited saccade-related burst neurons in the SC specifies the detailed kinematics (trajectories and velocity profiles) of saccadic eye-head gaze shifts (abstract L8-9) and
- c) that "the same principles may apply to a wide range of saccadic eye-head gaze shifts with strongly varying kinematics" (abstract L10-11).

I have now inserted words like 'may' and 'could' in the abstract.

When the author declares having shown that the population of recruited saccade-related burst neurons in the SC "specifies the detailed kinematics (trajectories and velocity profiles)" of saccadic eye-head gaze shifts (abstract L8-9), the readers should be warned that:

l) this conclusion is based upon the analysis of the correlation between a particular analysis of neuronal activity and the instantaneous velocity of gaze shifts. The methods that he used for quantifying this activity are popular (many electrophysiologists used them). However not all neurophysiologists did.

Indeed, by convolving each action potential with a Gaussian, the author transforms the sequence of discrete events into a continuous function (the so-called spike density function), which permits to study its correlation with another continuous function like the velocity profile of gaze orienting movement.

However, this method has several drawbacks about which the readers should be warned.

Firstly, the transformation of the sequence of spikes into a continuous function feeds the illusion that a kinematic parameter is encoded by the activity of neurons and transmitted to post-synaptic neurons. In fact, the post-synaptic neurons integrate input from multiple neurons located in several regions. If the time course of SC activity looks like the velocity of eye (or gaze) saccade, how do we explain the fact that the firing rate of burst neurons does not (see their quasi-constant interspike intervals in Hu et al. 2007). Secondly, the cumulative Gaussian curves create an event (the peak firing rate) that may not have any physiological counterpart if its value and its timing differ from the actual shortest interspike interval and its timing. Secondly, the symmetry of the Gaussian curve creates a situation in which each convolved spike exerts an influence backward in time, shifting the putative peak firing rate toward the gaze saccade onset.

These points are well taken. Clearly, the discrete point process of action potentials generated by a single neuron will never correlate well with a continuous movement trace, let alone the very noisy discrete measure of the neuron's firing rate when it is directly determined from the discrete inter-spike intervals. Thus, convolving spikes with some kernel changes the representation of the discrete neural activity into a continuous signal and I took an often-used Gaussian kernel with a width of 4 ms, as is also done by others in the field. However, the suggestion that the kernel itself could produce a signal related to the movement trajectory as an artefact even when the underlying spikes would have no such relation whatsoever, is not supported by my recordings. If the SC spikes would be regularly spaced (i.e., would have a constant firing rate), the convolved spike density would also be a (slightly wobbly) constant signal.

It's important to note, however, that the movement-field description (Eqn. 9, and the (new) figures 4C, 5A, 6, and S5) is not based on the convolved spike density (as a measure for firing rate), or cumulative spike density (as a measure for spike count) but on the actual discrete and recorded number of spikes.

But yes, in relating the instantaneous cumulative number of spikes to the instantaneous straight-line trajectory of gaze displacement (as predicted by Eqn. 10), I show the continuous cumulative spike density vs. the continuous gaze displacement in Fig. 5B. However, to convince the reader and the reviewer that the same result follows when instead taking the discrete cumulative spike count to make this correlation, I have now illustrated this point in the Supplemental material, in the new Figures S13 and S14 (the blue traces).

In my revision, I took the point on the influence of spike convolution on the representation of neural activity very seriously, and therefore I also implemented a different convolution scheme, in which the kernel width was taken identical to the instantaneous interspike interval (adaptive kernels). I also describe this important issue in more detail the Methods section, and included an additional set of figures in the Supplemental Material to illustrate the effects.

I show data of the true instantaneous (discrete) firing rates (in blue) in the new Supplemental Figure S15 for the same 25 trials as in Fig. 10B, together with the spike densities of the two kernel-based methods (fixed 4 ms, in black, and adaptive kernels, in green). Clearly, correlations drop to lower values for the adaptive kernels (the histogram mode shifts from $r=0.88$ to $r=0.75$) if the kernel representation approaches the instantaneous discrete firing rates better and better. Yet, the overall conclusion of my analysis that these relations are real and differ profoundly when the trials are scrambled, still holds. I now show this in the new Supplemental Figures S16 and S17 which compare the two kernel methods directly (similar histograms as in Fig. 10C and D).

As explained in the first review (sixth major concern and comments on the previous Fig. 9), if a causal relation exists between the firing rate of neurons and the gaze velocity profile, then spikes emitted AFTER the time of gaze peak velocity cannot influence the gaze peak velocity value.

If they do, it is a mathematical artefact resulting from convolving each spike with a symmetric curve (Gaussian).

Therefore, this reviewer invites the author to warn the readers about this concern and to document both the time of peak firing rate (estimated by the spike density function) and the time of shortest interspike interval (ordinate), both as a function of the time of peak gaze velocity (abscissa).

A symmetrical Gaussian ensures that the peak activity stays at the right timing, and that it is not delayed, as could happen with a kernel described by a causal exponential decay (weighting only $t > 0$).

I determined the timing of the peaks of both the estimated (4 ms kernel-based) firing-rate profiles and the time of peak gaze velocity for all trials concerned. I now included many more trials than in my earlier version by relaxing the (earlier) demand to having at least three identical gaze-shift vectors within a 2×2 deg bin. This increased the total number of identified gaze shifts to 538 for 8 cells. Clearly, determining the exact moment of a peak is only reliable when signals are sharply single-peaked, which is often not the case, even for gaze-velocity profiles (especially in the ipsi-eye condition, as can be seen in Fig. 3B and S9). Yet, for the population of responses the peak spike-density (4 ms kernels) led the peak gaze velocity on average by 12.4 ms. I added an inset in Figure 9B

that shows that for the majority of gaze shifts in that panel the points lie below the diagonal (meaning: cell leads). Supplemental Figure S12 shows the distribution of onsets for all 538 trials.

II) the conclusion that the population of recruited saccade-related burst neurons in the SC specifies the detailed kinematics of gaze shifts cannot be supported by the reported data insofar as the physical trajectory of gaze direction (as schematized in Fig. S3 by $G_1(t)$ or $G_2(t)$) is projected onto the imaginary line segment (DELTA_G) that connects the starting gaze position to the final gaze position. Because of this projection, it is not the detailed kinematics that is described.

Here, the reviewer correctly identifies the major conceptual point of my study, which is indeed what I claim in this paper: the SC neurons do NOT specify the ACTUAL gaze velocity. A single cell cannot do that as it can, at best, contribute a straight (mini) displacement when its connections to the brainstem during a gaze shift remain fixed (which seems a reasonable assumption to me, although Peel et al (2021) suggested that it could change under FEF inactivation). It is for that reason that I illustrate the concept of my analysis and interpretations in Fig. S3 for two identical gaze shift vectors with different (curved) movement trajectories. The idea is (and this is supported by the data) that for each fixed gaze-shift vector the number of spikes is the same, but that the firing patterns differ, because the underlying gaze-velocities differ: a curved saccade is slower than a straight saccade (and it's also slower when the head contribution increases, see Figs. 2, 3B). As a single cell only 'sees' a straight trajectory (namely the one that equals its optimal movement-field vector), the speed with which the projections of the measured curved trajectories move along the cell's optimal straight line differs too, and this is reflected in the cell's firing rate. I claim that this recalculated speed is the signal that is issued by each SC cell's firing rate as its desired straight gaze-shift velocity profile, which will often differ from the actual curved gaze velocity.

The total desired gaze-velocity profile is (according to the summation hypothesis) issued by the total population and would (according to our Eqn. 1) be determined by the weighted sum of the contributions from all involved SC cells.

However, note that in this paper, I do NOT test this latter prediction, as I do not reconstruct desired gaze-velocity profiles from the joint population activity of the 43 recorded neurons.

One problem (yet to be solved) for testing Eqn. 1 directly, is that it would also have to incorporate an explicit model for the brainstem-cerebellar circuitry (where the scheme in Fig. 1B is just one of many different possibilities). For eye-only saccades we got away with a very simple linear brainstem model (Fig. 1A; Goossens & vO 2006), but it is unlikely that a similar simple linear filter would suffice for eye-head gaze shifts. This is also suggested by the analysis of the influence of head-movement amplitude on the single-trial correlations (Fig. 11B). Performing such simulations for eye-head gaze shifts falls beyond the scope of the present paper, as it will be a full study in itself.

Let's consider the trajectory $G_2(t)$ in Fig. S3, it is below the imaginary line segment DELTA_G. We can imagine its symmetrical image $G_{2SYM}(t)$ with respect to the same line segment DELTA_G. The projection of $G_2(t)$ onto DELTA_G is identical to the projection of $G_{2SYM}(t)$ in spite of different trajectories, in spite of different ratios between the horizontal and vertical velocities. (note that, apart from the sign, the ratios remain the same) Yet, the firing rate of SC neurons can still support these two different trajectories ($G_2(t)$ and $G_{2SYM}(t)$) if the detailed distribution of activity is different in the SC, skewed toward the medial border of the SC during $G_2(t)$, skewed toward the lateral border of the SC during $G_{2SYM}(t)$. This is a point that the author could have explained to the readers. I thought I did; it is exactly the point of my analysis. Although the summed population model of Eqn. 1 would allow for the generation of curved trajectories too (e.g., when the cells in the population would fire asynchronously, but still in some coordinated way), I do not consider this possibility here, as I have no evidence for this (I recorded single units, and as explained above, I did not attempt to reconstruct the population-encoded gaze shift through Eqn.1). Instead, I conjecture a large amount of synchronization in the population because of the high single-trial correlations that I found.

However, the concern remains that IMAGINARY line segment (DELTA_{G2(t0)}) is correlated with the ARTIFICIAL estimate of neurons' firing rate (density function). This reviewer does not want to prevent the author from analyzing his data the way he likes; she/he merely requests that the author warns the readers about its associated limitations and about the theoretical estimated and empirical data.

I do not understand why this would be a concern. This is exactly what I aim to show from the start. As the reviewer correctly argues, the SC is too far away from the system's output to be able to encode the gaze shift in all its details. I fully agree with this, but it doesn't necessarily mean that the SC can have no impact on the actual execution of gaze shifts. As a car driver I control the gas pedal to make it drive faster or slower, and I use the steering wheel to make it go left or right. Still, although my foot and wrist movements do not specify the exact details of the engine dynamics and forces on its different mechanical parts, I suppose that everybody would agree that it's the driver who controls the car's trajectory.

Despite these two major concerns, this reviewer is inclined to agree with the idea that the population of active neurons in SC steers gaze toward the target location and that the saccade curvature results from fluctuations in the firing rate of recruited neurons.

Note that this is NOT what I have argued. I hope to have explained this better in the section above. I rather propose that additional signals in the downstream systems (related to the list of points raised above by the reviewer, and

the conceptual scheme of Fig. 1B in the Introduction) may significantly modify the straight-line desired vectorial command signal from the SC population.

However, as mentioned in her/his first review, the development of this idea largely benefited from electrical microstimulation studies in the head-restrained and head-unrestrained monkey (Stanford et al. 1996; Freedman et al. 1996).

I cordially disagree with this. Microstimulation at best supports the idea but is hampered with potential artifacts that I would not like to go into. The development of the dynamic ensemble-coding idea is predominantly inspired by SC recordings, not on microstimulation results. The observation that despite a wide variation in saccade kinematics and trajectories the number of spikes in the saccade-related burst remains unaffected is the main inspiration for my analysis. The present study extends the head-restrained saccade study of Goossens and vO (2006, 2012). This idea in turn builds on the population-encoding idea of the SC motor map, first forwarded by James McIlwain (1980), which was later incorporated into a quantitative model description of SC movement fields by Ottes and colleagues (1986), who (yes) used the seminal microstimulation data of Robinson 1972, to determine the motor map parameters), and subsequently by Van Gisbergen and myself (1987) in the (static) ensemble-coding model. This model was subsequently tested (and judged wrong) by Lee, Rohrer and Sparks in their seminal 1988 Nature micro-lesion paper. Goossens and I argued (in 2006) that their results (the systematic deviations of the saccade vector away from the lesioned site, as well as slower saccades) could be explained equally well by our dynamic ensemble-coding model. We demonstrated this by using actual recordings from 140 neurons to drive the linear black-box brainstem model, here shown in Fig. 1A. The idea that the SC issues a dual code (both the vector, and the kinematics) has been proposed by many others, including David Sparks, but first by Berthoz and colleagues (1986).

There is a problem in which an encoding scheme based on vector *averaging* (i.e., a center-of-gravity calculation) can be combined with the kinematics idea, as vector averaging (1) cannot explain why saccades become smaller when the amount of SC activation decreases (e.g., by manipulating electrical stimulation parameters) because even a single spike would suffice to encode the total vector (vector averaging normalizes for the total activity of the population), and (2), therefore, the kinematics are determined downstream. In contrast, our dynamic ensemble-coding scheme reconciled the dual role of the SC (indeed, further supported by stimulation results of Stanford et al., 1996, but earlier also by us in vO et al., 1990) in one and the same mechanism: the number of spikes determines the overall saccade vector, and the firing rate during the burst relates to its kinematics.

These studies showed that electrically-evoked gaze shifts stop as soon as the microstimulation stops and that a minimum number of electrical pulses is required to evoke the saccade vector (amplitude and direction) specific to each site of stimulation. (but as explained above, this is not in line with vector averaging)

They also showed that reducing the stimulation frequency (increasing the duration of the interval between two electrical pulses) evoked slower saccades, the amplitude of which was maintained if the stimulation was prolonged (see Figures 5, 8 and 12 of Stanford et al. 1996 and Figure 3 and 4 of Freedman et al. 1996). Although this reviewer appreciates very much the author's efforts to corroborate, with electrophysiological recordings in the head-unrestrained monkey, the hypothesis that Dr Sparks and his colleagues defended 27 years ago, she/he does not share the same surprise as the author wants to transmit to the readers (abstract L11).

I highly appreciate the work of Dave Sparks and his students, but I wish to highlight here that (1) the population vector idea was first conceptualized by James McIlwain (1980) and was quantitatively formulated by us in 1987. (2) The dependence of saccade properties on microstimulation parameters (including the saturation of saccade amplitude with stimulation current intensity) was first published by us (in 1990). Like the present study, also Stanford et al. (1996) concluded: "The results ... support the hypothesis that independent signals relating to saccade amplitude and velocity are derived from the spatial and temporal aspects of collicular activity."

By neither reminding their results nor by explaining how his recording results are consistent with them, the author does not do a fair justice to the seminal studies of Dr Sparks' group. Likewise, it is also unclear why the author does not discuss his results with regard to the suggestion that the locus of activity in the Superior Colliculus would encode target eccentricity (relative to gaze direction) instead of gaze saccade amplitude (Bergeron et al. 2002; 2003).

The idea that the SC activity would encode the dynamic motor error ('target eccentricity': i.e., the ongoing distance of the eye (or gaze) to the oculocentric target location) either as a moving hill, or as a static hill, has been refuted by many others. I mention these concepts in the Introduction, but space limitations of the journal prevent me from providing an extensive literature review on all the possible ideas on the SC that have been proposed in the past.

Finally, it is important reminding the readers that the firing rate of neurons also depends upon the animal's alertness and motivation, both of which do not vary from one trial to the next. This massive and sluggish influence likely explains the loss of trial-by-trial correlation between the firing rate and gaze velocity when the velocity profiles are shuffled (Fig. 10).

I kindly disagree with this argument, as the shuffling serves an entirely different purpose. It's inconceivable to me how the absence of (putative) changes in alertness would preserve the correlations between strongly varying velocity profiles and firing-rate patterns when they are randomly shuffled.... Or does the reviewer suggest that in case there is no variation in alertness there is also no variation in kinematics and firing rates, and therefore everything will be invariant from trial to trial and shuffling will have no effect on the correlations? In other words, the

variation in kinematics would not be related to the task requirements (changes in the initial eye- and head positions) but to alertness and motivation?

Still concerning the relation to previous studies, the report of the results obtained by Peel et al. (2021) is biased. During FEF inactivation, the number of action potentials emitted by SC neurons is neither increased, nor slight, nor restricted to memory-guided saccades (L141-143). Their figure 1b shows an impressive impairment of delayed-saccades: the total number of spikes during FEF cooling is reduced almost by half (N=13) in comparison to the number of spikes recorded during control saccades (N=22).

Point taken. It shows that the FEF are crucial in the generation of memory-guided delayed saccades.

However, they also show that the direct visual-evoked and gap saccades were hardly affected by FEF cooling. Also, the neural activity without cooling the FEF is unaffected either: the number of spikes was the same for all saccade types (memory, delayed, gap and direct) for the normal functioning system. In my paper, I focus on the normal functioning system.

It's appreciated though that the SC output may be influenced by task-dependent factors, for example, after acute lesions of important input (Peel et al. 2021), or during ocular pursuit (Goffart et al., 2017). I have restated my summary on the Peel et al. results accordingly and now also mention the Goffart et al. (2017) paper in this context.

The horizontal and vertical amplitudes of saccades were barely changed whereas their velocity was severely reduced. The longer interspike intervals associated with the slowing of saccades is consistent with the influence of SC activity upon saccade velocity. Sparks, Lee & Rohrer (1990) reported a similar dramatic decline in velocity with no change in saccade direction and amplitude (see their Figure 3).

Although Sparks and colleagues could not record the firing rate of neurons, the observations of Hanes et al. (2005) that the response fields of neurons adjacent to a small collicular lesion do not expand leads to infer that the lidocaine injection also reduced the number of active neurons. Thus, the suppression of neurons and the lower firing rate of peripheral neurons (peripheral with respect to the center of the normally active population of neurons) can account for the reduced velocity.

It is also worth reminding that Soetedjo et al. (2002) showed that after muscimol injection in the pontine reticular formation, the number of spikes emitted by SC neurons remained the same despite prolonged saccade duration (as expected by the author). However, they provided a different explanation (feedback to the SC). Note that projections from the nucleus prepositus hypoglossi to the SC are consistent with such feedback. It is unclear why the author ignores this anatomical fact (Hartwich-Young et al. 1990). All these observations constitute precious empirical clues that are useful to identify the physiological parameters that can vary from one trial to the next, i.e., the duration of activity, the firing rate, the number of spikes and the number of active neurons.

Feedback from the brainstem into the SC that could potentially instantiate a dynamic motor-error encoding in the motor map has been refuted by others (and also by us, e.g. Goossens and Van Opstal, 2000), and therefore, although the anatomical projection may be there, its functional significance remains unclear.

From the viewpoint of this reviewer, these neurophysiological results obtained sometimes during tedious experiments in the awake and trained monkey do not deserve less emphasis than theoretical speculations such as "additive" (L354, 356) and "multiplicative" (L320, 347, 357) noises in the brain. The neurophysiological meaning of these notions is unclear and possibly misleading. Unit recordings show that the firing rate of any SC neuron is rarely the same from one trial to the next in spite of quasi-identical initial conditions. In his article, the author prefers reporting the mismatch between his equations and the real data rather than reporting the actual data.

Neither the variability of interspike intervals (during the burst) nor the variability of cumulative spikes counts are documented.

I kindly disagree. The spike-density functions are a direct measure (albeit smoothed) of the changes in interspike intervals. If interspike intervals would be constant, also the spike-density functions would have been constant. This is clearly not the case.

Further, all my movement-field plots reflect the variability in (actual discrete) spike counts, not of reconstructed smoothed spike counts. The new analysis (in the revised paper) performed on the noise in Figure 6 (and in the Supplemental Material shown for 19 other cells) provides a precise quantification of the variability in the actual spike counts for all trials, saccade vectors, and for all neurons.

By not showing the staircase of segments composing the cumulative spike numbers, the author prevents the readers from realizing that the emission of action potentials by SC neurons is neither a periodic process nor a continuous one.

As indicated above, the spike counts in my movement-field model are not based on the smoothed spike-density functions. These only serve a visual purpose in the figures. To avoid this confusion, I have now indicated this in the figure captions by specifying 'spike count' for the true spike count, and 'cumulative spike density' for the smoothed spike count. Similarly, I have replaced 'Firing Rate' in all figures by 'Spike Density'.

As I show in the Supplemental Material, however, the cumulative spike-count measure yields essentially the same results as the original discrete spike count.

It should also be noted that my major analysis is on cumulative spike count, and not on instantaneous firing rate. Firing rate is a derived quantity, literally obtained by taking the derivative of the original signal. The firing-rate

measures, however, whether smoothed or not, will always be noisier than the signals related to spike counts (because differentiation amplifies high frequencies).

We have also shown earlier that the number of spikes in the SC burst is a better descriptor for an SC movement field than other derived measures for neural activity, like the peak firing rate, or the average firing rate over some (fixed, or saccade-duration dependent) time window, even though under stereotyped conditions (fast visual-evoked saccades with little variability) all these measures yield the same movement field (Ottens et al., 1986). However, the predictive power breaks down as soon as saccades show increased variability. Then, the only measure that remains invariant is the number of spikes in the burst.

Freedman and Sparks (1997) showed that the interspike intervals are neither constant nor reproducible from one trial to the next (see their Fig. 5).

I claim the same thing in my analysis. I think that there is no discrepancy.

By barely reporting the measured data, he prevents the readers not only from understanding what is variable but also comparing them with observations reported by others, seeing consistency and discrepancies.

Still concerning the report of real data, this reviewer noticed that the values of gaze amplitude and direction in Table S1 do not match the values plotted in Fig. S2. For instance, according to Table S1, $R_0=98.2$ and $\text{THETA}_0=-68.9$ for the neuron Sa0907. This neuron is not visible in Fig. S2. Reciprocally, Fig. S2 shows two blue circles with R values larger than 80 whereas Table S1 reports only one R value larger than 80 ($R_0=94.6$, neuron Pc2406).

Thanks for noticing this error. I have now replotted the recording sites onto the SC motor map for all cells, based on the actual numbers in Table S1.

Likewise, contrary to the author's claim, the data in Fig. 6A do not show "the variability of the burst-related spike count as [a] function of the expected total number of spikes in the burst". In the legend of Fig. 6A, one reads "the error in the predicted number of spikes" (L938). If this definition is correct, then one is led to infer that the plotted error (ordinate axis) corresponds to the difference between the number of spikes predicted by equation 9 and the number of spikes that were actually measured. In other words, it is an index of the goodness of the model (the equation) to capture the data (measured values). Consequently, plotting the difference between the prediction of a model and the data as a function of the predicted number of spikes does not describe a so-called "signal dependent noise" (L305) if by "signal dependent noise", the author refers to something related to the firing rate of neurons.

The reviewer would be correct if the movement-field model would have a systematic error in the prediction. However, this is not the case. The mean number of spikes predicted by the model is the same as the actual number of spikes delivered by the cell across all trials. I therefore can take the model prediction as the measure for the true expected average number of spikes. We find that the variation around the mean increases with the mean, but that the average deviation does not increase with the mean (this is constant). This is the definition of signal-dependent noise.

The difference between the number of spikes predicted by an equation and the number of spikes emitted by a biological neuron has no physiological meaning. Plotting this difference as a function of a predicted value is even more speculative. This other example illustrates the necessity to warn the readers about differences between empirical data and theoretical notions.

I cordially disagree with this statement.

Regarding the increase of the standard deviation with the mean number of spikes (Fig. 6B), it would be useful to show this relation for the neuron sa0107 also (Fig. 5B).

This is shown in the Supplemental Material. The cell indicated by sa01jul09 is the same as sa0107.

Indeed, we can estimate (although with some difficulty) the various amplitudes of gaze shifts that were recorded with this neuron (note that a plot or an inset showing the variability of amplitude and direction of gaze shifts would be helpful in Fig. 5).

I'm afraid it won't really help, as the same number of spikes is emitted (and explained by its movement-field properties, even if no mathematical function is used to describe it ...) by a large number of saccade vectors, over a considerable range of amplitudes and directions. It describes an ellipsoid-like curve around the movement-field center. To illustrate this point, and to help the reader understand this better, I now included an inset figure in Fig 6B, in which I selected the elicited gaze vectors for the number of spikes between $N_{\text{psk}} \pm 1$ (with $N_{\text{psk}} =$ either 10 or 30) in two different ways: first, based on the model prediction of Eqn. 9, and second on the basis of the actual measured spike counts. The reader can appreciate that saccade vector distributions are very similar, but also that they define a curve around the movement field center of the cell.

I strongly believe that the use of mathematical functions to describe the responses of a neuron to a wide variety of conditions and vectors with a limited set of parameters is very useful. It's not trivial that it's possible to fully describe a cell's movement field during gaze shifts (our 'dynamic movement field' model) with only 5 free parameters, leading to coefficients of determination (r^2 , 'variance explained') of 0.9 or higher for more than 30,000 included data points.

Note Fig. 5B shows a counter example to the increase of the standard deviation with the mean number of spikes. Considering a number of spikes close to the maximum ($N > 25$), the values of straight-line gaze displacement ranged from approximately 25 to 48 degrees.

Curiously, considering a smaller number of spikes ($N = 14$), the values of straight-line gaze displacement ranged from approximately 15 to 66 degrees. In other words, the variability is not larger with more numerous spikes. The description of Fig. 5B should be improved because it contains a lot of information. On the one hand, we know that the measured number of spikes depends upon the location of the visual target with respect to the center of the response field (RF). It is larger when the visual stimulus falls in the center of the RF (leading to the maximum saccade-related burst) than when the visual stimulus falls closer to the boundary of the RF (leading to a less vigorous burst). On the other hand, we also know that the size of the RF increases with the target eccentricity, and thus with the location of collicular activity along the rostrocaudal axis (see Fig. 2 of Taouali et al. 2015 and Fig. 8 of Munoz & Wurtz 1995). Therefore, we wonder whether the relation between the mean measured number of spikes and the standard deviation of the measured number of spikes is not the consequence of presenting the target at various locations in the response field of neurons. Finally, it is worth reminding the readers that the firing rate also depends upon the alertness and the motivation of the animal. Keeping these physiological facts in mind, the readers can eventually understand, or not, the meaning of “multiplicative noise in the cell’s activity”.

When assessing the variability in the number of spikes as function of the mean number of spikes, one should look at panel 5A. The intrinsic neural variability is not related to the (intended or actual) saccade amplitude, but to the strength of the neuronal signal (which in our analyses is quantified by the cell’s number of spikes). The same number of spikes is issued by a (limited, but nevertheless considerable) range of saccade vectors, (see now also Fig 6B for an example) and the movement field model gives an accurate estimate of these vectors. The model (5 free parameters if we include eye position) does a good job, as it explains 90% of the variability in the number of spikes across all saccade vectors and initial eye-positions. The mean error of the model is zero; it is unbiased. Yet, there is considerable neural variability around the estimated mean, which increases systematically with the mean. This aspect of multiplicative noise is analysed in Figs. 6 and 7 and in Supplemental Figure S10.

Panel 5B is a completely different plot, with a different message. It is not based on any model fit, other than that the 2D gaze trajectories were transformed into the projected straight lines to make this plot possible. As the curvatures of the individual gaze shifts were quite limited, the projected gaze shifts do not differ much from the original gaze shifts. What this plot shows is that the ‘phase trajectories’ (the relation between the instantaneous displacement of gaze and the cumulative number of spikes follows a straight line; these lines are quantified later in Figure 8, showing that the instantaneous deviation of the actual current number of spikes remains within a few spikes from the straight line throughout the trajectory).

Panel 5C then shows that the wide variation seen in the orientations of these straight lines is (again for more 90%) explained by the dynamic movement-field model.

The statement of “excellent agreement between data and model” (L366) illustrates the theoretical inclination of the author. Unfortunately, such a statement is qualitative and rushed. After drawing the $y=x$ line, one sees that the majority of data points are situated above the diagonal. In other words, the predicted slopes overestimate the measured slope. Table S1 tells us that for cell Sa0508, when the gaze amplitude is 11 deg, the predicted number of spikes is 23, making a ratio of 2.09 spikes per degree.

The turquoise line in Fig. 8A shows that for a predicted slope of a slope of 2.09, the measured slope is 1.82 spikes per degree. In other words, for a gaze amplitude of 11 deg, the average measured number of spikes was $11 \times 1.82 = 20$ spikes, i.e., 3 spikes less than the predicted number of spikes ($N_0 = 23$).

Here, one should include in the expectation the predicted variability for the neuron, which is described by the multiplicative noise (as analysed, e.g., in Fig. 6). For a mean expected number of 23 spikes, and a correlation of variance of about 0.2 (one sigma), the expected number of spikes could vary between 14 and 32 spikes, and thus yielding (slightly) different slopes from trial to trial. One cannot refute a model prediction on the basis of a single number. To help the reader, I now added an equation in the text to include the effect of noise on the predicted number of spikes of the standard Movement-Field model (Eqn. 9).

Equation (9) overestimated the measured number of spikes by 15% for a gaze amplitude of 11 deg. Unfortunately, from looking at Fig. 8A, it is impossible to see neither the number of spikes measured during a gaze shift of 44 deg amplitude, nor the predicted number of spikes. Therefore, to facilitate the readers’ understanding, before plotting the predicted number of spikes as a function of the measured number (as in Fig. 8A), the author should plot these two numbers (measured and predicted numbers of spikes) as a function of gaze amplitude for iso-directional gaze shifts ($\text{THETA} = \text{THETA}_0 \pm$ e.g. 5 deg or 10 deg).

I cordially decline to follow this suggestion. I believe to now have substantially documented the validity of the movement-field model in Figures 4 and 5 and I show additional examples in the Supplemental Material (three more cells), and all relevant numbers from the movement-field fits in Table S1. Showing multiple figures for every

cell that I illustrate in the paper is simply not possible, given the space limits of the journal to a maximum of 10 figures.

The plot in Fig. 8A could then be presented using open and filled symbols to distinguish iso-directional gaze shifts from others. Then, the slopes and y-intercepts of the relations between predicted and measured spikes could be documented for all cells so that the readers can see whether equation (9) systematically overestimated the measured number of spikes for each cell.

Figure 8B shows that there is no systematic deviation from the raw straight-line fits, and Fig. S6B shows the mean and standard deviations of the instantaneous differences between the dynamic movement-field predictions of Eqn. 9 (which is entirely based on the simple static movement-field model and therefore has no extra free parameters!) and the measured instantaneous values (like shown in Fig. 5C) for all 43 neurons. I believe that this figure adequately responds to the reviewer's concern.

Below, the author will find more detailed comments on the figures and on the text. They are not exhaustive because reading this article was rather time-consuming and difficult for this reviewer.

COMMENTS ON THE OTHER FIGURES:

Figure 3:

panel A: Looking at the values for the head at gaze onset, this reviewer suspects that the title of the ordinate axis is G/H Displacement instead of G/H Position. It is also very rare that gaze direction is strictly equal to zero. **It is not. The gaze directions vary but the amplitudes were selected to a bin of 1.0 deg width.** The legend should be corrected accordingly (L900 and 901).

Ok, done

Panel B: one red curve (out of eight) is shifted toward negative values. It seems that the onset of the corresponding gaze shift was not properly detected (its velocity is larger than 250 deg/s at gaze onset). The author should check the impact of this mistake on the red curves illustrating the spike density function and the mean cumulative spike count.

This indeed reflected a rare detection error for this particular gaze shift. I identified the trial and corrected its onset by hand (10 ms delayed). This had a negligible effect on the cumulative spike density curves.

Figure 4:

Panel C: The blue curves are barely visible. The author could use either a lighter blue (e.g., turquoise) or change the color of the background from black to grey. A scale for the color codes (number of spikes) would be welcome or an indication of the number of spikes corresponding to the thin white contour lines. What is Fopt? The same comments hold for Fig. S5.

Ok, changed. I also replaced F_{opt} by N_0 (the fitted maximum number of spikes in the burst) and added a color bar for the number of spikes. I did the same for the cells in Fig. S5.

Figure 5:

Legend:

L929: NSTMF instead of NMFST, in accordance with equation 9 (L771) **Ok, changed**

The author name this plot "Dynamic movement field" whereas the term "phase trajectory" is used in several places in the text (L289, L293, L364, L378, L523, L962). If the two terminologies denote the same plot, the author should use only one to remove ambiguities. If they do not denote the same plot, then the author add an explanatory note.

They do not. The phase trajectories of Fig. 5B represent nearly raw data, as explained above (instantaneous measured number of spikes (smoothed, or not smoothed, doesn't really matter, as shown in Fig. S13) vs. measured gaze displacement along the straight line connecting initial and final position (the DELTAG in Fig. S3). The dynamic movement field is shown in panel 5C. The static movement field in panel 5A. Panel 5B shows the wide range of dynamic response patterns (lines with many different slopes), which are quantitatively explained by the dynamic movement field.

Panel B: Ordinate axis: Which "Cumulative Spike Count" is plotted? The predicted spike count (as in panel C) or the measured spike count? It would be useful to make it clear in the figure.

It's the measured cumulative spike count. Is now indicated.

Panel C: It may be preferable to plot the measured (ordinate) as a function of the predicted cumulative spike count (abscissa) as in Fig. 4C and Fig. S5. Maybe not.

I prefer not to, but thanks for the observation, because I now did the same for Figures 4C and S5.

Figure 6:

Title of y-axis: Model's prediction error

I hope that 'Error' suffices.

Figure 8:

Panel A: The scales of x and y-axes do not facilitate the calculations that some readers would like to perform. A step of 0.5 is preferable to a step of 0.52. It is fine to use 3.0 as maximum values. Drawing the diagonal $y=x$ will

enable the readers to see that the predicted slope overestimates the measured slope (majority of data points above the diagonal). The turquoise line likely illustrates the slope of the scatter. Unfortunately, it is not defined in the legend. Its slope, y-intercept and the number of datapoints should be documented in the legend.

Ok, updated

Panels A and B: The cell identification number (Sa0508) should be inserted in order to facilitate the distinction between the plots corresponding to one cell and those corresponding to several cells.

ok, done

Figure 9:

Panel A: As requested in the previous review, it is preferable, for conventional reasons, to use the word “amplitude” instead of “shift”.

Ok, changed; sorry to have missed that...

Panel B: An inset should be added to show the delay between the time of peak firing rate and the time of peak gaze velocity. It is important for the readers to realize that in order to influence the peak velocity, the peak firing rate leads and not lags time to peak firing rate (see major comment).

Ok, I have now added this information.

Panel C: A graph plotting the time to peak gaze velocity as a function of time to peak firing rate should be added to assure that the peak firing rate led (and not lagged) time to peak firing rate (see major comment).

See my response above and the Supplemental Figs. S7 (multiple regression) and S12 (peak onsets).

Table 1

The names of parameters do not match between table 1 and L778.

Legend, second line: FMFST=? corrected

Fig. S5:

Legend: last sentence: “faster” instead of “fastest”? corrected.

MORE DETAILED COMMENTS:

L30-35: It would be useful to warn the readers that gaze shifts may also consist in multiple saccades (e.g., Bergeron & Guitton 2002; Anastasopoulos et al. 2015). The readers should also be warned that very large gaze shifts performed within a single step may result from experimental constraints (reinforcement learning) if the monkeys are required to perform their primary gaze saccades so that gaze lands within a (relatively) small window for obtaining a reward.

I did not observe multiple-step gaze shifts in my data base, and my analysis is based on first-saccade responses. Correction saccades, if any, were typically very small, and were never related to an associated burst of activity.

L 48-49: The references Quinet & Goffart (2007, 2009) are inappropriate to raise an objection against the author’s statement. Indeed, the results of both studies may indicate that the activity of neurons in the caudal fastigial nuclei contributes to coupling the eye and head movements insofar as inactivating one side impairs the eye-head coupling. I removed ‘but see’ Together with other anatomical and electrophysiological data (see their discussion), they indicate that eye-head gaze shifts are under cerebellar control and that the fastigial exerts its influence primarily upon the oculomotor network. This reviewer invited the author to consider these works in order to warn the readers that the kinematics of eye and head movements is also influenced by the activity of neurons in the caudal fastigial nucleus and that a gaze feedback control is controversial.

I would like to stress that my study does not directly concern any conclusions related to the tedious interplay between brainstem and cerebellar circuitries. My data do not argue for or against any of the proposed mechanisms, including gaze feedback. I explained my point of view on this above. In this study, I am only concerned with the SC firing patterns during a wide variety of gaze-shift kinematics. I relate SC activity to the final output of the system, which are the gaze and head movement. I do not use my recordings to simulate a gaze shift, which would require an eye-head coupled motor circuit model. Although the latter circuitry is highly interesting, I do not wish to speculate (yet) on its detailed organization on the basis of my recordings.

L140: “Peel et al. (2021)” instead of “Peel et al. (2020)”. corrected

L143: “smaller” instead of “slightly higher”. changed

L147: The word “however” should be removed because the following sentence (“the variability ... difficult to detect”) is consistent with the previous sentence (“Further, Zhang et al. (2022) ... for all conditions”). Or the argument is not clear. I replaced ‘however’ by ‘yet’: there is a contradiction between the two sentences. I want to make clear that the conclusion by Zhang et al. is not warranted because of the limited range in saccade kinematics. One can then not conclude that neural activity is not related to saccade kinematics.

L153: This reviewer fears that the suggestion that “the SC has no central role in saccade control (Daye et al. 2014)” (L153) misleads the readers. She/invites the author to either remove it or insert “Moreover” before “immediate deficits in saccade metrics ...” (L156) and add “immediately” between “observed” and “after acute reversible microlesions” (L157). done

L162-166: It would be useful to inform the young readers that this crucial role of both SC and FEF was suggested by the lesion study performed by Schiller, True & Conway (1980). I added the reference.

L196: It would be useful to add a reference showing that the rotation axes of the head do not intersect in a fixed

point. I now added the study of Medendorp et al.

L216: “during experiment Sa1007” instead of “of experiment Sa1007” corrected

L220: “when the eye looked ipsilateral to the target” is clumsy and potentially confusing because the target is being foveated. Does the author mean “when the eye was deviated toward the side ipsilateral to the direction of the imminent gaze shift”? yes, that is what I mean. I reformulated this according to the reviewer’s suggestion.

L223: “with ipsilateral eye deviations” instead of “with the eye ipsilateral”? corrected

L224: It would be neat to document the percentages of prolongation (for the duration) and reduction (for the peak velocity). This is hard to capture in a single number, as it depends on the gaze-shift amplitude, as well as on gaze-shift direction. As can be seen, the kinematics were quite variable too, with a considerable amount of overlap. Instead, I now refer the reader to Fig. 3B to get an idea on the approximate size of these effects.

L234: “Figure 3 shows a selection of gaze shifts recorded during this experiment and initiated from the three initial eye positions” instead of “Figure 3 shows a selection of gaze shifts from this experiment from the three initial eye positions” copied.

L237-248: The author should check whether the statements about the peak firing rate and its timing remain true after modifying Fig. 3. L249: “ipsi-condition (blue)” instead of “contra-condition (blue)” Yes, I checked and verified that nothing has changed after correcting the single trial.

L244-253: The author should warn the reader that the interspike intervals vary considerably across the trials and that the mean cumulative spike counts masks the variable firing rate of SC neurons, which contrasts with the clock-like firing rate of premotor burst neurons (Hu et al. 2007).

I now alert the reader to the intertrial variability. As I do not use my recordings for gaze-shift simulations, I see no immediate need to refer to brainstem recordings. As a side point (at the risk of irrelevance) I could refer the reviewer to Van Opstal and Kasap (2019), where we explain in the Discussion that even for the case in which single neurons have a flat firing rate, the total population can still faithfully encode the actual gaze-velocity profile. I just mean to say that when individual neurons may show no correlation whatsoever with a particular motor-output variable, it doesn’t necessarily mean that these neurons do not control that motor output variable. However, this argument is not needed for the motor SC, as single cells do seem to encode the motor output kinematics to a high degree.

L249: “the timing of its peak” instead of “its peak location” done

L258: “all action potentials” instead of “all neural responses” indeed, changed.

L264-265: Choi & Guitton (2009) showed that the population the burst does not always ends before gaze end (see middle row of Figure 6C). Therefore, this reviewer suggests replacing “as reported in earlier studies (Freedman & Sparks, 1997; Choi and Guitton, 2009)” by “as reported by Freedman & Sparks (1997), but not always (see e.g., Choi & Guitton 2009)”. ok

L268: “toward the center of the movement field” instead of “into the center of the movement field”. corrected

L268: “as well as” instead of “as well:” changed

L271: “cumulative spike count (CSC) instead of “CSC” because the abbreviation has not been previously defined. done

L271-272: The author remarks that “the CSC trajectories nearly fully overlap along the $y=x$ diagonal, despite the considerable variation in gaze-shift kinematics”. The overlap indicates a good match between the measured and the predicted cumulative spike counts”. Note that the methods used to measure the number of spikes (Fig. 5) are not documented.

I now added: measured cumulative spike count vs. predicted by Eqn. 12

L275: “which is indeed seen in the CSC trajectories”: an arrow would be useful to indicate at which location the reader must look to see.

I added: which can indeed be seen in the displaced endpoints of the trajectories.

L274: $N_{opt}=30$ whereas $F_{opt}=25$ in Fig. 4C. What is F_{opt} ? What is N_{opt} ?

Changed; they are identical.

L291: Fig. 5B shows that for a given value of measured total number of spikes (e.g., $N=16$), the straight-line gaze displacement amplitude can range from 19 deg to 62 deg? The author should explain why a large

Yes, this is because all saccade vectors on an ellipsoid around the movement field center are associated with the same number of spikes. Especially MFs with a large vector amplitude span a large range of saccade vectors. I now added a statement to this effect in the revised text (lines 353-356).

L301: “predicted” instead of “expected” changed

L302: “ranged from 16 to 24 spikes” instead of “varied between 16 and 24 spikes” ok

L303: “the mismatch between the predicted and the measured number of spikes” instead of “this variability”. By using the word “variability”, the author makes a semantic drift which leads the readers to confound the variable firing rate of neurons visible across the trials (as illustrated in Fig. 3 and Fig. 4) with the ability of an equation to fully account for the measured values.

I copied the reviewer’s suggestion.

L303: +/- 20% ? How was this calculated ? Is it the average value?

This value refers to the example: Delta is 4 spikes which is 20% of 20 spikes.

L307-312: This paragraph is unclear and should be rephrased.

I now added more information to this paragraph and refer to different figures for further clarification. Lines 382-390

L309-310: “a predicted number of spikes is calculated for all saccade vectors that lie ...” would be preferable instead of “a particular expected number of spikes is predicted for all saccade vectors that lie ...”? **changed**

L318: “predicted” instead of “expected” **ok**

L317-318: See major comment **see response.**

L322: “slope=0.141” instead of “Cv=0.141” **ok**

L332: Suggested change: “invariant to changes in starting eye positions” instead of “invariant to changes in starting eye positions” **I added initial**

L344: For the sake of simplicity, the sentence “Figure 7C provides histograms for the coefficients of variation (slopes of the relationship), Fig. 7D shows the distribution of correlations” can be replaced by “Figure 7C provides histograms for the slopes of the relationship and Fig. 7D the distribution of correlation coefficients between the average number of spikes and standard deviation. **changed**

L346: The wording “the neuronal variability” is vague. The author meant “the variability in the number of spikes during the gaze movement related burst”. **ok**

L347: The author should explain the physiological meaning of “multiplicative noise”.

The spiking of a neuron, N (number of spikes), is a stochastic variable, and can be described as a function of its total synaptic signal input, S , and noise. Two types of noise are considered: additive noise, n_{add} , which is a constant amount of (Gaussian) noise that is independent of the stimulus conditions (e.g., often seen as spontaneous activity), and signal-dependent noise, n_{sig} . Taken together, $N = n_{\text{add}} + S(1+n_{\text{sig}})$. The former type of noise is due to external factors, like the external neuronal input current, or the ionic pump mechanism, while the latter is an intrinsic noise that is thought to be part of the nonlinear spike-generating mechanism, in combination with stochasticity of neural conductances that transform the external input current in (voltage-gated) membrane potential fluctuations. Multiplicative noise leads to larger response variability for stronger inputs. It has been suggested that an efficient control mechanism (like the saccadic system) should account for this variability, which is manifest at larger saccades (e.g., main sequence saturation of peak velocity; increased endpoint variability for larger saccades) and has been hypothesized by several researchers as a particular feature of neuronal encoding that actually improves, rather than hampers, overall system performance. Therefore, according to some investigators, it serves a purpose. It has also been part of theories regarding optimal (movement) control.

L349-351: For consistency, the sentence “The median value for the population, $CV = 0.24$, is comparable to the mean $CV = 0.28 \pm 0.16$ reported by Goossens and Van Opstal (2012) for head-restrained eye saccades from >100 neurons across the SC motor map.” Could be replaced by “The median slope value for the population (0.24) is comparable to the mean slope = 0.28 ± 0.16 reported by Goossens and Van Opstal (2012) for >100 neurons across the SC motor map during head-restrained eye saccades.” **Ok**

L351-352: The sentence “The low values of CV ($\ll 1.0$) 352 indicate that SC cells are typically characterized as being low variance” should be rephrased because it is unclear. It sounds like a tautological statement. **I removed this remark.**

L358-359: The sentence “for a typical maximum of 25 spikes, the standard deviation is $0.24 \times 25 = \pm 6$ spikes (i.e., ranging from 19-31 spikes)” is unclear and seems to be wrong. The author estimates the minimum value to $25 - 6 = 19$ spikes and the maximum value to $25 + 6 = 31$ spikes, but the interval mean \pm one standard deviation does not contain 100 % but 68 % of values. In fact, 99 % of values are contained in the interval mean \pm 3 standard deviations.

The reviewer is correct; I now changed that. The range would run between about 13-37 spikes, which is also in line with Fig. 6A. Line 480.

L364: Could “(as in Fig. 5B)” be added after “phase trajectories”? If it cannot, then it is unclear what the measured slope is. **Added**

L368-379: This paragraph is unclear and should be rephrased. This reviewer understands that Fig. 8B plots the difference between measured cumulative number of spikes and the predicted cumulative number of spikes. Contrary to the predicted cumulative number of spikes, the measured cumulative number of spikes is NOT a line, i.e., a continuous function. The cumulative number of spikes is a step function (like a staircase) with irregular steps (see for instance Sparks 1976, 1978, Sparks & Porter 1983; Moschovakis et al. 1988; Freedman & Sparks 1997; Goffart et al. 2017). It is important to warn the young and unwary readers that the interspike interval is neither constant nor reproducible from one trial to the next. Therefore, this reviewer requests a figure that shows the difference between the predicted and the measured cumulative number of spikes by plotting 1) both of them as a function of time from saccade onset to gaze end and 2) the difference between the measured interspike intervals and the predicted interspike intervals. See also major comment.

I have now done so. In Supplemental Material Fig. S13, I plotted for the 25 trials that are also shown in Fig. 10B, the instantaneous gaze displacement along the straight line (red), the cumulative (staircase-like) spike counts (blue), the cumulative spike density with fixed 4 ms Gaussian kernel (black), and the cumulative spike density for the adaptive spike kernels (green). Every panel also provides the correlations between the different traces: r_{4-D} is the correlation between red and black; r_{AD-D} is the correlation between red and green; r_{NS-D} is the correlation between

red and blue (requested by the reviewer). The reader may convince her/himself that these correlations hardly differ (up to the third decimal).

In figure S14 I normalized all traces (gaze shifts and spike counts/cumulative spike densities) and resampled all traces to 1000 points. This analysis has been done by Goossens and Van Opstal in 2006 (it's Eqn 3) and it underlies the dynamic spike-count model of Eqn. 1. It also seems to hold pretty well for the head-unrestrained gaze shifts. It shows how instantaneous gaze trajectories and instantaneous spike-count trajectories are related for all saccades. The perfect model is indicated by the cyan line (Eq. 10). Fig. S14A corresponds to the 25 trials of Fig. S13; it shows the results for the three different spike-count measures. Fig. S14B shows the result for all 350 trials of this cell. In Fig. S14C I show the averaged trajectories (based on the true spike counts; light-grey background shows all 350 trials) for the three different initial eye-position conditions of the experiment with their profound differences in underlying gaze kinematics. These average trajectories are smooth and remain very close to the ideal diagonal line. The inset shows the instantaneous deviations, indicating a systematic small 'overshoot' (largest for the ipsilateral condition, in line with Fig. 11B), and a small late 'undershoot' of about 5%. I agree that the model is not perfect, but I hope that the reviewer agrees with me that the deviations are very small. Also, the results of my analysis of instantaneous spike counts do not depend on the smoothing kernels.

L402-409: See major comment See my response regarding the peak-timing analysis

L471: "Kolmogorov" instead of "Kolomogorov" **changed**

Reviewer 2:

I have read Dr. Van Opstal's response to reviews and the revised manuscript very closely, and I thank him for his considered responses. He has address all of my points thoroughly in his response. The only minor modification to the manuscript that I would suggest, based on my previous reviews, is in lines 194-195 (referencing the clean version), in regards to the new text on the non-linearity of head biomechanics. As written, it sounds like the non-linearities arise solely from the comparative rotational axes. In reality, the head's inertia and the complexity of force development in muscles that can be dynamically loaded (e.g., the force velocity curve) are likely dominant factors in the nonlinearity. My recommendation is to include such factors explicitly in the text near 194-195.

I have now done so (Lines 222-223).

REVIEWERS' COMMENTS:

Reviewer #1 (Remarks to the Author):

Extending the conclusions of Smalianchuk et al. (2018) to gaze shifts made with the unrestrained head, the author concludes that the "SC population activity encodes the instantaneous kinematics of the desired gaze shift through its firing rates" based on a "tight correlation" between the velocity profile and the spike density profile of individual neurons (L728-729, L734, L746-748, L752). He contends that "the finding that single-trial and single-unit firing dynamics at a central neural stage correlate well with the instantaneous motor output of a highly complex and nonlinear synergistic system (comprising the multiple-degrees of freedom oculomotor, head-motor, and vestibular systems; see Fig. 1B) is quite remarkable" (L769-771).

In spite of careful readings of his manuscripts, this reviewer remains unconvinced. In each of her/his reviews, she/he exposed several concerns that prevented her/him from admitting the author's conclusion. The author and this reviewer agree on several points. However, regarding the correlation between the spike density and the saccade velocity, this reviewer suspects that it is actually an artefact for the following reasons.

1) From looking at the figures (Fig. 3C: activity of neuron Sa1007, Fig. 4B, Fig. 10B and Fig. S1: activity of neuron Sa3006, Fig. S15: activity of neuron Sa0107), the reader is led to the impression that all saccade-related neurons in the SC cease emitting action potentials shortly before the gaze shift ends. However, this inference may result from the author's choice of example neurons. Indeed, all saccade-related neurons in the SC do not exhibit an abrupt cessation of activity shortly before saccade end. In the head-unrestrained monkey, Choi and Guitton (2009) documented several neurons that continue to fire after the end of the gaze shift (see the middle row of their Figure 6C). In the head-restrained monkey, numerous studies documented neural discharges that persist after saccade end (Anderson et al. (1998); Goossens and van Opstal (2000); Keller et al. (2000), Munoz and Wurtz (1995), Munoz et al. (1996); Rodgers et al. (2006); Sparks and Mays (1980); Waitzman et al. (1991)). The author did not document how many neurons exhibited this persistent activity after gaze saccade end. It is also unclear whether his analysis was only restricted to the so-called "clipped" saccade-related cells.

2) If the time course of the instantaneous spike density and the time course of "gaze-track" velocity look remarkably similar for the neuron (Sa0107) illustrated in Fig. 10 A-B, it is also because the spikes that preceded the saccade-related burst were removed. The graph at the bottom of Fig. 10A shows that twenty milliseconds before gaze onset, the firing rate rises from 0 to 200 spikes per second. However, examination of Fig. 4A (same neuron) reveals that such an enhancement (from zero to 200 spikes/s) is rare. It seems that the author removed the spikes that were emitted before the onset of an analysis interval (elapsed from 20 ms before gaze saccade onset to 20 ms before gaze saccade end). However, numerous studies (see references listed above) reported collicular saccade-related neurons that emit spikes sometimes within a long prelude before saccade onset.

Thus, by selecting only the spikes emitted from 20 ms before saccade onset to 20 ms before saccade end, the spike density exhibits a rising phase like the acceleration part of the velocity profile, and a decline like the deceleration part of the velocity profile. The correlation between the instantaneous firing rate and the gaze velocity may be the consequence of selecting a portion of the neuron's activity.

The author may wish to explain to the readers that the premotor neurons are sensitive to the spikes that collicular neurons emit during a specific time interval, that this interval starts 20 ms before saccade onset and terminates 20 ms before saccade end. He may wish to add that the onset and the end of this interval is determined by the pause of firing from a specific group of inhibitory neurons located in the nucleus raphe interpositus: the so-called omnipause neurons. If so, then the author should warn the readers that this scenario remains controversial. Indeed, if this hypothesis were true,

then experimentally increasing the pause duration should lead to hypermetric saccades. Empirical studies actually show the lesion of these neurons does not lead to dysmetric saccades (Kaneko 1996; Soetedjo et al. 2000). The author may also explain why these results are not convincing.

3) Moreover, plotting the number of trials (or responses) as a function of the correlation coefficient between spike density and gaze velocity (Fig. 10) does not teach us anything about the proportion of neurons that were concerned. Fig. 10C shows that for one neuron (Sa0107), the correlation coefficient ranged from 0.8 to 1.0 in approximately 330 trials out of a total number of 664 trials. Fig. 10D shows that for 20 best-recorded cells, the correlation coefficient ranged from 0.8 to 1.0 in approximately 1022 trials out of 3981 trials. Thus, 32% (330/1021) of the correlation coefficients that ranged from 0.8 to 1.0 are only due to neuron Sa0107. How many neurons account for the remaining 691 correlations? Three neurons like Sa0107 would be sufficient to account for the 1021 trials. In other words, the claim of a "tight correlation" between the velocity profile and the spike density profile of individual neurons" (L728-729, L734, L746-748, L752) is a hasty conclusion if it concerns a small percentage of neurons. Three neurons out of 20 best recorded cells or three neurons out of 43 cells correspond to a small percentage of cells (15% and 7 %, respectively).

For these reasons and for other reasons exposed in her/his previous reviews (see also reservations expressed by Goffart et al. 2018 about the encoding of velocity by central neurons), this reviewer thinks that the author's conclusions are not yet sufficiently founded.

However, for the sake of pedagogy to the uninformed readers and for the sake also of reminding the tremendous knowledge acquired by previous neurophysiological studies in the awake and trained monkey, this reviewer supports the publication of the author's work as long as the readers are warned about its limitations and shortcomings.

Minor comments:

Abstract:

The last sentence sounds weird. The author may remove "we hypothesize that"

L5: perhaps "amplitude and direction" instead of "coordinates"?

Introduction:

L157: in Goffart et al. (2017), the saccades were not triggered during smooth pursuit tracking but from fixating a static target. The author could replace "occurring during smooth pursuit tracking" by "toward a moving target". Moreover, the word "always" could be inserted between "does not" and "incorporate" because the difference of firing rate between "centrifugal" and "centripetal" did not concern all cells.

L164: this reviewer suggests removing "FEF-mediated" because one cannot exclude the involvement of the cortico-ponto-cerebellar channels originating in the parietal eye fields.

L166: the author might add "in the SC" after "microlesions".

L173: perhaps "primary" instead of "central"?

L174: the author might add "other neural regions such as" between "taken over by" and "the FEF".

Results:

L272: the author might add "various" before "directions"

L418: For the readers who are not familiar with the notion of "multiplicative noise", it would be helpful to explain its meaning.

L486: "by fitting a linear regression line through the measured phase trajectory" instead of "by fitting linear regression lines through the measured phase trajectories"?

L496: "as a function of" instead of "as function of"

L501: what does "c.q." mean?

L506-508: the author forgot to remove his comment.

L610-611: "for the trial (#670) that yielded the largest number of spikes" instead of "for trial 670 that yielded the largest number of spikes"

Discussion:

L836: "Moschovakis" instead of "Moschiovakis"

References:

Page 28, second reference: "Moschovakis" instead of "Moschiovakis"

Page 29, 16th reference: "Sparks DL" instead of "Sparks DJ"

Methods:

L927: the minimum value of the range of successful trials is unclear: "1-0"

L950: "gaze" instead of "eye-".

L1065: "a SC neuron" instead of "an SC neuron"

L1072: "velocity" instead of "vlocity"

L1137: "as a function of" instead of "as function of"

Figure 4A:

The dashed line that indicates the gaze onset does not look cyan but green.

Figure 10A:

It seems that the panel A does not show all action potentials that the cell (Sa0107) emitted but only those that were emitted during the interval elapsed from 20 ms before gaze shift onset to 20 ms before gaze shift end (small black lines). Figure 4A shows numerous examples in which the same cell exhibits a prelude activity. The author should warn the reader that the curve representing the purportedly "instantaneous" firing rate (black) at the bottom of Fig. 4A is rare, or that it is not the true instantaneous firing rate. This reviewer wonders whether it is a fictive instantaneous firing rate, merely created by selecting the spikes emitted during a time interval that the author chose to elapse from 20 ms before gaze shift onset to 20 ms before gaze shift end.

Table 1: the eye position sensitivity "EPSILON ZERO" should be replaced by "EPSILON" to be consistent with the definition in the methods (L1027) and the numerous instances elsewhere in the text (pages 11 and 14).

Based on the major comment, it would be useful to add information about the firing rates at two times: 20 ms before saccade onset and at saccade end.

Figure S10:

L1268-1269: there is no Fig. 6C

REFERENCES:

- Anderson, R. W., Keller, E. L., Gandhi, N. J., & Das, S. (1998). Two-dimensional saccade-related population activity in superior colliculus in monkey. *Journal of Neurophysiology*, 80(2), 798-817.
- Choi, W. Y., & Guitton, D. (2009). Firing patterns in superior colliculus of head-unrestrained monkey during normal and perturbed gaze saccades reveal short-latency feedback and a sluggish rostral shift in activity. *Journal of Neuroscience*, 29(22), 7166-7180.
- de Carrizosa, M. A. D. L., Morado-Díaz, C. J., Miller, J. M., de la Cruz, R. R., & Pastor, Á. M. (2011). Dual encoding of muscle tension and eye position by abducens motoneurons. *Journal of Neuroscience*, 31(6), 2271-2279.
- Goffart, L., Burreilly, C., & Quinton, J. C. (2018). Neurophysiology of visually guided eye movements: critical review and alternative viewpoint. *Journal of Neurophysiology*, 120(6), 3234-3245.
- Goossens, H. H. L. M., & Van Opstal, A. J. (2000). Blink-perturbed saccades in monkey. II. Superior colliculus activity. *Journal of neurophysiology*, 83(6), 3430-3452.
- Hu, X., Jiang, H., Gu, C., Li, C., & Sparks, D. L. (2007). Reliability of oculomotor command signals carried by individual neurons. *Proceedings of the National Academy of Sciences*, 104(19), 8137-8142.
- Kaneko, C. R. (1996). Effect of ibotenic acid lesions of the omnipause neurons on saccadic eye movements in rhesus macaques. *Journal of neurophysiology*, 75(6), 2229-2242.
- Keller, E. L., Gandhi, N. J., & Vijay Sekaran, S. (2000). Activity in deep intermediate layer collicular

neurons during interrupted saccades. *Experimental brain research*, 130, 227-237.

Munoz, D. P., & Wurtz, R. H. (1995). Saccade-related activity in monkey superior colliculus. I. Characteristics of burst and buildup cells. *Journal of neurophysiology*, 73(6), 2313-2333.

Munoz, D. P., Waitzman, D. M., & Wurtz, R. H. (1996). Activity of neurons in monkey superior colliculus during interrupted saccades. *Journal of Neurophysiology*, 75(6), 2562-2580.

Rodgers, C. K., Munoz, D. P., Scott, S. H., & Paré, M. (2006). Discharge properties of monkey tectoreticular neurons. *Journal of neurophysiology*, 95(6), 3502-3511.

Soetedjo, R., Kaneko, C. R., & Fuchs, A. F. (2002). Evidence that the superior colliculus participates in the feedback control of saccadic eye movements. *Journal of neurophysiology*, 87(2), 679-695.

Smalianchuk, I., Jagadisan, U. K., & Gandhi, N. J. (2018). Instantaneous midbrain control of saccade velocity. *Journal of Neuroscience*, 38(47), 10156-10167.

Sparks, D. L., & Mays, L. E. (1980). Movement fields of saccade-related burst neurons in the monkey superior colliculus. *Brain research*, 190(1), 39-50.

Waitzman, D. M., Ma, T. P., Optican, L. M., & Wurtz, R. H. (1991). Superior colliculus neurons mediate the dynamic characteristics of saccades. *Journal of Neurophysiology*, 66(5), 1716-1737.

Please, find below my point-to-point rebuttal to all points raised by the reviewer.

Reviewer #1 (Remarks to the Author):

Extending the conclusions of Smalianchuk et al. (2018) to gaze shifts made with the unrestrained head, the author concludes that the "SC population activity encodes the instantaneous kinematics of the desired gaze shift through its firing rates" based on a "tight correlation" between the velocity profile and the spike density profile of individual neurons (L728-729, L734, L746-748, L752). He contends that "the finding that single-trial and single-unit firing dynamics at a central neural stage correlate well with the instantaneous motor output of a highly complex and nonlinear synergistic system (comprising the multiple-degrees of freedom oculomotor, head-motor, and vestibular systems; see Fig. 1B) is quite remarkable" (L769-771).

I removed all subjective markers like 'remarkable' and 'surprising' from the abstract and manuscript.

In spite of careful readings of his manuscripts, this reviewer remains unconvinced. In each of her/his reviews, she/he exposed several concerns that prevented her/him from admitting the author's conclusion. The author and this reviewer agree on several points. However, regarding the correlation between the spike density and the saccade velocity, this reviewer suspects that it is actually an artefact for the following reasons.

1) From looking at the figures (Fig. 3C: activity of neuron Sa1007, Fig. 4B, Fig. 10B and Fig. S1: activity of neuron Sa3006, Fig. S15: activity of neuron Sa0107), the reader is led to the impression that all saccade-related neurons in the SC cease emitting action potentials shortly before the gaze shift ends. However, this inference may result from the author's choice of example neurons. Indeed, all saccade-related neurons in the SC do not exhibit an abrupt cessation of activity shortly before saccade end. In the head-unrestrained monkey, Choi and Guitton (2009) documented several neurons that continue to fire after the end of the gaze shift (see the middle row of their Figure 6C). In the head-restrained monkey, numerous studies documented neural discharges that persist after saccade end (Anderson et al. (1998); Goossens and van Opstal (2000); Keller et al. (2000), Munoz and Wurtz (1995), Munoz et al. (1996); Rodgers et al. (2006); Sparks and Mays (1980); Waitzman et al. (1991)). The author did not document how many neurons exhibited this persistent activity after gaze saccade end. It is also unclear whether his analysis was only restricted to the so-called "clipped" saccade-related cells.

The plot in Fig. 4B was made in response to the explicit request of the reviewer in the first rebuttal. At no instance in the (earlier or later versions of the) manuscript it was suggested that SC cells only fire from 20 ms before gaze-shift onset to 20 ms before gaze-shift offset. Also in my recorded population, such cells are rare, although a considerable proportion stopped their firing just prior to or near the gaze-shift offset. Nearly all cells have a prelude, and some also persist to deliver some low-frequency post-saccadic spikes. This can even be observed in Fig. 4B, which I now also indicate in the text. This was also the case for the Goossens and Van Opstal (2006) study, where we developed the concept of the dynamic spike-count model that is tested in this paper.

To test the possibility that SC firing rates may carry a code for the instantaneous (gaze-)saccade kinematics, one should follow a consistent analysis and stick to it. In this case, I followed up on our earlier work with the head-restrained monkey (Goossens and Van Opstal, 2006, 2012). To test the consequences of the spike-count concept for eye-head gaze shifts with considerable variability in their kinematics I here followed exactly the same procedures. The hypothesis requires, however, that only spikes are included within a time window that equals the saccade duration and is taken at a reasonable lead time (somewhere between 12 and 20 ms; here, I took 20 ms, like in the Goossens study). Indeed, this idea implicitly incorporates the concept of the omnipause neurons as a gate that is opened whenever the saccade is made and that is closed during inter-saccade fixation intervals. Which mechanism(s) decide(s) on the exact timings of opening and closing the omnipause gate is left unspecified in the model, although we have speculated in our earlier studies that a cerebellar-brainstem pathway might be involved. Thus, pre- and post-saccadic SC spikes falling out of the pause-cell mediated saccade window cannot contribute directly to the motor output as they do not have access to the saccadic burst generator and the motor circuitry.

If one would include all spikes delivered by the SC cell and calculated something like a mean firing rate, or a peak firing rate, or the total number of spikes over the entire (or a fixed part) of the activity, the only output measure that consistently relates to these quantities is the saccade vector. This type of analysis has been done by many others, and I had no intention to repeat that here. Meaningful correlations with the instantaneous movement kinematics cannot be made with such analyses. These can only be done when the signals refer to the same time frame. In Goossens and Van Opstal (2000) we showed, however, that the number of spikes in the saccade-determined time window is by far the best descriptor for the saccade properties, which remains invariant for fast as well as very slow, blink-perturbed, saccades. Here, I show that the same holds for fast and slow eye-head gaze shifts.

2) If the time course of the instantaneous spike density and the time course of "gaze-track" velocity look remarkably similar for the neuron (Sa0107) illustrated in Fig. 10 A-B, it is also because the spikes that preceded the saccade-related burst were removed. The graph at the bottom of Fig. 10A shows that twenty milliseconds before gaze onset, the firing rate rises from 0 to 200 spikes per second. However, examination of Fig. 4A (same neuron) reveals that such an enhancement (from zero to 200 spikes/s) is rare. It seems that the author removed the spikes that were emitted before the onset of an analysis interval (elapsed from 20 ms before gaze saccade onset to 20 ms before gaze saccade end). However, numerous studies (see references listed above) reported collicular saccade-related neurons that emit spikes sometimes within a long prelude before saccade onset.

See my comment above.

Thus, by selecting only the spikes emitted from 20 ms before saccade onset to 20 ms before saccade end, the spike density exhibits a rising phase like the acceleration part of the velocity profile, and a decline like the deceleration part of the velocity profile. The correlation between the instantaneous firing rate and the gaze velocity may be the consequence of selecting a portion of the neuron's activity.

This may be partly true for spikes in a narrow time bin around $t = -20$ ms, but the argument of fake correlations with gaze velocity does not hold for arbitrary time windows. It is precisely for this reason that I included the re-shuffling analysis, which shows that the peak with high correlations disappears when the spike train is no longer correlated with its own associated gaze shift. To (hopefully) further convince the reviewer, I now also repeated the analysis of the spike-density correlations after including prelude activity up to 100 ms before saccade onset. I show the result for the discussed cell sa0107, which has prominent pre-saccadic activity, in new Supplemental Figure S12. It can be seen that the effect on the spike-density function around the -20 ms time point is quite limited, and, as expected, has little detrimental influence on the correlations with gaze-velocity. I refer to this point in a new section in the Methods (see below).

The author may wish to explain to the readers that the premotor neurons are sensitive to the spikes that collicular neurons emit during a specific time interval, that this interval starts 20 ms before saccade onset and terminates 20 ms before saccade end. He may wish to add that the onset and the end of this interval is determined by the pause of firing from a specific group of inhibitory neurons located in the nucleus raphe interpositus: the so-called omnipause neurons. If so, then the author should warn the readers that this scenario remains controversial. Indeed, if this hypothesis were true, then experimentally increasing the pause duration should lead to hypermetric saccades. Empirical studies actually show the lesion of these neurons does not lead to dysmetric saccades (Kaneko 1996; Soetedjo et al. 2000). The author may also explain why these results are not convincing.

I now included a new paragraph in the Methods section ("Neural Response Selection") that refers to this topic. I mention the Paré and Guitton (1998) study, which supports the idea that the omnipause neurons are silent with high timing precision during eye-head gaze shifts. However, I strongly feel that an excessive discussion on the results of lesion and microstimulation experiments performed in the raphe nucleus falls beyond the scope of my study.

3) Moreover, plotting the number of trials (or responses) as a function of the correlation coefficient between spike density and gaze velocity (Fig. 10) does not teach us anything about the proportion of neurons that were concerned. Fig. 10C shows that for one neuron (Sa0107), the correlation coefficient ranged from 0.8 to 1.0 in approximately 330 trials out of a total number of 664 trials. Fig. 10D shows that for 20 best-recorded cells, the correlation coefficient ranged from 0.8 to 1.0 in approximately 1022 trials out of 3981 trials. Thus, 32% (330/1021) of the correlation coefficients that ranged from 0.8 to 1.0 are only due to neuron Sa0107. How many neurons account for the remaining 691 correlations? Three neurons like Sa0107 would be sufficient to account for the 1021 trials. In other words, the claim of a "tight correlation" between the velocity profile and the spike density profile of individual neurons" (L728-729, L734, L746-748, L752) is a hasty conclusion if it concerns a small percentage of neurons. Three neurons out of 20 best recorded cells or three neurons out of 43 cells correspond to a small percentage of cells (15% and 7 %, respectively).

I thank the reviewer for drawing my attention to this potentially devastating point, if indeed correct. However, most cells yielded trials with high correlations. The level at which a correlation becomes meaningful is somewhat arbitrary, but I now provided more detailed information regarding the number of cells that yielded correlations above $r=0.71$, together with the proportion of responses of the associated data set for the 20 best-recorded cells in the new table S2 in Supplemental Materials. This level of correlation explains at least 50% of the variation in the relationships. The table shows that for 15/20 cells the percentage of trials in which the correlation exceeded 0.71 was 39%, or higher. Only for 4 cells it was lower than 10%. Overall, the total number of trials with $r>0.71$ for the 20 cells was $1901/3981 = 48\%$. For the mentioned neuron, sa0107, this occurred in $287/350$ trials (82%). In the table it can be seen that this particular neuron thus contributed $287/1901 = 15\%$ to the high-end correlations in the histogram of Fig. 10D (for $N_{\text{spk}} \geq 15$). This is now also mentioned in the revised text.

For these reasons and for other reasons exposed in her/his previous reviews (see also reservations expressed by Goffart et al. 2018 about the encoding of velocity by central neurons), this reviewer thinks that the author's conclusions are not yet sufficiently founded.

I hope that my additional analyses and added information in text, new Table, and extra Figure, have now (at least partially) taken away some of the reviewer's reservations.

However, for the sake of pedagogy to the uninformed readers and for the sake also of reminding the tremendous knowledge acquired by previous neurophysiological studies in the awake and trained monkey, this reviewer supports the publication of the author's work as long as the readers are warned about its limitations and shortcomings.

I sincerely thank the reviewer for the supportive intellectual tour de force in reviewing my paper. The many to-the-point remarks have challenged me to sharpen my arguments and dig deeper into my analyses. I believe that this enterprise has greatly improved the paper.

It's perhaps unfortunate that space limitations prevent me from including all extra material into the manuscript, but fortunately there is the possibility for refer readers to the Supplemental Material.

Minor comments:

Abstract:

The last sentence sounds weird. The author may remove "we hypothesize that" L5: perhaps "amplitude and direction" instead of "coordinates"? **done**

Introduction:

L157: in Goffart et al. (2017), the saccades were not triggered during smooth pursuit tracking but from fixating a static target. The author could replace "occurring during smooth pursuit tracking" by "toward a moving target". Moreover, the word "always" could be inserted between "does not" and "incorporate" because the difference of firing rate between "centrifugal" and "centripetal" did not concern all cells.

I incorporated the suggestions of the reviewer.

L164: this reviewer suggests removing "FEF-mediated" because one cannot exclude the involvement of the cortico-ponto-cerebellar channels originating in the parietal eye fields. **done**

L166: the author might add "in the SC" after "microlesions". **done**

L173: perhaps "primary" instead of "central"? **done**

L174: the author might add "other neural regions such as" between "taken over by" and "the FEF". **done**

Results:

L272: the author might add "various" before "directions" **done**

L418: For the readers who are not familiar with the notion of "multiplicative noise", it would be helpful to explain its meaning.

It is described in the text by the definitions of additive (constant background noise) vs. multiplicative (signal-dependent) noise and made quantitatively explicit by the (un-numbered) equation. The equation is not numbered as it is not actually used in the analyses that follow. The functional significance of multiplicative noise is debated, but certain control theories hold that the presence of multiplicative noise underlies the main-sequence properties of saccades and that it renders the system more robust and precise. Thus, multiplicative noise (which is signal-strength dependent) could be an inherent functional property at certain stages of the neural encoding. I here show that it is present in the activity (quantified as number of spikes in the saccade-related burst) of most of my recorded SC neurons.

L486: "by fitting a linear regression line through the measured phase trajectory" instead of "by fitting linear regression lines through the measured phase trajectories"? **replaced by "each measured phase trajectory"**

L496: "as a function of" instead of "as function of" **done**

L5601: what does "c.q." mean? **Here, it means: "or, equivalently," (changed)**

L506-508: the author forgot to remove his comment. **done**

L610-611: "for the trial (#670) that yielded the largest number of spikes" instead of "for trial 670 that yielded the largest number of spikes" **done**

Discussion:

L836: "Moschovakis" instead of "Moschiovakis" **done**

References:

Page 28, second reference: "Moschovakis" instead of "Moschiovakis" Page 29, 16th reference: "Sparks DL" instead of "Sparks DJ" **done**

Methods:

L927: the minimum value of the range of successful trials is unclear: "1-0" L950: "gaze" instead of "eye-". **Check Table 1 for monkey P!**

L1065: "a SC neuron" instead of "an SC neuron" **done**

L1072: "velocity" instead of "vlocity" **done**

L1137: "as a function of" instead of "as function of" **done**

Figure 4A:

The dashed line that indicates the gaze onset does not look cyan but green. **Changed**

Figure 10A:

It seems that the panel A does not show all action potentials that the cell (Sa0107) emitted but only those that were emitted during the interval elapsed from 20 ms before gaze shift onset to 20 ms before gaze shift end (small black lines).

This is indeed the case; I here only included the spikes within the analysis window from 20 ms before gaze-shift onset to 20 ms before gaze-shift offset. I indicate this now explicitly in the figure caption.

Figure 4A shows numerous examples in which the same cell exhibits a prelude activity. The author should warn the reader that the curve representing the purportedly “instantaneous” firing rate (black) at the bottom of Fig. 4A is rare, or that it is not the true instantaneous firing rate.

I suppose the reviewer refers to Figure 10A? Indeed, in all analyses of Figures 4C – 13, I only took the spikes from ON-20 to OFF-20. I noted that I forgot to replace the ordinate text in Fig. 10A by Spike density (SD) and Cumulative Spike Density (CSD). I have now done so, and also replaced ‘firing rate’ by ‘spike density’ wherever required. So, indeed, I did not calculate the ‘true’ firing rate in this plot.

This reviewer wonders whether it is a fictive instantaneous firing rate, merely created by selecting the spikes emitted during a time interval that the author chose to elapse from 20 ms before gaze shift onset to 20 ms before gaze shift end.

Yes, it shows the spike density that is calculated from the windowed spikes in ON-20 to OFF-20 ms. In Methods I explain this point explicitly in a new section named ‘Neural Response Selection’, where I argue why I do this, and I also warn the reader that (and why) I did not correct for a (small) influence of potential spikes in the prelude activity prior to the ON-20 ms time point.

To quantify its potential effect, I re-calculated the correlation coefficients after including the prelude and OFF-20 ms influence, and the result showed a negligible difference. I illustrate this for the example neuron Sa0107 (from Fig. 4) in Figure S12, which shows a single trial, the two correlations plotted against each other for all 664 trials, and the traces for the 25 highest-activity trials for this neuron (including the one in Fig. S12A). The lack of a strong influence on the correlation is understood from the fact that the spike density kernels from spikes at previous time points have a very small influence (Gaussian decay) on time points further on (only a few ms) whereas the correlation coefficients typically rely on the inclusion of 60 - 200 data samples (typical durations of the (large) gaze shifts; see Figure 2 for the amplitude-duration data). This also holds for the (fewer and much less high-frequency) spikes around the saccade offset.

Table 1: the eye position sensitivity “EPSILON ZERO” should be replaced by “EPSILON” to be consistent with the definition in the methods (L1027) and the numerous instances elsewhere in the text (pages 11 and 14). *done*
Based on the major comment, it would be useful to add information about the firing rates at two times: 20 ms before saccade onset and at saccade end. *done*

Figure S10:
L1268-1269: there is no Fig. 6C *It should refer to Fig. 7C*